# Linguistic Properties and Model Scale in Brain Encoding: From Small to Compressed Language Models

## Abstract

Recent studies have shown that full-precision Transformer-based large language models (LLMs) increasingly improve predictions of human brain activity as model parameters are scaled. However, the corresponding growth in size and computational cost limits their interpretability and practical deployment, particularly in applications such as brain-computer interfaces (BCIs), which demand low-latency and efficient models. To address this gap, two efficiency-oriented approaches that have emerged can be used: (i) adopting small language models (SLMs), which achieve competitive performance at substantially lower computational cost, and (ii) compressing LLMs through quantization, which reduce computational demands while retaining much of their original capacity. However, it remains unclear whether such SLMs or compressed LLMs can effectively capture brain-relevant representations and achieve brain alignment comparable to that of full-precision LLMs. Specifically, our study is motivated by four key questions: (i) can compressed LLMs preserve brain alignment, which is critical for deployment; (ii) how do compressed LLMs compare to SLMs, informing trade-offs for practical applications; and (iii) if ultra-low-resource applications demand even smaller footprints, can compressed SLMs still maintain alignment with brain activity? (iv) Which aspect of linguistic competence (discourse, morphology, syntax, semantics, or reasoning) most strongly influences brain alignment as model size or quantization method varies? In this work, we systematically evaluate LLMs (7B), SLMs (1B and 3B), and their quantized variants to assess how model scale and compression jointly affect brain alignment, using fMRI recordings collected during naturalistic story listening. Our findings indicate that 3B SLMs achieve brain prediction performance comparable to both full-precision and compressed LLMs across whole-brain and core language regions. In contrast, 1B SLMs show a significant drop in brain alignment, particularly in semantic-processing regions. Notably, while most quantization methods preserve alignment, GPTQ quantization leads to reduced brain alignment across both LLMs and SLMs. Finally, benchmarking with the FlashHolmes suite shows that quantization primarily degrades discourse, syntax, and morphology, while leaving overall brain alignment intact.

## 1 Introduction

Modern language models (e.g., GPT*, BERT), though trained only on text, predict human brain activity to an impressive degree (Toneva & Wehbe, 2019; Schrimpf et al., 2021; Goldstein et al., 2022; Oota et al., 2022; Lamarre et al., 2022; Caucheteux & King, 2022; Antonello et al., 2021; Tuckute et al., 2023; Oota et al., 2024a). More recently, studies have examined neural scaling laws (Kaplan et al., 2020; Hoffmann et al., 2022; Li et al., 2024) in fMRI encoding, showing that brain prediction performance improves as model size and training data increase (Matsuyama et al., 2023; Antonello et al., 2024; AlKhamissi et al., 2025). However, scaling up language models leads to a proportional increase in computational cost (Faiz et al., 2024; Diaz & Madaio, 2024; Villalobos et al., 2024), limiting their interpretability and constraining deployment in resource-sensitive domains such as neuroAI and brain–computer interfaces (BCIs).

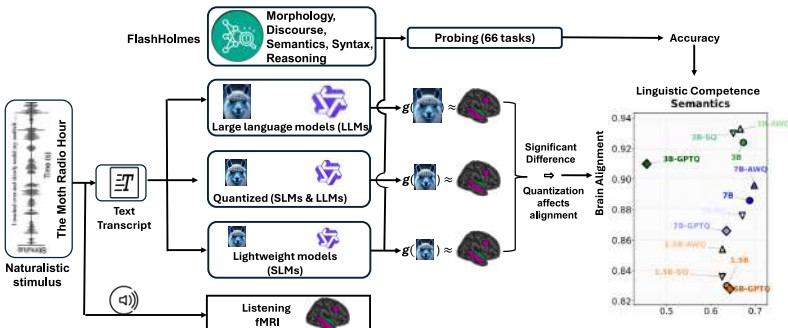

Figure 1: Approach to test the effect of linguistic competence on the alignment between language models (LLMs, SLMs, and their quantized variants) and brain recordings. Participants listened to naturalistic Moth Radio Hour stories in English while fMRI responses were recorded. The corresponding transcripts were processed through the models to extract internal representations, which were mapped to fMRI responses using voxel-wise encoding models to estimate brain alignment. We then compared brain alignment across large, small and their respective quantized models, testing whether quantization significantly reduces alignment and thus altering linguistic competence. In parallel, the same models were evaluated on the FlashHolmes benchmark (66 tasks spanning morphology, syntax, semantics, discourse, and reasoning) to assess their linguistic competence. We then compared brain alignment with linguistic competence across large, small, and quantized models, testing how compression affects both brain predictivity and task-level linguistic skills.

To reduce computational costs, researchers have pursued two efficiency-oriented approaches: (i) compressing LLMs through pruning, distillation, or quantization, and (ii) developing small language models (SLMs) that achieve competitive performance compared to their larger off-the-shelf models with fewer resources (Touvron et al., 2023; Gemini Team et al., 2023; Yang et al., 2024). Among these, quantization is particularly attractive because it can be applied post-training with minimal overhead, making it a practical choice for efficient deployment especially with modern hardware accelerators (e.g., INT8 support). Several popular quantization techniques, such as activation-aware weight quantization (AWQ) (Lin et al., 2024), post-training compression for generative pretrained transformers (GPTQ) (Frantar et al., 2023), and SmoothQuant (Xiao et al., 2023), significantly reduce model size and memory usage, often preserving performance on standard language tasks (Namburi et al., 2023; Kuzmin et al., 2023).

Despite these advancements, most prior work in brain encoding has focused on large, full-precision language models (LLMs) (Antonello et al., 2024; AlKhamissi et al., 2025), while in the AI community, quantization has mainly been evaluated for efficiency and benchmark performance on standard NLP tasks (Namburi et al., 2023; Kuzmin et al., 2023). It remains unclear whether efficiency-oriented small language models (SLMs) or compressed LLMs can capture brain-relevant representations to the same degree. This motivates us to explore whether SLMs can retain brain alignment comparable to LLMs, while reducing computational demands. In addition, in the context of brain-inspired language modeling, it is equally important to examine whether compression techniques alter the linguistic competence of the model and, more critically, whether they lead to the loss of brain-relevant information. Yet, to date, no systematic investigation has explored how different quantization techniques affect language model performance specifically in the context of brain alignment. To address this gap, we systematically evaluate SLMs, LLMs and their quantized variants in the context of fMRI encoding, offering a structured understanding of how reducing LM scale and compression impacts both linguistic competence and brain alignment.

Using brain recordings of participants listening to naturalistic stories from the moth-radio-hour dataset (Deniz et al., 2019), we evaluate the linguistic competence and brain alignment of SLMs, LLMs and their quantized variants. We focus on two small language models: LLaMA-3.2 (1B and 3B) (Touvron et al., 2023), and Qwen2.5 (1.5B and 3B) (Yang et al., 2024), their larger versions (LLaMA-3.1 8B, and Qwen-2.5 7B), and quantized versions (AWQ, GPTQ and SmoothQuant). To analyze linguistic competence, we use the FlashHolmes benchmark (Waldis et al., 2024), consisting of 66 datasets grouped into 5 categories: morphology, discourse, semantics, syntax, and reasoning. This allows us to investigate which types of linguistic information are differentially captured by large models, small models, and their quantized counterparts, and whether these linguistic properties

remain relevant to brain activity. Overall, our work represents a foundational exploration into how model compression methods like quantization, affects the encoding performance of small and large models. We demonstrate how quantized representations differ between SLMs and LLMs, and how these differences relate to both linguistic competence and brain alignment (see Fig. 1 for workflow). Our work attempts to answer the following research questions.

1. Can SLMs achieve brain alignment comparable to, or better than, LLMs and their quantized variants?
2. How do different quantization methods (AWQ, GPTQ, SmoothQuant) impact the brain alignment of both LLMs and SLMs?
3. For ultra-low-resource scenarios, can compressed SLMs with even smaller footprints still maintain alignment with brain activity?
4. How do variations in model size or quantization method affect different aspects of linguistic competence (discourse, morphology, syntax, semantics, and reasoning) in shaping brain alignment?

To address these research questions, we evaluate models along two dimensions: (1) **Brain Alignment**: If SLMs and quantized models show brain alignment comparable to larger models, they can serve as efficient alternatives for predicting brain activity. Conversely, very small models are expected to show reduced alignment, and post-training quantization may further affect performance. (2) **Linguistic Competence**: If small or quantized models lose linguistic competence on specific tasks, even when brain alignment remains intact, this would indicate that compression disproportionately affects certain linguistic properties (e.g., syntax, discourse).

Our analysis leads to the following important conclusions: (1) Quantized variants of LLMs preserve most of their brain alignment, especially under AWQ and SmoothQuant, with minimal loss. By contrast, GPTQ consistently reduces alignment by 5–7%, particularly in semantic-processing regions, underscoring that not all compression methods are equally viable. (2) 3B SLMs achieve brain alignment comparable to their larger 7B–8B LLMs, while 1B/1.5B models consistently show significant drop in brain alignment. This suggests that 3B SLMs retain brain-relevant information to a similar degree as large models, whereas smaller models largely lose it. The preserved alignment in 3B models appears to be driven by their ability to maintain semantic representations that correlate strongly with activity in higher-order regions such as the inferior frontal gyrus (IFG), angular gyrus (AG), and posterior cingulate cortex (PCC). (3) When comparing compressed 3B SLMs (via AWQ and SmoothQuant) with their full-precision counterparts and with LLMs, we find that they achieve comparable brain alignment while requiring substantially fewer resources. This positions 3B SLMs as a strong alternative, offering alignment on par with compressed LLMs but at much lower cost. (4) A trade-off emerges between linguistic competence and brain alignment: AWQ and SmoothQuant largely preserve both linguistic competence and brain alignment across SLMs, LLMs and their quantized variants , GPTQ disproportionately harms higher-order linguistic properties such as discourse, reasoning, and syntax, and 1B SLMs highlight the limits of scale by maintaining task performance but failing to capture brain-relevant geometry.

Overall, our findings refine existing scaling laws by showing that 3B SLMs preserve brain alignment comparable to 7B LLMs, while 1B SLMs and certain quantization methods (e.g., GPTQ) disproportionately reduce brain alignment.

## 2 RELATED WORK

**SLMs and quantized language models.** Recent research has examined how neural scaling laws (Kaplan et al., 2020; Hoffmann et al., 2022; Diaz & Madaio, 2024) formalize the relationships between model performance, size, data, and computational resources, revealing that performance improves as these factors are scaled. In contrast, SLMs such as LLaMA-3.2, Qwen-2.5, MiniCPM, DeepSeek-R1, and Gemma-2B have been developed to balance performance with efficiency, enabling deployment in resource-constrained settings (Touvron et al., 2023; Yang et al., 2024; Hu et al., 2024; Guo et al., 2025; Gemma Team et al., 2024). Moreover, these models achieve competitive performance with significantly fewer parameters, suggesting that efficiency-oriented design can partially offset the advantages of scale.

To further reduce computational demands, a wide range of model compression techniques have been proposed, including pruning, quantization, knowledge distillation, parameter sharing, and matrix de-

composition (Gupta & Agrawal, 2022). While pruning can also yield strong performance without reducing memory footprint (unstructured pruning) or reduce memory at the expense of accuracy (structured pruning), and distillation often requires task-specific retraining. In contrast, quantization can be applied post-training with minimal overhead, while generally maintaining accuracy on standard NLP benchmarks (Gupta & Agrawal, 2022; Lin et al., 2024; Frantar et al., 2023; Xiao et al., 2023). For these reasons, our work focuses on quantization as a practical and widely adopted approach to model compression in the context of brain alignment.

**Scaling-laws of language models in fMRI.** Our work also relates to a growing literature that investigates the alignment between human brains and language models. A number of recent studies have used LLMs to predict both text-evoked and speech-evoked brain activity to an impressive degree (Antonello et al., 2024; Matsuyama et al., 2023; AlKhamissi et al., 2025). For brain encoding, Antonello et al. (2024) compared SLMs (e.g., OPT-125M (Zhang et al., 2022)) with much larger ones (up to OPT-175B and LLaMA-33B/66B). They found that using contextual embeddings from the larger models boosts encoding-model performance by roughly 15% relative to their smaller counterparts. Recently, AlKhamissi et al. (2025) further examined whether the functional specialization observed in the human brain can be identified in LLMs and they successfully identified a language network analog in all LLMs. While prior works by Antonello et al. (2024); AlKhamissi et al. (2025) established scaling laws for brain encoding with full-precision models, our work is the first to systematically investigate the brain alignment of SLMs and LLMs, and the impact of post-training compression on these neural alignments.

**Linguistic properties and brain alignment.** Our work also relates to previous studies that investigated the linguistic properties encoded in language models and their impact on predicting brain recordings. Several studies adopt a direct approach by perturbing the internal representations of language models using residual methods, thereby estimating the effect of a specific feature on brain alignment (Toneva et al., 2022b; Oota et al., 2024b;a). In contrast, other studies follow an indirect approach, first estimating the brain alignment of a model and then independently evaluating its performance on a natural language processing task (e.g., approaches presented by Schrimpf et al. (2021); Goldstein et al. (2022)). Our study is complementary to both: we consider quantization as a way of optimizing internal representations, while following an indirect approach by independently examining linguistic competence across 66 linguistic tasks and estimating brain alignment.

## 3 METHODOLOGY

### 3.1 NATURALISTIC BRAIN IMAGING DATASET

We use a publicly available fMRI dataset (Deniz et al., 2019) collected while nine participants listened to narrative stories from the Moth Radio Hour. The dataset consists of 3,737 training and 291 testing samples (TRs: Time Repetition) each, chosen for their unique auditory and high-level language processing in the brain. Following Deniz et al. (2019), we examine this dataset using the Glasser Atlas' multi-modal parcellation of the cerebral cortex, targeting 180 ROIs per hemisphere (Glasser et al., 2016). This includes one early sensory processing region (early auditory) and eight language-relevant regions, encompassing broader areas of linguistic processing: angular gyrus (AG), lateral temporal cortex (ATL and PTL), inferior frontal gyrus (IFG), inferior frontal gyrus orbital (IFGOrb), middle frontal gyrus (MFG), posterior cingulate cortex (PCC), and dorsomedial prefrontal cortex (dmPFC) based on the Fedorenko's language parcels (Milton et al., 2021; Desai et al., 2023). More details about the dataset and ROI functionality details are reported in the Appendix A and Table 2.

**Estimating cross-Subject prediction accuracy.** To account for the intrinsic noise in biological measurements, we adapt the method proposed by Schrimpf et al. (2021); Oota et al. (2024a); AlKhamissi et al. (2025) to estimate the ceiling value for a model's performance for the moth-radio-hour fMRI dataset. Note that the estimated cross-subject prediction accuracy is based on the assumption of a perfect model, which might differ from real-world scenarios, yet offers valuable insights into model's performance. We present the average cross-subject prediction accuracy across voxels for the *listening fMRI* dataset in Appendix B.

### 3.2 SMALL LANGUAGE MODELS AND THEIR LARGER COUNTERPARTS

To investigate whether small language models align with human language processing in the brain and have better or similar alignment with their larger counterpart models, we consider two popular

Table 1: Pretrained Transformer-based language models: SLMs, LLMs and their quantized variants.

| Model Family | SLMs | | LLMs | | Model Family | SLMs (GB) | | | LLMs (GB) | | |
|---|---|---|---|---|---|---|---|---|---|---|---|
| | Size | Layers | Size | Layers | | AWQ | GPTQ | SmoothQuant | AWQ | GPTQ | SmoothQuant |
| LLaMA-3.2 | 1B (2.47GB) | 16 | 8B (16.1GB) | 28 | LLaMA-3.2 | 1.56 | 1.02 | 2.02 | 5.73 | 5.70 | 9.08 |
| | 3B (6.4GB) | 28 | | | | 3.04 | 2.26 | 4.40 | | | |
| Qwen2.5 | 1.5B (3.1GB) | 28 | 7B (15.1GB) | 28 | Qwen2.5 | 1.61 | 1.15 | 2.25 | 5.57 | 5.58 | 8.67 |
| | 3B ( 6.1GB) | 36 | | | | 2.69 | 2.10 | 4.02 | | | |

modern SLMs which are publicly available on Huggingface (Wolf et al., 2020): LLaMA-3.2 (1B and 3B) (Touvron et al., 2023), and Qwen2.5 models (1.5B and 3B) (Yang et al., 2024). We provide more details, including model-parameters and layers details in Table 1.

**Extracting text representations.** For text transcripts from the Moth Radio Hour dataset, we follow previous work in extracting hidden-state representations from each layer of the language models for a fixed-length input (Toneva & Wehbe, 2019; Aw & Toneva, 2023; Oota et al., 2024b;a). To obtain the stimulus features from these pretrained models, we constrain the tokenizer to use a maximum context of 20 words. Given the constrained context length, each word is successively input to the network with at most $C$ (=20) previous words. For instance, given a story of $M$ words and considering the context length of 20, while the third word's vector is computed by presenting ($w_1$, $w_2$, $w_3$) as input to the network, the last word's vector $w_M$ is computed by presenting the network with ($w_{M-20}$, ..., $w_M$). The pretrained Transformer model outputs token representations at different layers. We use the #words $\times$ 768 dimension vector from each hidden layer to obtain word-level representations from each pretrained Transformer language model. The preprocessing and HRF delays is detailed in Appendix A.

## 3.3 POST-TRAINING QUANTIZATION TECHNIQUES ON LANGUAGE MODELS

Quantization can dramatically reduce memory usage and accelerate inference by mapping model weights and activations to lower-precision formats such as INT8, INT4, or FP8 arithmetic. In this work, our goal is not only to assess efficiency but also to examine how quantization impacts linguistic competence and brain alignment. To achieve this, we perform three widely-used quantization techniques: (1) Activation-aware Weight Quantization (AWQ) (Lin et al., 2024), which adjusts weight scales using activation statistics to enable highly accurate 4-bit or 8-bit compression with minimal quality loss; (2) GPTQ (Frantar et al., 2023): applies post-training, gradient-guided weight quantization method that delivers near-lossless INT8/INT4 speed-ups; and (3) SmoothQuant (Liu et al., 2024; Xiao et al., 2023), which jointly quantizes weights and activations by equalizing variance to reduce memory usage and latency while preserving accuracy. We provide more details, including model-parameters after quantization in Table 1.

## 3.4 FLASH-HOLMES BENCHMARK

FlashHolmes is a streamlined version of the Holmes benchmark (Waldis et al., 2024), designed to efficiently evaluate the linguistic competence of language models. The FlashHolmes benchmark covers nearly 200 probing datasets spanning 66 linguistic tasks. The linguistic tasks are grouped into five major categories: (1) Morphology (19 tasks, e.g., subject–verb agreement and irregular word forms) (Warstadt et al., 2020; Huebner et al., 2021), (2) Syntax (75 tasks, e.g., constituent labeling and filler-gap dependencies) (Conneau et al., 2018; Warstadt et al., 2020), (3) Semantics (67 tasks, e.g., semantic role labeling and natural language inference) (Wang et al., 2018), (4) Discourse (28 tasks, e.g., coreference resolution and discourse relation prediction) (Webber et al., 2019), and (5) Reasoning (19 tasks, e.g., paraphrasticity with negation and antonyms) (Vahtola et al., 2022). Overall, these linguistic tasks allow FlashHolmes to probe a wide spectrum of linguistic phenomena, making it a suitable tool for evaluating both SLMs and LLMs and their quantized variants. FlashHolmes maintains a high-ranking precision for evaluating models while requiring only about 3% of the computational resources needed for the full Holmes benchmark. This makes it a practical tool for quick and frequent evaluations of language models, especially during research and development or for comparing a large number of models efficiently.

## 4 EXPERIMENTAL SETUP

**Voxel-wise encoding model.** To explore how language model representations encode in the brain when listening to narrative stories, we use layer-wise representations from SLMs, LLMs, and quantized models and using them in a voxelwise encoding model to predict brain responses. Our first hypothesis is that if SLMs achieve brain alignment comparable to their larger counterparts, they can serve as efficient alternatives for predicting brain activity. If reducing the model size via quantization leads to a significant drop in brain alignment compared to the uncompressed model, we interpret this as evidence that compression removes brain-relevant information. Thus, the quality of these encoding model predictions indicates how strongly language model representations capture brain-relevant information. To perform voxel-wise encoding, we train an fMRI encoding model using bootstrap ridge regression (Tikhonov & Arsenin, 1977) to predict the fMRI recording associated with each voxel as a function of the stimulus representations obtained from the language models. Before the bootstrap ridge regression, we first z-score each feature channel separately for training and testing. This is done to match the features to the fMRI responses, which were also z-scored for training and testing. Formally, at the time step (t), we encode the stimuli as $X_t \in \mathbb{R}^{N \times D}$ and brain region voxels $Y_t \in \mathbb{R}^{N \times V}$, where $N$ is the number of training examples, $D$ denotes the dimension of the concatenation of delayed 4 TRs, and $V$ denotes the number of voxels. To find the optimal regularization parameter for each feature space, we use a range of regularization parameters that is explored using cross-validation. The main goal of each fMRI encoding model is to predict brain responses associated with each brain voxel given a stimulus. The detailed hyperparameter settings and statistical significance tests are provided in Appendix C and Appendix D.

**Normalized alignment.** The final measure of a model's performance is obtained by calculating Pearson's correlation between the model's predictions and brain recordings. This correlation is then divided by the estimated cross-subject prediction accuracy and averaged across voxels, resulting in a standardized measure of performance referred to as normalized alignment. During normalized alignment, we select the voxels whose cross-subject prediction accuracy is $\geq 0.05$.

**Encoding and probing of Flash-Holmes benchmark.** To evaluate the linguistic competence of language models, we use the FlashHolmes benchmark (Waldis et al., 2024), as detailed in Section 3.4. For each task in FlashHolmes, we apply classifier-based probing to the internal representations of SLMs, LLMs, and quantized models. Our second hypothesis (as mentioned in Sec. 1) is that SLMs or quantized models may show reduced linguistic competence on specific tasks, even if overall brain alignment remains intact. In this case, FlashHolmes can reveal which linguistic properties (e.g., syntax or discourse) are disproportionately affected by compression. Thus, FlashHolmes complements voxel-wise encoding models by identifying what kinds of linguistic information are preserved or lost in small and quantized models.

To perform probing, following Waldis et al. (2024), we train a simple linear classifier on the representations obtained from SLMs, LLMs, and quantized models. Model performance is then aggregated across tasks within each linguistic category, producing scores that reflect the accessibility of different types of linguistic information in the representations.

## 5 RESULTS

### 5.1 EFFECTS ON BRAIN ENCODING PERFORMANCE (SLMS VS. LLMS VS. QUANTIZED LMS)

To examine whether SLMs achieve brain encoding performance comparable to larger models, we compare the voxelwise encoding performance of two families of language models (Qwen-2.5, and LLaMA-3.2), at different scales. For the small models, we also apply post-training quantization and measure normalized brain predictivity at both the whole-brain level and within language-specific regions. Fig. 2 shows average normalized brain predictivity across participants and layers.

**Wholebrain analysis.** Across the whole brain (Fig. 2 top-row), we find that 3B models across both the families consistently exhibit high degree of brain alignment that is not significantly different from their 7B and 14B counterparts, suggesting that SLMs at or above the 3B scale can capture brain-relevant representations as effectively as larger models. Furthermore, quantized variants of LLMs and 3B SLMs largely preserve whole-brain alignment, with AWQ and SmoothQuant maintaining performance while GPTQ leads to measurable degradation. In contrast, the smallest

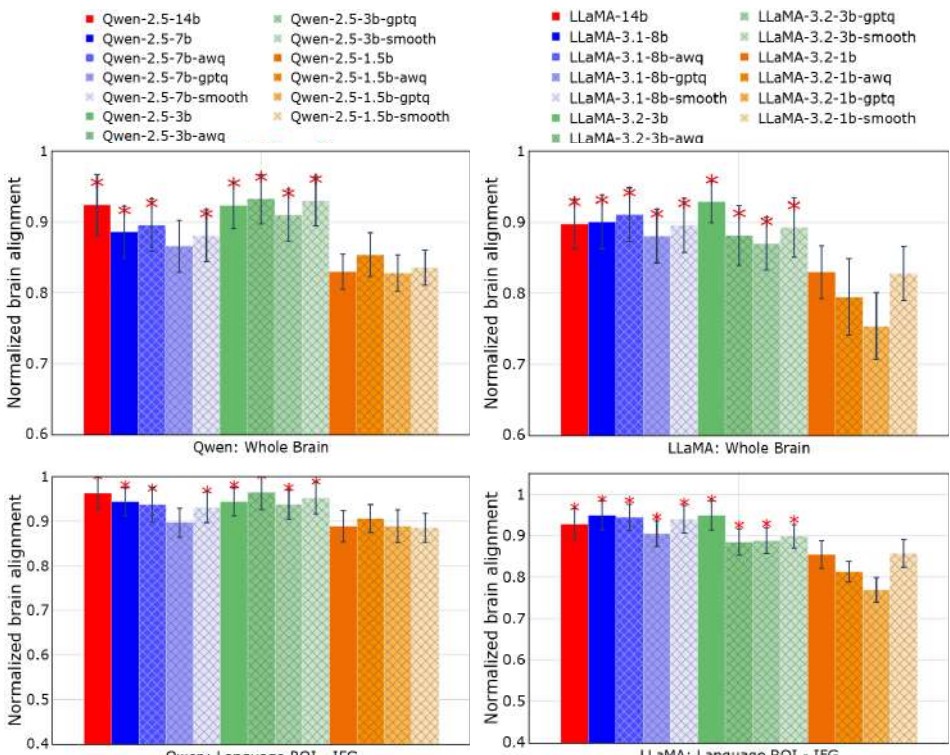

Figure 2: Normalized brain alignment was computed by averaging across participants, layers, and voxels. Red: 14b, Blue: 7b/8b, Green: 3b, Orange: 1.5b, Solid: full-precision SLMs/LLMs, Patterned: quantized models. * at a particular bar indicates that the model's prediction performance is significantly better than 1b/1.5b SLMs. The top row shows whole-brain normalized alignment, while the bottom row focuses on a language-selective ROI (IFG). Left plots correspond to Qwen2.5 and right plots to LLaMA-3.2. Plots for other regions are in Figs. 9 and 11 in Appendix E.

1B-1.5B models show a significant drop in alignment, indicating that models below 3B parameters lose predictive power for brain encoding, this is consistent across all two language families. We also group the quantized variants and present a comparison of the base vs. quantized models in Fig. 8 in Appendix E.

**Language ROI analysis.** In the language-selective ROI, IFG (Fig. 2, bottom row), we find that across the two language models: (1) both 3B SLMs and their larger counterparts exhibit very high brain alignment (2) retain most of their alignment even after applying quantization on 3B SLMs, and (3) 1B/1.5B SLMs show a significant drop in alignment, and indicating that models below the 3B scale lose predictive power for brain encoding in IFG. Together, these results suggest that models below 3B parameters lose predictive power for brain encoding, both in whole-brain analyses and in IFG, known for Syntax processing region. We present other language ROIs in Figs. 9 and 11 in Appendix E. We also group the quantized variants and present a comparison of the base vs. quantized models across language ROIs in Figs. 10 and 12 in Appendix E.

Interestingly, these patterns generalize to other semantic language ROIs such as AG, PCC, and dmPFC, where small models consistently underperform compared to 3B and larger versions. By contrast, in ROIs such as ATL, PTL, and MFG, alignment remains largely stable across model sizes and families, suggesting that these regions are less sensitive to model scale. Together, these findings highlight that semantic regions are particularly vulnerable to alignment loss in smaller models, whereas higher-level integrative regions (e.g., ATL, PTL, MFG) show robustness across scales.

**Effect of quantization on brain encoding performance.** When applying post-training quantization (AWQ, GPTQ, SmoothQuant), the 3B models maintain stable alignment across all techniques, with only marginal reductions relative to their uncompressed versions, as shown in Fig. 2. This demonstrates that efficiency gains from quantization do not significantly compromise neural predictivity at

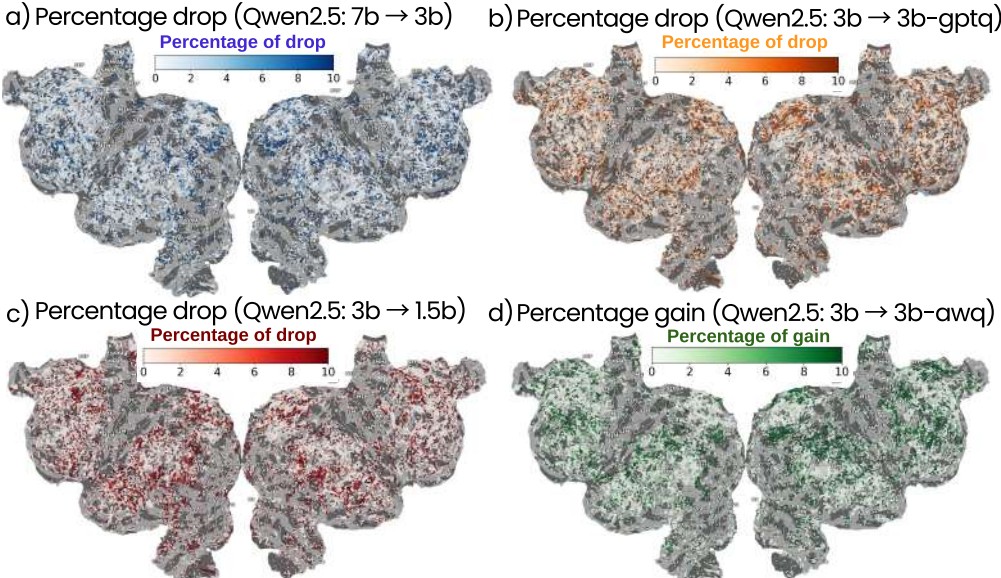

Figure 3: Qwen2.5: Percentage change in brain alignment across model scales and quantization methods, shown on the flattened cortical surface of a representative subject (subject-5). Blue, orange, and red voxels indicate regions of information loss ((a) LLMs → 3B SLMs, (b) 3B SLMs → 3B SLMs GPTQ, (c) 3B SLMs → 1.5B SLMs, respectively), (d) while green voxels highlight regions of improvement for 3B SLMs AWQ over 3B SLMs. White voxels denote regions with no change. Results for other participants for Qwen2.5 and LLaMA models are in Appendix F.

the 3B scale. By contrast, quantized 1B–1.5B models remain under-aligned, reinforcing that their limited capacity, rather than representational redundancy, is the primary bottleneck.

Fig. 3 illustrates voxel-wise percentage changes in brain alignment when transitioning across model scales and quantization strategies. We make the following observations: (i) Scaling down from LLMs to 3B SLMs (Fig. 3 (a)): reductions in brain alignment are negligible in the bilateral temporal lobe and remain under 5% in the parietal cortex and IFGorb. Large cortical regions remain white, indicating that 3B SLMs preserve brain-relevant representations comparable to LLMs. The blue-marked voxels are sparse and localized, suggesting only limited information loss. (ii) 3B SLMs to 3B SLMs GPTQ (Fig. 3 (b)): applying GPTQ leads to widespread orange-marked voxels, reflecting consistent losses across distributed cortical regions. While some areas remain preserved (white voxels), the extent of information loss is greater than that observed with downscaling alone, confirming that GPTQ disproportionately disrupts brain-relevant alignment. (iii) 3B SLMs to 1.5B SLMs (Fig. 3 (c)): relative to the 3B baseline, we observe extensive red-marked voxels, especially in temporal and language-related regions, indicating larger drops in alignment. This demonstrates the limits of scaling, as ultra-small models fail to capture brain-relevant representations. (iv) 3B SLMs → 3B SLMs AWQ (Fig. 3 (d)): AWQ produces localized green-marked voxels, indicating regions of improved alignment relative to the 3B baseline, particularly in the IFG and AG. Most of the cortex remains unchanged (white), suggesting that AWQ maintains representational fidelity while offering modest regional gains.

Taken together, these results support Hypothesis 1: 3B SLMs achieve brain alignment comparable to larger LLMs and remain robust under AWQ and SmoothQuant quantization. By contrast, 1B–1.5B models consistently underperform, and GPTQ disproportionately disrupts brain-relevant alignment. Importantly, this pattern holds across both the Qwen and LLaMA model families.

## 5.2 IMPACT OF LINGUISTIC COMPETENCE IN LANGUAGE MODELS AND BRAINS

While the previous analyses showed that 3B SLMs maintain brain alignment comparable to LLMs, reduced alignment is observed for 1B SLMs and for quantized variants, particularly those using GPTQ. To further investigate the linguistic competence of SLMs, LLMs, and their quantized counterparts, and to examine whether linguistic competence influences brain alignment, we benchmark these models on FlashHolmes and analyze brain alignment trends across linguistic tasks. We consider three possible conditions: **Condition1:** When brain alignment is preserved but task-specific

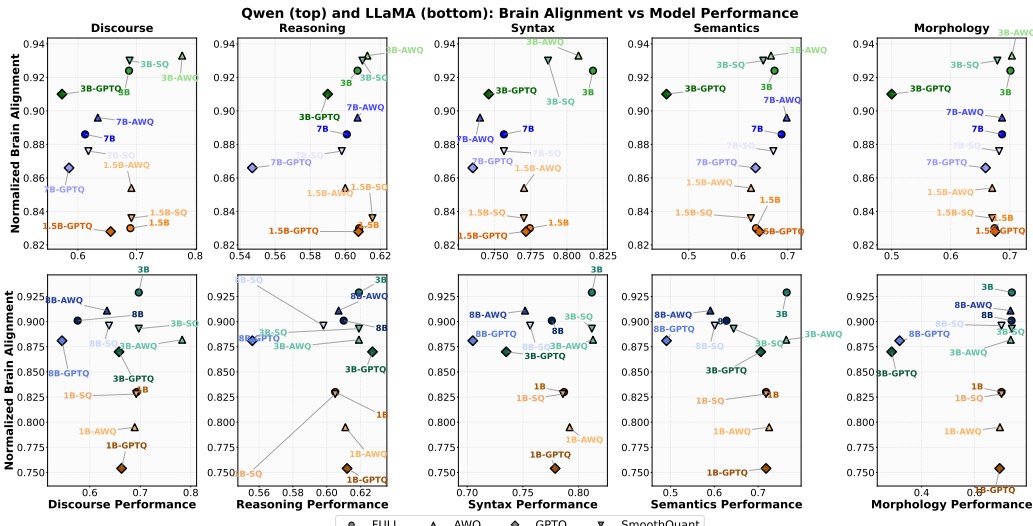

Figure 4: Tradeoff between normalized brain alignment and linguistic competence performance on FlashHolmes Tasks (Qwen and LLaMA Model Families). Blue: 7b/8b, Green: 3b, Orange: 1.5b.

linguistic competence is reduced, the model maintains brain-relevant representational geometry while losing certain linguistic skills, suggesting that compression harms function without disrupting overall alignment. **Condition2:** When task-specific linguistic competence is maintained but brain alignment is reduced, the model preserves its functional linguistic capacity while losing neural-relevant properties, indicating that performance and alignment can diverge as reflections of representational fidelity. **Condition3:** When both brain alignment and linguistic competence increase or decrease together, compression exerts a global effect on representational quality, with joint degradation reflecting broad loss of fidelity and joint improvement pointing to synergistic gains. Overall, this trade-off framework allows us to identify which aspects of linguistic competence are disproportionately affected by compression, even when global brain alignment remains stable.

Figs. 4 illustrate how scaling and quantization affect the relationship between linguistic competence and brain alignment across SLMs, LLMs, and their quantized variants. We make the following observations: (i) For LLMs and 3B SLMs, whole-brain alignment scores remain largely unchanged between full and quantized models, particularly under AWQ and SmoothQuant. However, GPTQ consistently reduces performance in discourse, reasoning, and morphology, while also negatively impacting brain alignment. This suggests that GPTQ preserves overall representational geometry but disproportionately degrades higher-order linguistic skills, especially discourse and reasoning. For 3B SLMs, performance on the FlashHolmes tasks is generally preserved–and in the case of discourse even improved–while the brain score remains comparable to the full model. This suggests that AWQ compression maintains linguistic skills without substantially affecting brain-relevant representations. For 1B SLMs and their quantized variants, performance across the five linguistic tasks is preserved, but brain alignment scores are significantly reduced compared to 3B SLMs. This indicates that 1B SLMs can maintain linguistic skills but fail to retain brain-relevant representations, highlighting a divergence between task performance and neural alignment.

Overall, our analysis reveal a clear trade-off between linguistic competence and brain alignment across SLMs, LLMs, and their quantized variants. AWQ and SmoothQuant largely preserve both measures, GPTQ undermines them–particularly higher-order linguistic competence such as discourse, reasoning, and syntax–and 1B SLMs highlight the limits of scale, maintaining task performance while failing to capture brain-relevant representations. Appendix Tables 3 and 4 provide detailed analyses of probing tasks across five categories of linguistic competence on the FlashHolmes, covering SLMs, LLMs, and their quantized variants (see Appendix G).

## 6 DISCUSSION AND CONCLUSION

In this work, we perform a comprehensive evaluation of LLMs, SLMs, and their quantized variants for brain encoding, using fMRI data collected while participants listened to naturalistic English sto-

ries. We show that 3B SLMs achieve brain encoding performance comparable to LLMs, whereas 1B SLMs consistently exhibit a significant decrease in brain alignment. Our analysis of compression on both SLMs and LLMs reveals that AWQ and SmoothQuant largely preserve brain alignment—with AWQ reducing model size by 60% while GPTQ consistently produces drops in brain alignment, particularly in semantic-selective language regions. The differential impact of quantization techniques provides practical guidance for deployment: AWQ emerges as the most effective choice, preserving $\sim$98% of brain alignment while achieving $\sim$2.3x compression. Overall, our work demonstrates that compressed and lightweight models hold promise for neuroAI applications, but only careful design choices preserve the linguistic representations that drive brain alignment.

## 7  LIMITATIONS

Prior work on scaling laws in fMRI studies has evaluated LLMs up to 72B parameters (Antonello et al., 2024), demonstrating improved brain alignment with increasing scale. However, our study is limited to 7B–8B LLMs, as compressing 13B models yields sizes comparable to uncompressed 7B models, reducing the interprtability value of such comparisons. Future work will extend this analysis to larger LLMs (e.g., 13B) to examine whether 3B SLMs can still maintain both linguistic competence and brain alignment relative to higher-capacity models and their quantized variants. Moreover, we do not systematically manipulate training data diversity or domain; we now highlight this explicitly in the limitations and note that a systematic dissection of training-data effects is an important direction for future work. Second, we focus exclusively on quantization as a compression method, since it is widely used and can be applied post-training. Other techniques, such as pruning or distillation, may have different effects on brain alignment and should be systematically explored. In particular, unstructured pruning can reduce the size of the model while maintaining the memory footprint, making it crucial to directly compare pruning and quantization in terms of both linguistic competence in model tasks and their impact on brain alignment. Third, our analyses are limited to fMRI during naturalistic story listening; future work could incorporate additional modalities (e.g., MEG, ECoG) or task paradigms to better capture the dynamics of language processing. Fourth, the current study focused only on brain encoding, but SLMs, LLMs, and their quantized variants could also be explored in brain decoding. Evaluating these models in decoding tasks would provide practical insights into how well they reconstruct stimuli and whether the generated outputs preserve linguistic competence-offering a complementary way to assess their utility. Addressing these limitations would provide deeper insights into how model scale, compression, and architecture impact brain alignment, particularly in the context of BCI applications.

## 8  REPRODUCIBILITY STATEMENT

Both the naturalistic stimuli (text transcripts corresponding to listening stories) and the fMRI recordings used in this study are publicly available, with pre-processing steps and experimental settings described in Section 3.1 and further detailed in Appendix A. The FlashHolmes benchmark is also publicly available and described in Section 3.4. Details on extracting representations from language models, as well as quantization methods, are provided in Sections 3.2 and 3.3. Implementation of voxel-wise brain encoding models and evaluation metrics is presented in Section 4, with hyperparameters listed in Appendix C. We will release the code upon publication of this paper.

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

# Overview of Appendix Sections

## A  NATURALISTIC LISTENING FMRI DATASET

We use the publicly available naturalistic story listening fMRI dataset provided by (Deniz et al., 2019). The dataset consists of 11 stories, 9 participants, and all participants listened to all the stories. The speech stimuli consisted of 10- to 15 min stories taken from The Moth Radio Hour and used previously (Huth et al., 2016). The 10 selected stories cover a wide range of topics and are highly engaging. The model validation dataset consisted of one 10 min story. All stimuli were played at 44.1 kHz using the pygame library in Python. The audio of each story was down-sampled to 11.5 kHz and the Penn Phonetics Lab Forced Aligner was used to automatically align the audio to the transcript. Finally the aligned transcripts were converted into separate word and phoneme representations using Praat's TextGrid object. The word representation of each story is a list of pairs (W, t), where W is a word and t is the onset time in seconds.

The total number of words in each story as follows: Story1: 2174; Story2: 1469; Story3: 1964; Story4: 1893; Story5: 2209; Story6: 2786; Story7: 3218; Story8: 2675; Story9: 1868; Story10: 1641; Story11: 1839 (test dataset)

To align the stimulus presentation rate with the slower fMRI data acquisition rate (TR = 2.0045 sec), where multiple words correspond to a single TR, we downsample the stimulus features to

match fMRI recording times using a 3-lobed Lanczos filter (Duchon, 1979), thus creating chunk-embeddings for each TR. To account for the slowness of the hemodynamic response (HRF), we model HRF using a finite response filter (FIR) per voxel and for each subject separately with 4 temporal delays corresponding to 8 secs.

## B  CROSS-SUBJECT PREDICTION ACCURACY

We present the average estimated cross-subject prediction accuracy across voxels for the *naturalistic listening fMRI* dataset in Fig. 5. We observe that the average estimated cross-subject prediction accuracy across voxels for the listening dataset is higher across subjects.

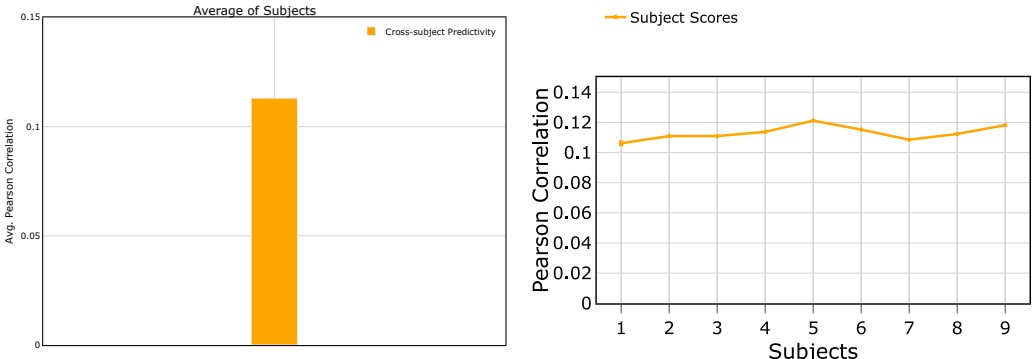

Figure 5: The estimated cross-subject prediction accuracy was computed across all participants for the Subset-Moth-Radio-Hour naturalistic listening fMRI dataset. The average cross-subject prediction accuracy is shown across predicted voxels where each voxel ceiling value is $\geq 0.05$.

Table 2: Detailed functional description of various brain regions.

| | |
|---|---|
| Early auditory | The early auditory region is the earliest cortical region for speech processing. This region is specialized for processing elementary speech sounds, as well as other temporally complex acoustical signals, such as music. |
| Late Language | Late language regions contribute to various linguistic processes. Regions 44 and 45 (Broca's region) are vital for speech production and grammar comprehension (Friederici, 2011). The IFJ, PG, and TPOJ clusters are involved in semantic processing, syntactic interpretation, and discourse comprehension (Deniz et al., 2019; Toneva et al., 2022a). STGa and STS play roles in phonological processing and auditory-linguistic integration (Vaidya et al., 2022; Millet et al., 2022; Gong et al., 2023). TA2 is implicated in auditory processing, especially in the context of language. |

## C  HYPERPARAMETER DETAILS

**Implementation details for reproducibility.** All experiments were conducted on a machine with 2 NVIDIA A100 GPUs with 40GB GPU RAM. We used bootstrap ridge-regression with the following parameters: MSE loss function, and L2-decay ($\lambda$) varied from $10^1$ to $10^3$; best $\lambda$ was chosen by tuning on validation data that comprised a randomly chosen 10% subset from train set used only for hyper-parameter tuning.

## D  STATISTICAL SIGNIFICANCE

To determine if normalized predictivity scores are significantly higher than chance, we use block permutation tests. We employ the standard implementation of a block permutation test for fMRI data, which is to split the fMRI data into blocks of 10 contiguous TRs and permute the order of

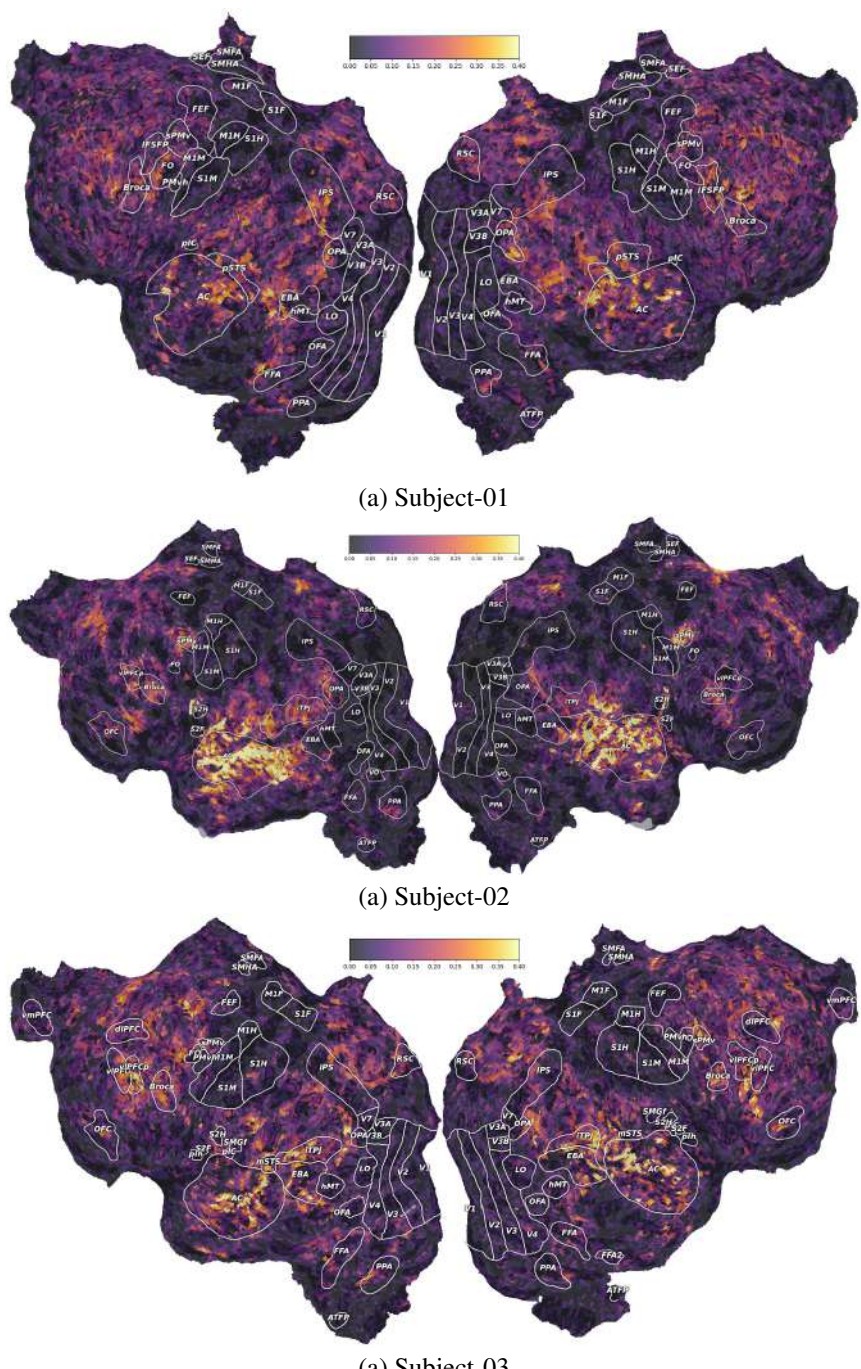

(a) Subject-01

(a) Subject-02

(a) Subject-03

Figure 6: Contrast of estimated cross-subject prediction accuracy for the participants for the listening condition. The color bar denotes Pearson Correlation.

these blocks, while maintaining the original order of the TRs within each block. By permuting predictions 5000 times, we create an empirical distribution for chance performance, from which we estimate the p-value of the actual performance. To estimate the statistical significance of performance differences, such as between the model's predictions and chance or quantized model predictions and chance, we utilized the Wilcoxon signed-rank test, applying it to the mean normalized predictivity for the participants. In all cases, we denote significant differences with an asterisk *, indicating cases where p$\leq$ 0.05.

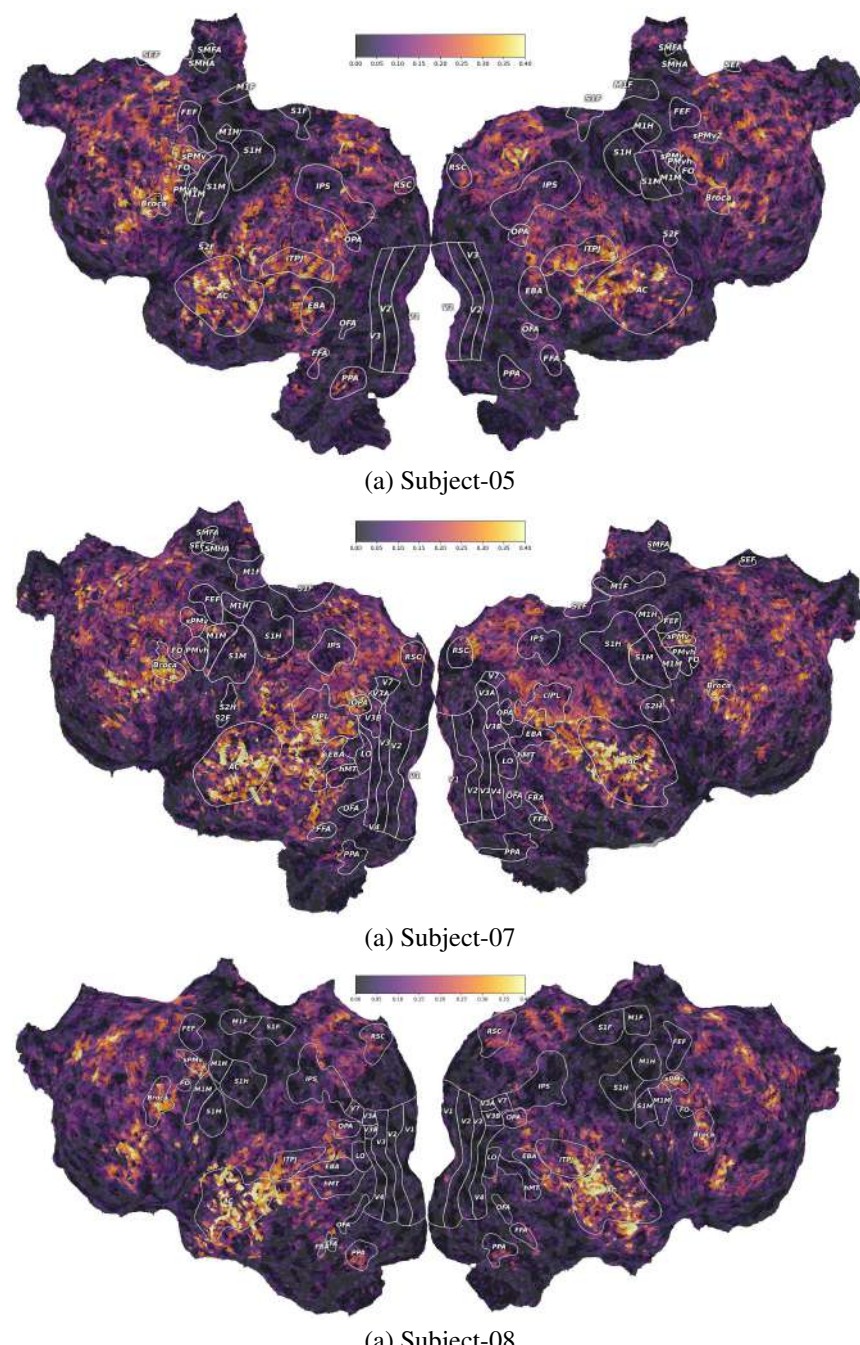

(a) Subject-05

(a) Subject-07

(a) Subject-08

Figure 7: Contrast of estimated cross-subject prediction accuracy for the participants for the listening condition. The color bar denotes Pearson Correlation.

# E   NORMALIZED BRAIN ALIGNMENT FOR LANGUAGE ROIS

Figs. 8 show the average normalized brain alignment across the whole brain for both language model families, comparing SLMs, LLMs, and the grouped quantized variants.

Figs. 9 and 11 present the average normalized brain alignment across language-selective ROIs–including AG, ATL, PTL, IFGOrb, MFG, PCC, dmPFC, and AC–for both language model families, comparing SLMs, LLMs, and their quantized variants.

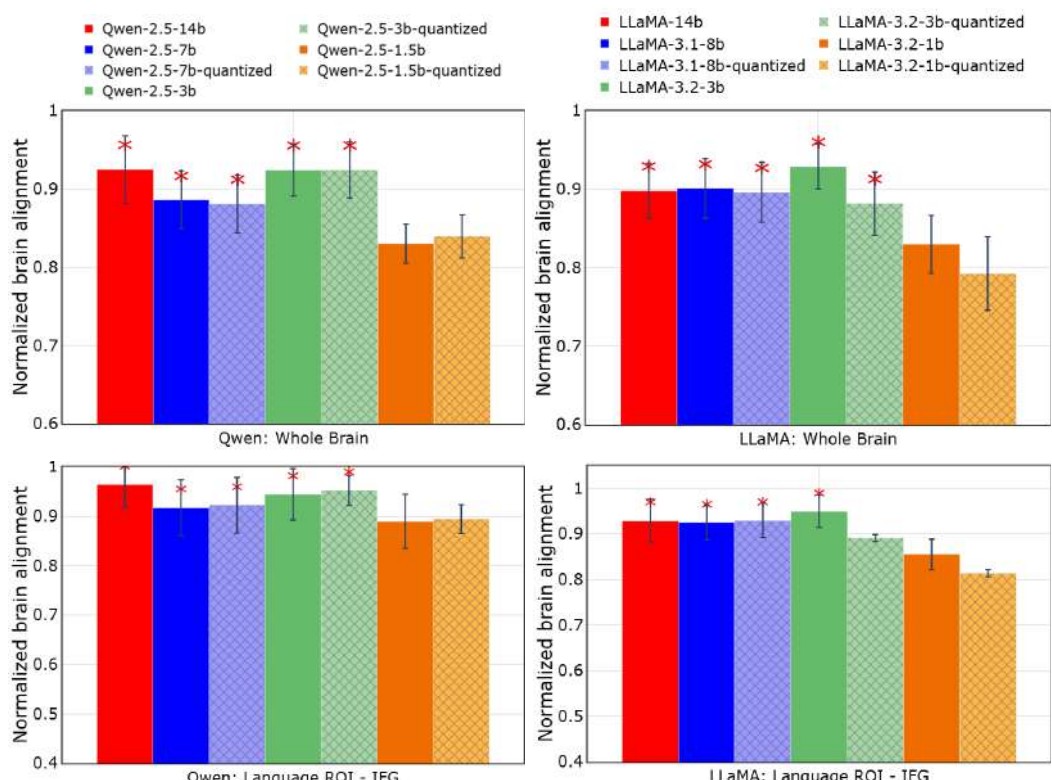

Figure 8: Qwen2.5 and LLaMA: Normalized brain alignment was computed by averaging across participants, layers, and voxels. Red: 14b, Blue: 7b, Green: 3b, Orange: 1.5b, Solid: full-precision SLMs/LLMs, Patterned: quantized models. * at a particular bar indicates that the model's prediction performance is significantly better than 1b/1.5b SLMs. The top row shows whole-brain normalized alignment, while the bottom row focuses on a language-selective ROI (IFG).

## F    CONTRAST OF ESTIMATED MODEL PREDICTION ACCURACY FOR VARIOUS SUBJECTS ACROSS THE TWO MODEL FAMILIES

Figs. 13, 14, 15, 16, and 17 show voxel-wise percentage changes in brain alignment across model scales and quantization strategies for Qwen2.5 in the remaining participants. Corresponding results for LLaMA-3.2 are provided in Figs. 18, 19, 20, 21, 22, and 23.

We make the following observations across two families of language models: (i) Scaling down from LLMs to 3B SLMs: reductions in brain alignment are negligible in the bilateral temporal lobe and remain under 5% in the parietal cortex and IFGorb. Large cortical regions remain white, indicating that 3B SLMs preserve brain-relevant representations comparable to LLMs. The blue-marked voxels are sparse and localized, suggesting only limited information loss. (ii) 3B SLMs to 3B SLMs GPTQ: applying GPTQ leads to widespread orange-marked voxels, reflecting consistent losses across distributed cortical regions. While some areas remain preserved (white voxels), the extent of information loss is greater than that observed with downscaling alone, confirming that GPTQ disproportionately disrupts brain-relevant alignment. (iii) 3B SLMs to 1.5B SLMs: relative to the 3B baseline, we observe extensive red-marked voxels, especially in temporal and language-related regions, indicating larger drops in alignment. This demonstrates the limits of scaling, as ultra-small models fail to capture brain-relevant representations. (iv) 3B SLMs → 3B SLMs AWQ: AWQ produces localized green-marked voxels, indicating regions of improved alignment relative to the 3B baseline, particularly in the IFG and AG. Most of the cortex remains unchanged (white), suggesting that AWQ maintains representational fidelity while offering modest regional gains.

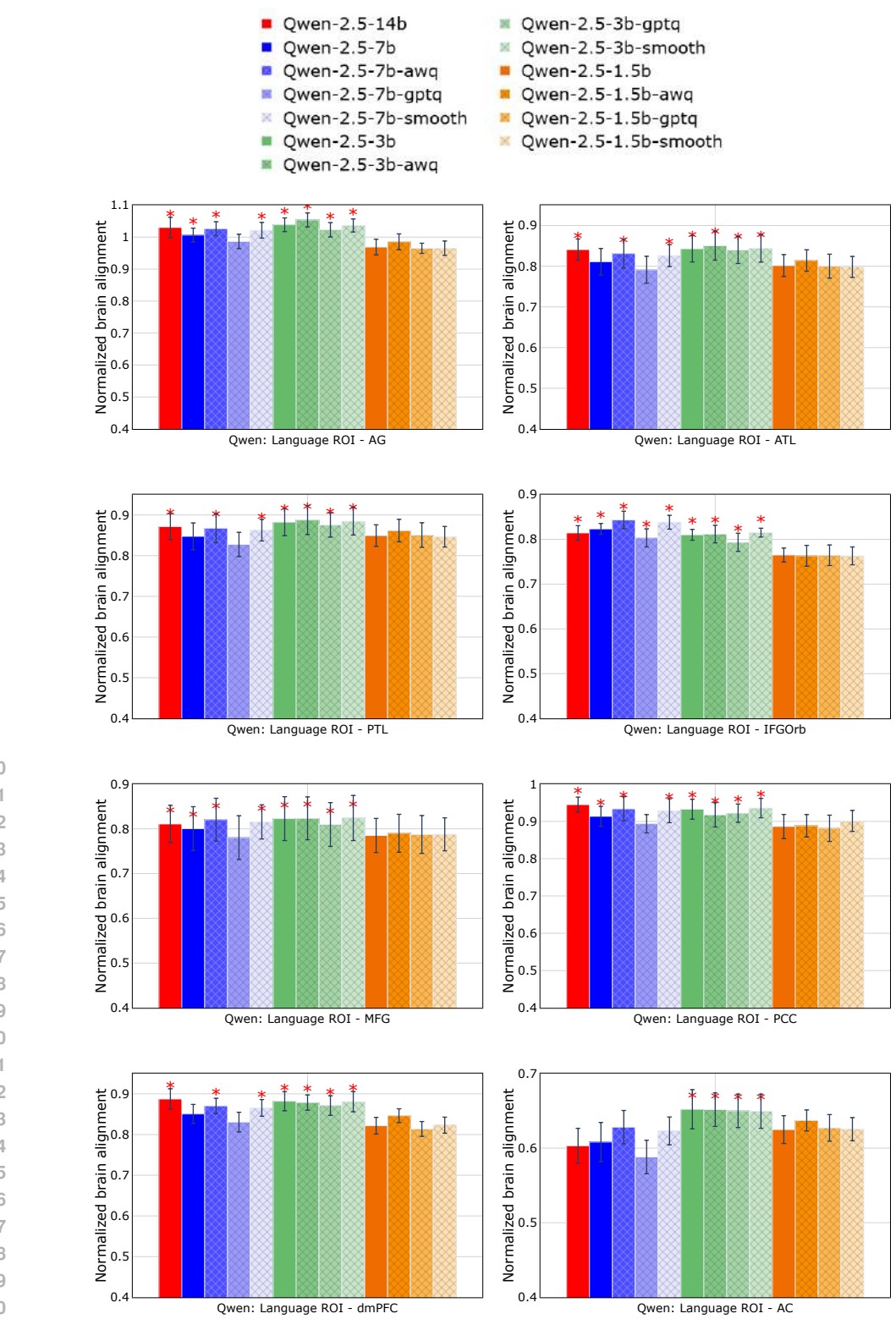

Figure 9: Normalized Predictivity of SLMs, LLMs, and Quantized Language Models for Qwen-2.5 models.

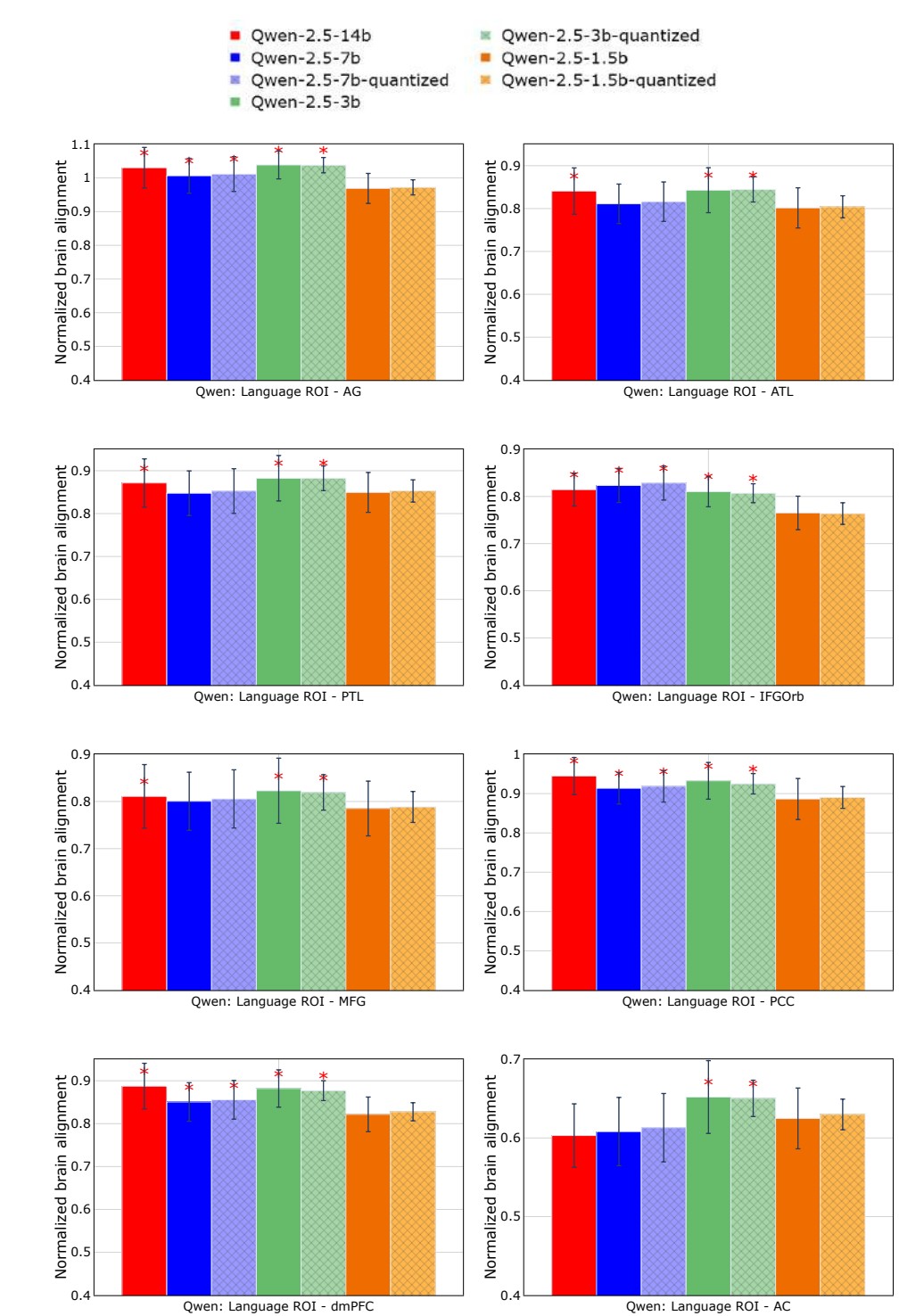

Figure 10: Normalized predictivity of Qwen2.5 SLMs and LLMs, including grouped comparisons of the base and quantized variants.

# G    IMPACT OF LINGUISTIC COMPETENCE FOR QWEN2.5 AND LLAMA-3.2 MODELS

From Table 3, we find that discourse competence emerges as the most influential factor. As models scale from smaller to larger versions, discourse probing scores (e.g., bridging, coreference) increase

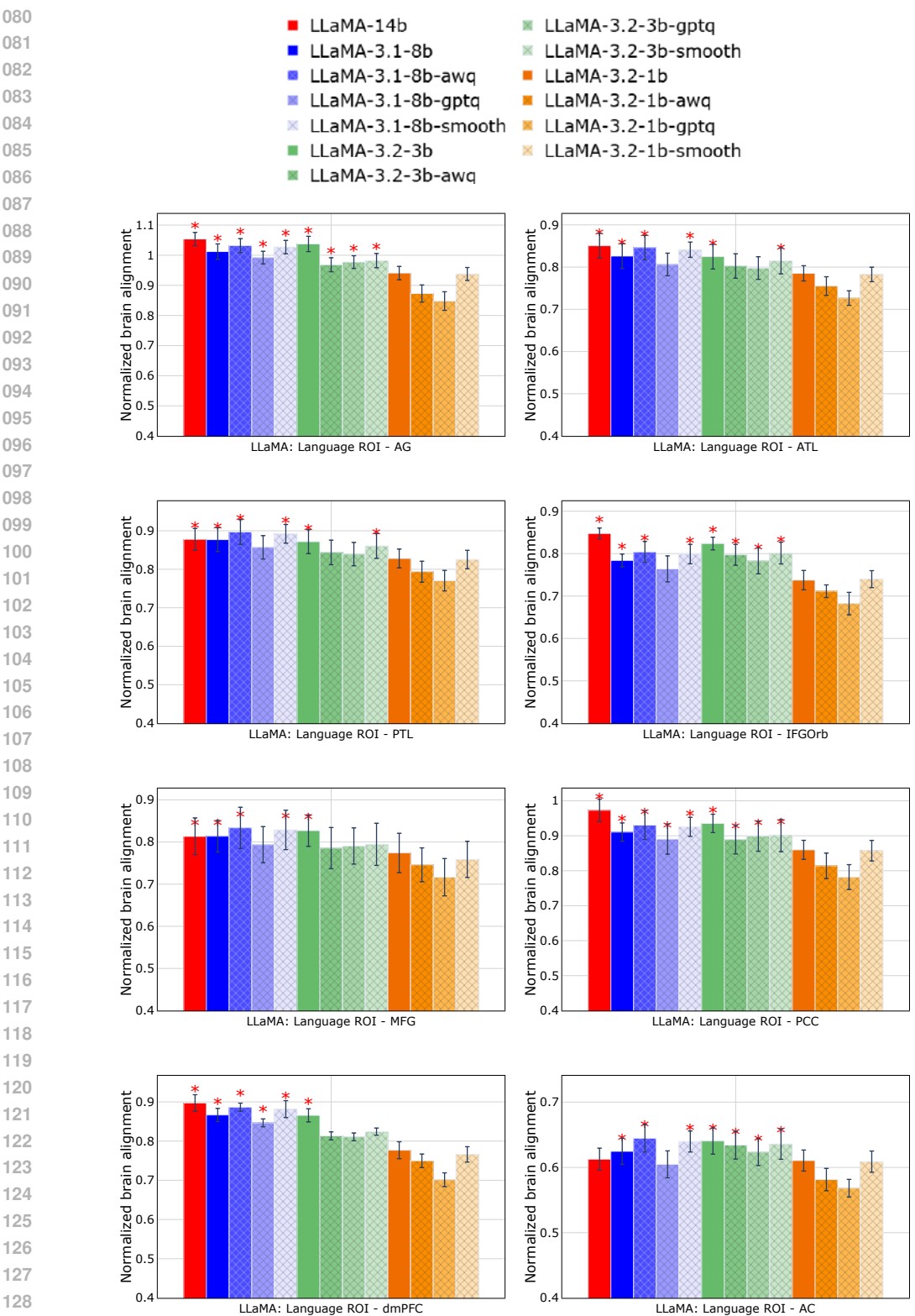

Figure 11: Normalized Predictivity of SLMs, LLMs, and Quantized Language Models for LLaMA-3.2 models.

substantially, and these improvements align closely with gains in brain predictivity. By contrast,

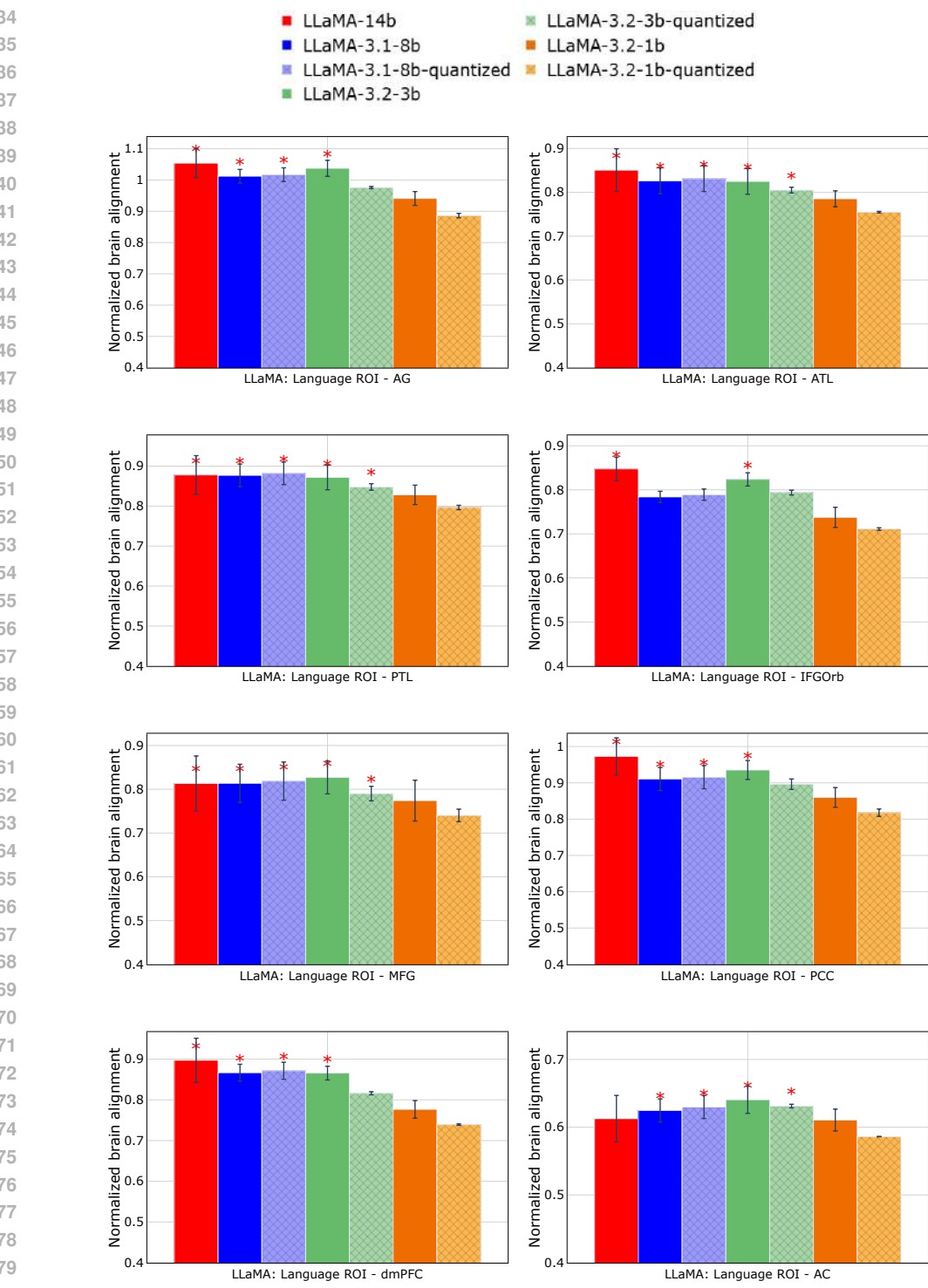

Figure 12: Normalized predictivity of LLaMA3.2 SLMs and LLMs, including grouped comparisons of the base and quantized variants.

morphology tasks exert minimal impact: although accuracy improves modestly with scale, alignment scores remain largely unchanged, suggesting morphology is not a primary driver of brain alignment. Syntax and semantics show moderate effects. Both improve steadily with scale, and their contributions parallel alignment increases, though their influence is consistently weaker than

a) Percentage drop (Qwen2.5: 7b → 3b)    b) Percentage drop (Qwen2.5: 3b → 3b-gptq)

c) Percentage drop (Qwen2.5: 3b → 1.5b)    d) Percentage gain (Qwen2.5: 3b → 3b-awq)

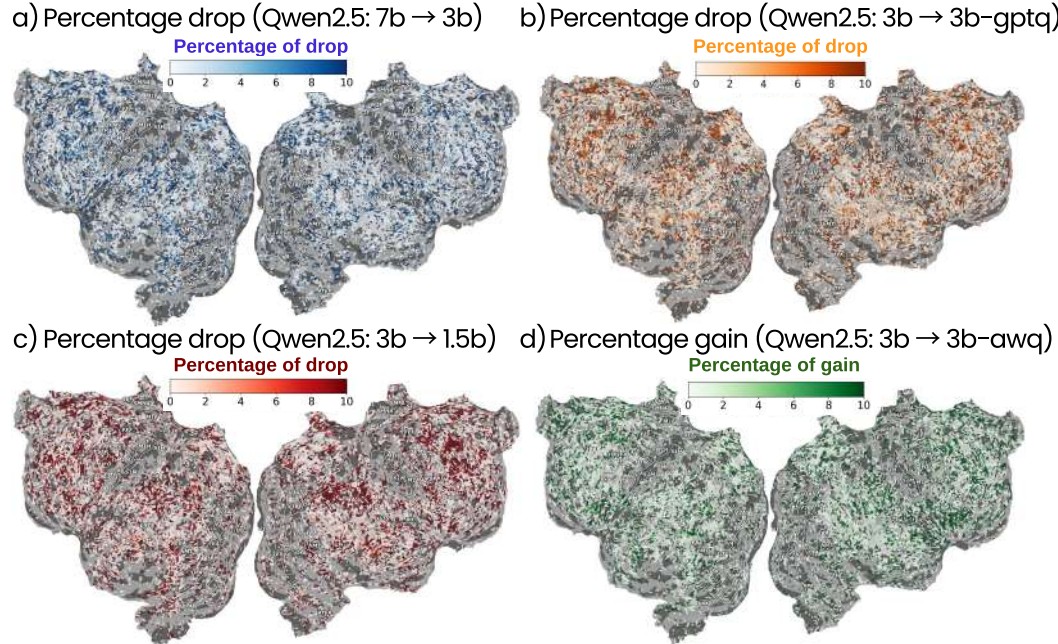

Figure 13: Qwen2.5: Percentage change in brain alignment across model scales and quantization methods, shown on the flattened cortical surface of a representative subject (subject-1). Blue, orange, and red voxels indicate regions of information loss ((a) LLMs → 3B SLMs, (b) 3B SLMs → 3B SLMs GPTQ, (c) 3B SLMs → 1.5B SLMs, respectively), (d) while green voxels highlight regions of improvement for 3B SLMs AWQ over 3B SLMs. White voxels denote regions with no change.

a) Percentage drop (Qwen2.5: 7b → 3b)    b) Percentage drop (Qwen2.5: 3b → 3b-gptq)

c) Percentage drop (Qwen2.5: 3b → 1.5b)    d) Percentage gain (Qwen2.5: 3b → 3b-awq)

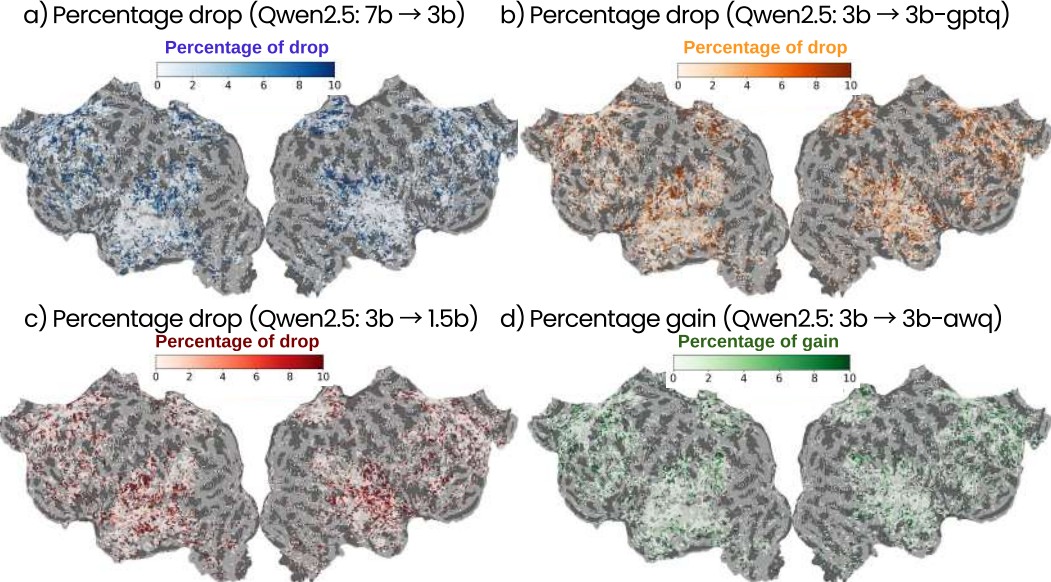

Figure 14: Qwen2.5: Percentage change in brain alignment across model scales and quantization methods, shown on the flattened cortical surface of a representative subject (subject-2). Blue, orange, and red voxels indicate regions of information loss ((a) LLMs → 3B SLMs, (b) 3B SLMs → 3B SLMs GPTQ, (c) 3B SLMs → 1.5B SLMs, respectively), (d) while green voxels highlight regions of improvement for 3B SLMs AWQ over 3B SLMs. White voxels denote regions with no change.

discourse. Reasoning tasks (e.g., negation detection, correspondence) exhibit stable performance across model sizes and quantization settings, with alignment remaining relatively robust. This suggests reasoning is less sensitive to scaling but also less explanatory of alignment gains.

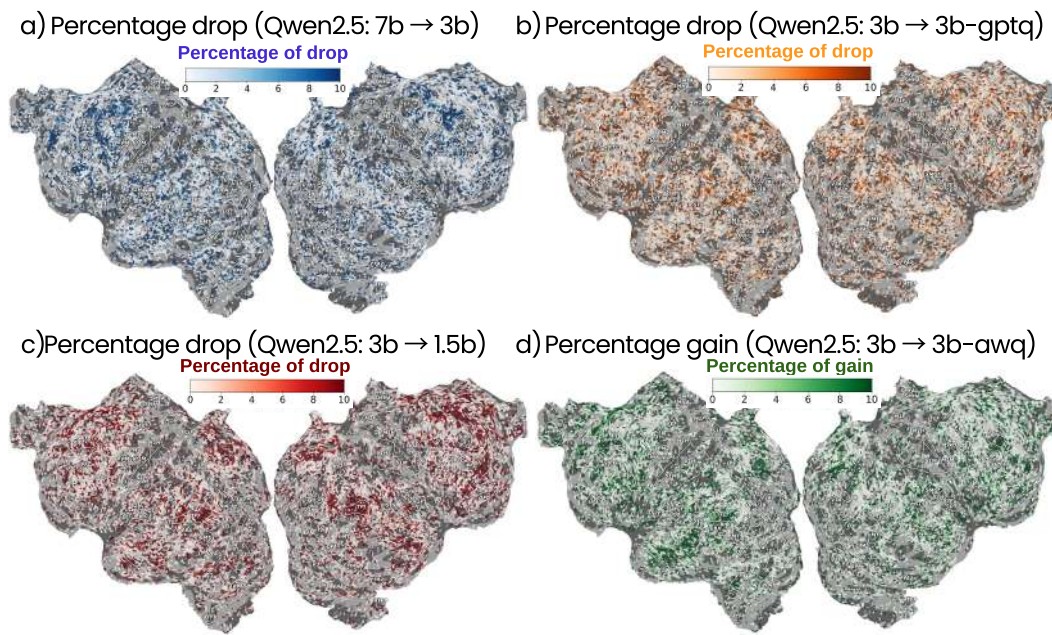

Figure 15: Qwen2.5: Percentage change in brain alignment across model scales and quantization methods, shown on the flattened cortical surface of a representative subject (subject-3). Blue, orange, and red voxels indicate regions of information loss ((a) LLMs → 3B SLMs, (b) 3B SLMs → 3B SLMs GPTQ, (c) 3B SLMs → 1.5B SLMs, respectively), (d) while green voxels highlight regions of improvement for 3B SLMs AWQ over 3B SLMs. White voxels denote regions with no change.

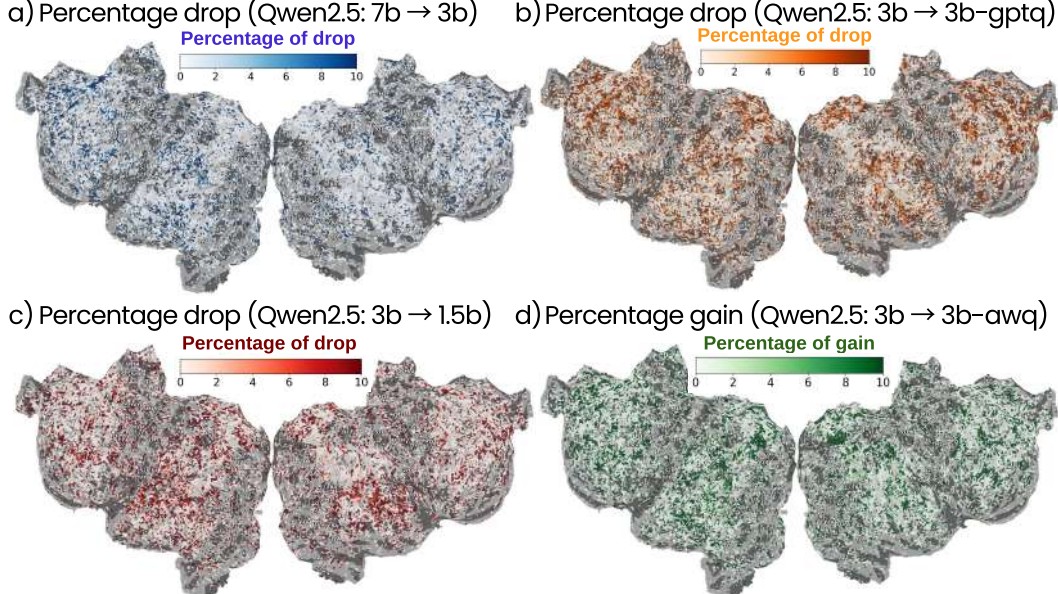

Figure 16: Qwen2.5: Percentage change in brain alignment across model scales and quantization methods, shown on the flattened cortical surface of a representative subject (subject-7). Blue, orange, and red voxels indicate regions of information loss ((a) LLMs → 3B SLMs, (b) 3B SLMs → 3B SLMs GPTQ, (c) 3B SLMs → 1.5B SLMs, respectively), (d) while green voxels highlight regions of improvement for 3B SLMs AWQ over 3B SLMs. White voxels denote regions with no change.

Taken together, these findings indicate that discourse-level representations are most critical for capturing brain-relevant information, followed by syntax and semantics, while morphology contributes

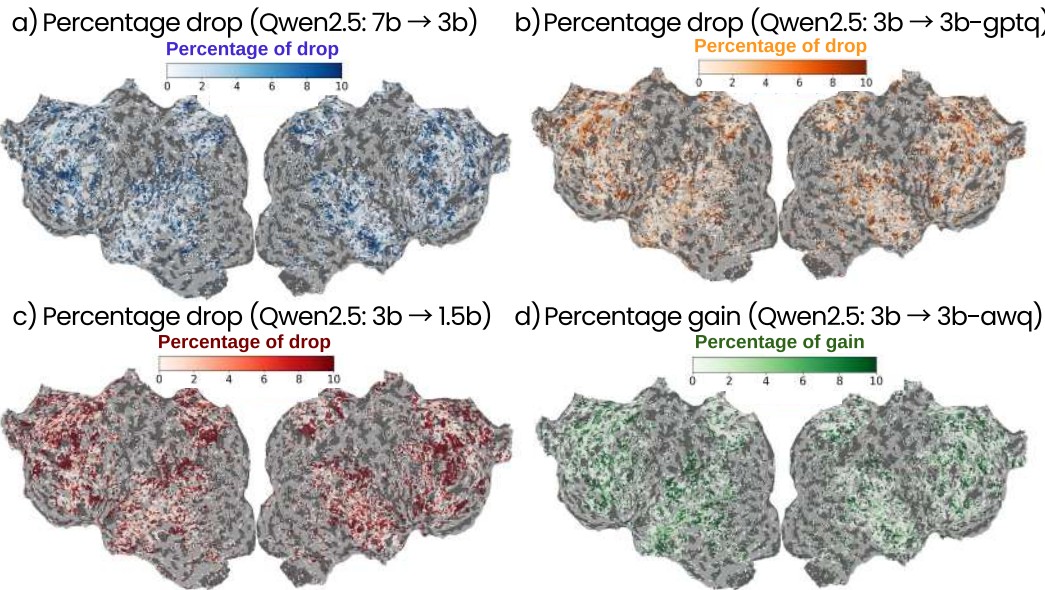

Figure 17: Qwen2.5: Percentage change in brain alignment across model scales and quantization methods, shown on the flattened cortical surface of a representative subject (subject-8). Blue, orange, and red voxels indicate regions of information loss ((a) LLMs → 3B SLMs, (b) 3B SLMs → 3B SLMs GPTQ, (c) 3B SLMs → 1.5B SLMs, respectively), (d) while green voxels highlight regions of improvement for 3B SLMs AWQ over 3B SLMs. White voxels denote regions with no change.

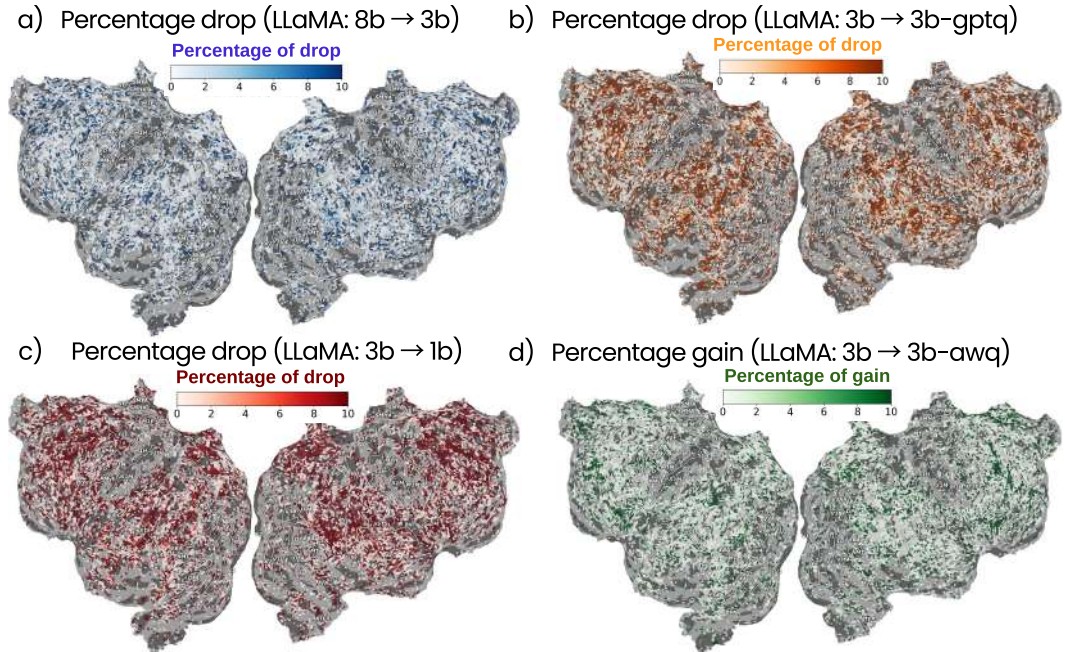

Figure 18: LLaMA-3.2: Percentage change in brain alignment across model scales and quantization methods, shown on the flattened cortical surface of a representative subject (subject-1). Blue, orange, and red voxels indicate regions of information loss ((a) LLMs → 3B SLMs, (b) 3B SLMs → 3B SLMs GPTQ, (c) 3B SLMs → 1.5B SLMs, respectively), (d) while green voxels highlight regions of improvement for 3B SLMs AWQ over 3B SLMs. White voxels denote regions with no change.

little. Reasoning, though robust to compression, does not account for the major alignment differences between small and larger models.

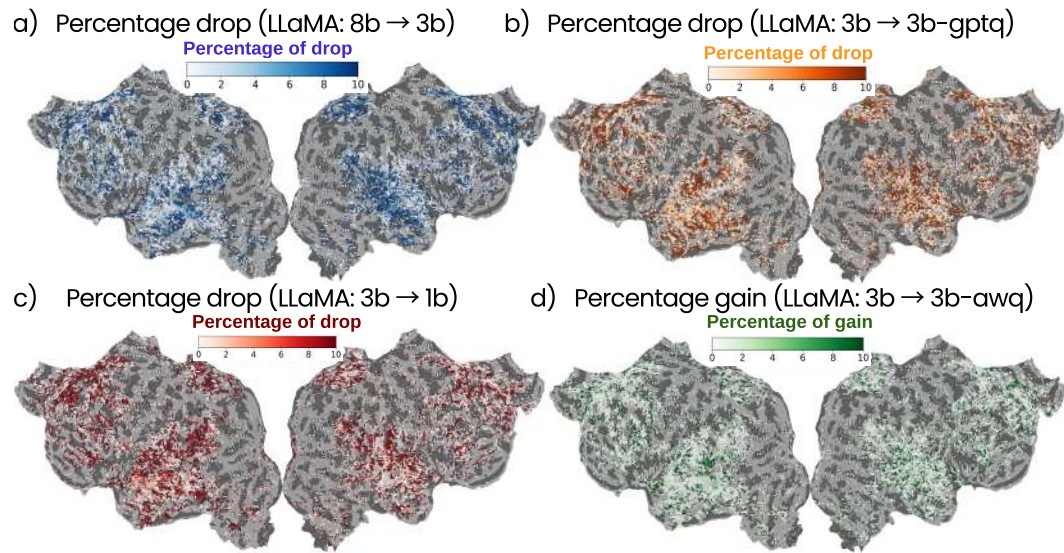

Figure 19: LLaMA-3.2: Percentage change in brain alignment across model scales and quantization methods, shown on the flattened cortical surface of a representative subject (subject-2). Blue, orange, and red voxels indicate regions of information loss ((a) LLMs $\to$ 3B SLMs, (b) 3B SLMs $\to$ 3B SLMs GPTQ, (c) 3B SLMs $\to$ 1.5B SLMs, respectively), (d) while green voxels highlight regions of improvement for 3B SLMs AWQ over 3B SLMs. White voxels denote regions with no change.

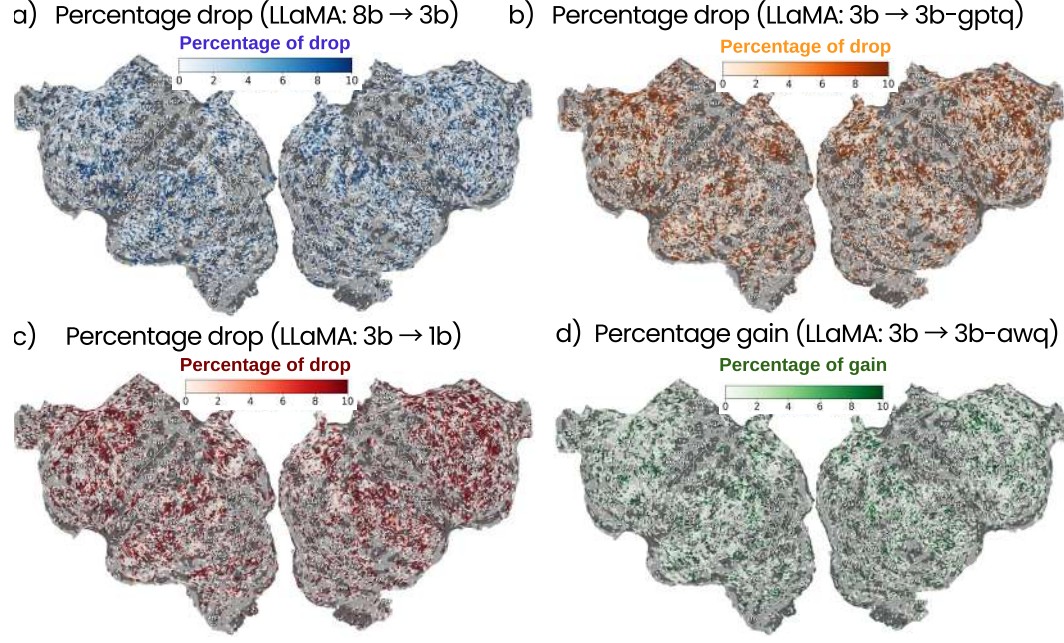

Figure 20: LLaMA-3.2: Percentage change in brain alignment across model scales and quantization methods, shown on the flattened cortical surface of a representative subject (subject-3). Blue, orange, and red voxels indicate regions of information loss ((a) LLMs $\to$ 3B SLMs, (b) 3B SLMs $\to$ 3B SLMs GPTQ, (c) 3B SLMs $\to$ 1.5B SLMs, respectively), (d) while green voxels highlight regions of improvement for 3B SLMs AWQ over 3B SLMs. White voxels denote regions with no change.

## H    LLM USAGE

We used OpenAI ChatGPT for grammar correction and language polishing.

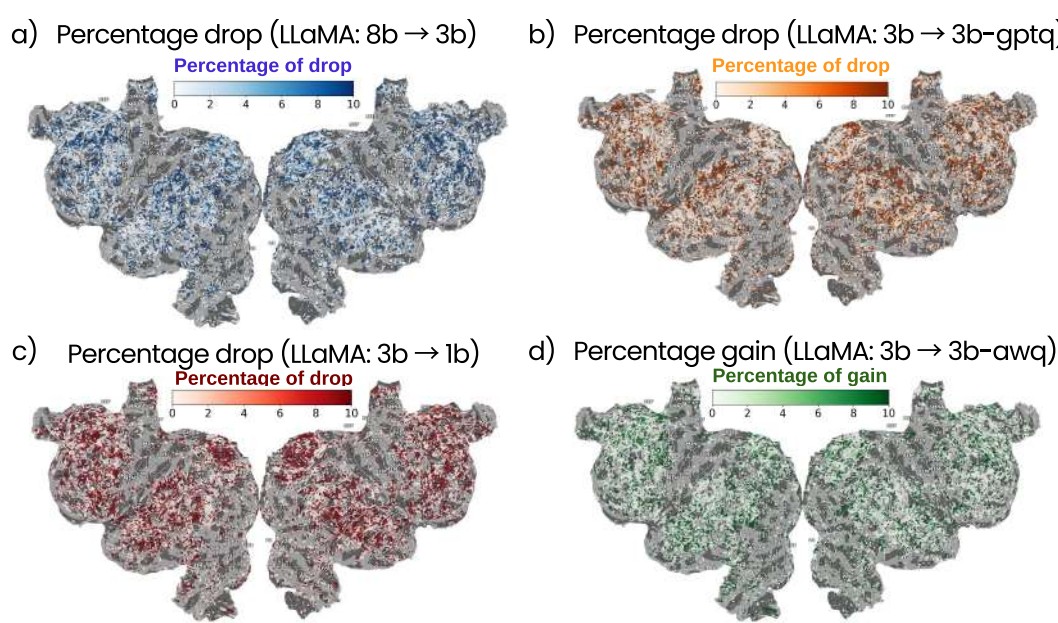

Figure 21: LLaMA-3.2: Percentage change in brain alignment across model scales and quantization methods, shown on the flattened cortical surface of a representative subject (subject-7). Blue, orange, and red voxels indicate regions of information loss ((a) LLMs → 3B SLMs, (b) 3B SLMs → 3B SLMs GPTQ, (c) 3B SLMs → 1.5B SLMs, respectively), (d) while green voxels highlight regions of improvement for 3B SLMs AWQ over 3B SLMs. White voxels denote regions with no change.

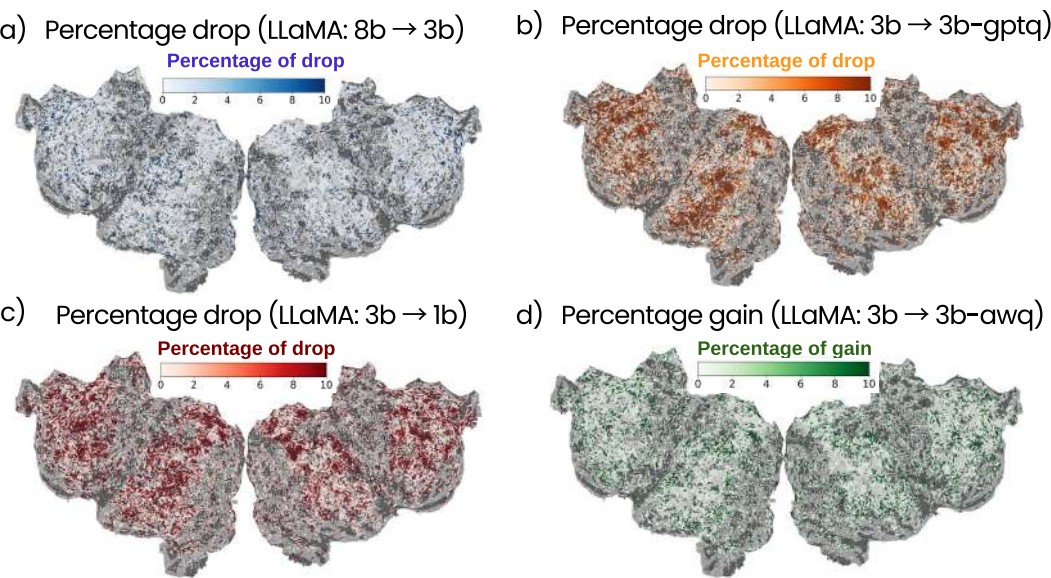

Figure 22: LLaMA-3.2: Percentage change in brain alignment across model scales and quantization methods, shown on the flattened cortical surface of a representative subject (subject-8). Blue, orange, and red voxels indicate regions of information loss ((a) LLMs → 3B SLMs, (b) 3B SLMs → 3B SLMs GPTQ, (c) 3B SLMs → 1.5B SLMs, respectively), (d) while green voxels highlight regions of improvement for 3B SLMs AWQ over 3B SLMs. White voxels denote regions with no change.

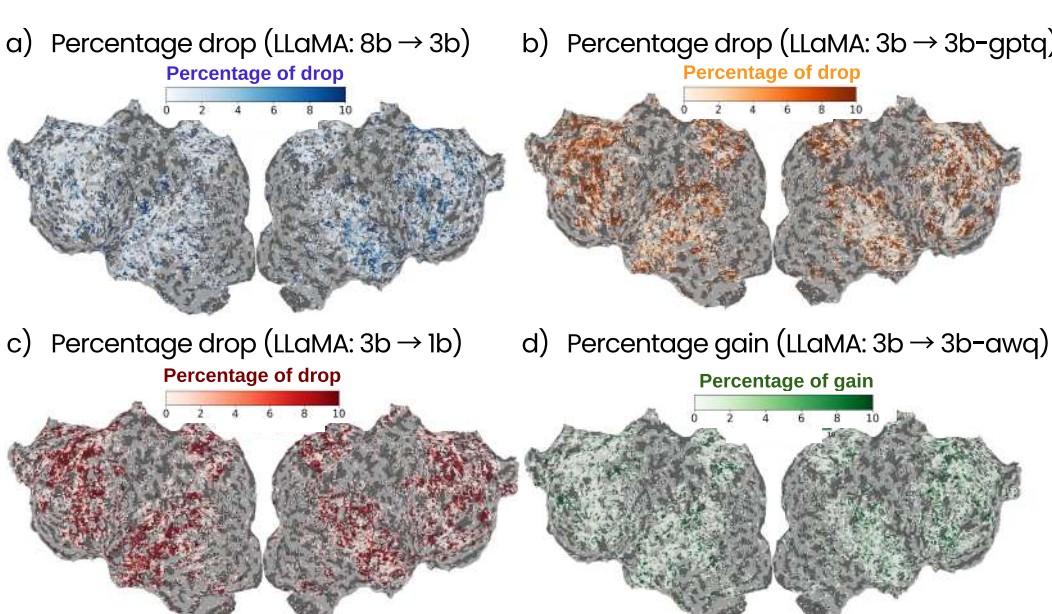

a) Percentage drop (LLaMA: 8b → 3b)

b) Percentage drop (LLaMA: 3b → 3b-gptq)

c) Percentage drop (LLaMA: 3b → 1b)

d) Percentage gain (LLaMA: 3b → 3b-awq)

Figure 23: LLaMA-3.2: Percentage change in brain alignment across model scales and quantization methods, shown on the flattened cortical surface of a representative subject (subject-8). Blue, orange, and red voxels indicate regions of information loss ((a) LLMs → 3B SLMs, (b) 3B SLMs → 3B SLMs GPTQ, (c) 3B SLMs → 1.5B SLMs, respectively), (d) while green voxels highlight regions of improvement for 3B SLMs AWQ over 3B SLMs. White voxels denote regions with no change.

Table 3: Representative FlashHolmes task scores for Qwen-2.5 (1.5B, 3B, and 7B). Quantized 3B models remain close to full 3B across tasks, while 1.5B models show larger drops, especially in discourse and reasoning. The 7B model achieves the strongest scores overall.

| Category | Task | Qwen-2.5 1.5B | | | | Qwen-2.5 3B | | | | Qwen-2.5 7B |
|---|---|---|---|---|---|---|---|---|---|---|
| | | FULL | AWQ | GPTQ | Smooth | FULL | AWQ | GPTQ | Smooth | FULL |
| Discourse | Bridging (edge) | 0.789 | **0.799** | **0.798** | **0.799** | **0.798** | 0.794 | 0.433 | 0.794 | 0.683 |
| | Bridging (sentence) | **0.800** | **0.800** | **0.800** | **0.800** | **0.800** | 0.799 | 0.410 | **0.799** | 0.667 |
| | Coreference | 0.411 | 0.399 | 0.405 | 0.399 | 0.352 | 0.433 | 0.410 | 0.357 | **0.794** |
| Morphology | Constituent (depth) | 0.802 | 0.801 | **0.809** | 0.801 | 0.752 | 0.752 | 0.801 | 0.752 | 0.755 |
| | Constituent (length) | 0.827 | 0.826 | **0.847** | 0.826 | 0.796 | 0.795 | 0.826 | 0.795 | 0.725 |
| Reasoning | Negation span classify | 0.742 | 0.741 | 0.746 | 0.743 | 0.812 | 0.810 | 0.808 | 0.811 | 0.950 |
| | Negation correspondence | 0.605 | 0.602 | 0.609 | 0.603 | 0.689 | 0.687 | 0.685 | 0.688 | 0.611 |
| | SemAntoNeg | 0.667 | 0.666 | 0.669 | 0.667 | 0.701 | 0.699 | 0.698 | 0.700 | 0.667 |
| Semantics | Object animacy | 0.994 | 0.994 | 0.991 | 0.994 | 0.988 | 0.988 | 0.994 | 0.988 | 0.805 |
| | Object gender | 0.546 | 0.535 | 0.532 | 0.535 | 0.482 | 0.402 | 0.531 | 0.402 | 0.807 |
| | Object number | 0.744 | 0.738 | 0.738 | 0.738 | 0.716 | 0.712 | 0.738 | 0.712 | 0.781 |
| Syntax | Adjunct island | 0.704 | 0.643 | 0.689 | 0.643 | 0.678 | 0.677 | 0.643 | 0.676 | 0.515 |
| | Anaphor gender agr. | 0.609 | 0.548 | 0.573 | 0.548 | 0.622 | 0.619 | 0.548 | 0.619 | 0.695 |
| | Anaphor number agr. | 0.631 | 0.611 | 0.582 | 0.613 | 0.647 | 0.661 | 0.613 | 0.662 | 0.860 |

Table 4: Representative FlashHolmes task scores for LLaMA-3.2 models (1B, 3B, and 8B). Quantized 3B models remain close to full 3B, while the 8B model achieves the strongest scores overall.

| Category | Task | LLaMA-3.2 1B | | | | LLaMA-3.2 3B | | | | LLaMA-3.1 8B |
|---|---|---|---|---|---|---|---|---|---|---|
| | | FULL | AWQ | GPTQ | Smooth | FULL | AWQ | GPTQ | Smooth | FULL |
| Discourse | Bridging (edge) | 0.792 | 0.801 | 0.800 | 0.799 | 0.801 | 0.799 | 0.433 | 0.798 | 0.703 |
| | Bridging (sentence) | 0.800 | 0.800 | 0.800 | 0.800 | 0.800 | 0.799 | 0.410 | 0.799 | 0.667 |
| | Coreference | 0.392 | 0.401 | 0.405 | 0.400 | 0.417 | 0.433 | 0.410 | 0.411 | 0.776 |
| Morphology | Constituent (depth) | 0.838 | 0.830 | 0.841 | 0.831 | 0.842 | 0.839 | 0.845 | 0.840 | 0.735 |
| | Constituent (length) | 0.877 | 0.875 | 0.890 | 0.876 | 0.897 | 0.896 | 0.895 | 0.894 | 0.705 |
| Reasoning | Negation span classify | 0.745 | 0.743 | 0.747 | 0.744 | 0.818 | 0.816 | 0.814 | 0.817 | 0.944 |
| | Negation correspondence | 0.612 | 0.610 | 0.614 | 0.611 | 0.701 | 0.699 | 0.697 | 0.700 | 0.611 |
| | SemAntoNeg | 0.672 | 0.670 | 0.673 | 0.671 | 0.709 | 0.707 | 0.706 | 0.708 | 0.667 |
| Semantics | Object animacy | 0.981 | 0.981 | 0.980 | 0.981 | 0.989 | 0.989 | 0.988 | 0.989 | 0.882 |
| | Object gender | 0.626 | 0.623 | 0.628 | 0.624 | 0.644 | 0.640 | 0.639 | 0.642 | 0.879 |
| | Object number | 0.713 | 0.710 | 0.714 | 0.711 | 0.720 | 0.718 | 0.717 | 0.719 | 0.876 |
| Syntax | Adjunct island | 0.668 | 0.667 | 0.669 | 0.667 | 0.744 | 0.742 | 0.741 | 0.743 | 0.700 |
| | Anaphor gender agr. | 0.678 | 0.675 | 0.680 | 0.676 | 0.739 | 0.738 | 0.736 | 0.737 | 0.650 |
| | Anaphor number agr. | 0.660 | 0.659 | 0.662 | 0.660 | 0.671 | 0.670 | 0.669 | 0.671 | 0.845 |

# I QUANTITATIVE ANALYSIS ACROSS MODEL FAMILIES

We quantify scaling differences by performing statistical significance across subjects for the best selective layer per model: Qwen2.5 in Table 5, LLaMA in Table 6 and DeepSeek 7. For Qwen2.5 model, the resulting mean best-layer scores are: 1.5B: 0.85+0.09, 3B: 0.92+0.08, 7B: 0.895+0.09, 14B: 0.93+0.10. Paired tests over subjects (n = 9) show that 3B and 14B are statistically indistinguishable (mean difference -0.0004, t(8) = -0.03, p ≈ 0.98), while both 3B and 14B significantly outperform the 1.5B model (3B vs. 1.5B: $\Delta$ = 0.07, t(8) = 4.89, p ≈ 0.004; 14B vs. 1.5B: $\Delta$ = 0.07, t(8) = 3.16, p ≈ 0.025). We also find a modest but significant advantage of 3B and 14B over 7B in best-layer alignment ($\Delta$ ≈ 0.04, p ≈ 0.02-0.04). These tests support our main claim in this regime: beyond 3B, increasing model size up to 14B yields at most modest gains in brain alignment, whereas 1B-1.5B models are reliably worse.

For the LLaMA model, as shown in Table 6, we find that (i) 14B ≈ 7B: No difference ($\Delta$ = -0.00, p ≈ 0.95) - statistically identical (ii) 3B ≈ 14B/7B: Slight advantage but not significant (p > 0.05) and (iii) All vs 1B: Highly significant differences (p < 0.001 for 3B and 7B). Overall, the 3B, 7B, and 14B models form a statistically equivalent top tier, all significantly outperforming the 1B model.

Analysis of DeepSeek models (14B, 7B, 3B, 1B parameters), as shown in Table 7 reveals a clear scaling hierarchy with the 14B and 3B models forming a statistically equivalent top tier. We make the following observations from Table 7: (i) 14B ≈ 3B: No significant difference (p ≈ 0.61), indicating 3B achieves 14B-level performance with 80% fewer parameters! (ii) 14B, 3B >> 7B: Highly significant advantages (p < 0.01), (iii) All >> 1B: Very large differences (all p < 0.001). Overall, across three independent model families (Qwen, LLaMA, DeepSeek) and using best-layer scores with paired tests over nine subjects, we find a consistent pattern: 1B-1.5B models are reliably worse in brain alignment, while 3B models already reach the same level as their 7B-14B counterparts. In Qwen and DeepSeek, 3B and 14B are statistically indistinguishable, whereas both significantly outperform the smallest models; in LLaMA, 3B and 14B again lie in a narrow, non-significantly different range, with 7B closely tracking 14B and clearly above 1B. These results do not overturn global scaling laws, but they do indicate a local plateau in the compressed 1-14B regime: once model capacity reaches ∼3B, further scaling yields at most modest gains in brain predictivity, while going below this threshold leads to a robust drop in alignment.

Table 5: Pairwise differences in Qwen best-layer encoding scores across 9 subjects. For each subject and model, we take the maximum encoding score across evaluated layers and then compute paired t-tests between models. $\Delta$ is the mean difference A–B over subjects.

| Comparison (A–B) | $\Delta$ | $t(8)$ | $p$ (two-sided, approx.) | Interpretation |
|---|---|---|---|---|
| 3B – 14B | 0.000 | -0.00 | 1.00 | 3B ≈ 14B (no difference) |
| 3B – 7B | 0.028 | 2.43 | 0.06 | 3B > 7B (small effect, trend) |
| 3B – 1.5B | 0.073 | 4.89 | 0.004 | 3B > 1.5B (clear, significant) |
| 14B – 7B | 0.028 | 2.02 | 0.10 | 14B > 7B (small effect, n.s.) |
| 14B – 1.5B | 0.073 | 3.16 | 0.025 | 14B > 1.5B (clear, significant) |
| 7B – 1.5B | 0.045 | 2.67 | 0.045 | 7B > 1.5B (moderate, significant) |

Table 6: Pairwise differences in LLaMA best-layer scores across models (paired $t$-tests over 9 subjects). $\Delta$ is mean(A–B).

| Comparison | $\Delta$ (A–B) | $t(8)$ | Sig. (2-sided) |
|---|---|---|---|
| 3B – 14B | 0.03 | 2.50 | n.s. ($p ≈ 0.05$) |
| 3B – 7B | 0.03 | 1.97 | n.s. ($p ≈ 0.11$) |
| 3B – 1B | 0.10 | 6.89 | $p < 0.001$ |
| 14B – 7B | -0.00 | -0.07 | n.s. ($p ≈ 0.95$) |
| 14B – 1B | 0.07 | 3.12 | $p < 0.05$ |
| 7B – 1B | 0.07 | 11.43 | $p < 0.001$ |

Table 7: Pairwise differences in DeepSeek best-layer scores (paired $t$-tests over 9 subjects). $\Delta$ is mean(A–B).

| Comparison | $\Delta$ (A–B) | $t(8)$ | Sig. (2-sided) |
|---|---|---|---|
| 14B – 3B | -0.01 | -0.54 | n.s. ($p ≈ 0.61$) |
| 14B – 7B | 0.07 | 4.84 | $p < 0.01$ |
| 14B – 1B | 0.19 | 11.18 | $p < 0.001$ |
| 3B – 7B | 0.08 | 4.18 | $p < 0.01$ |
| 3B – 1B | 0.19 | 10.85 | $p < 0.001$ |
| 7B – 1B | 0.11 | 9.28 | $p < 0.001$ |

## J  STATISTICAL VALIDATION OF QUANTIZATION EFFECTS

**Quantization Effects - Qwen2.5.** We now provide formal statistical tests and variability measures for the quantization comparisons. For each Qwen2.5 model (1.5B, 3B, 7B), and for each quantization method (Full (FP16), AWQ, GPTQ, SmoothQuant), we compute best-layer brain alignment per subject and run paired $t$-tests across subjects between methods (Table 8). Negative $\Delta$ in rows of the form "FP16–AWQ" indicates that AWQ outperforms FP16. For Qwen2.5–7B (Table 8, left), AWQ and SmoothQuant are significantly better than both FP16 and GPTQ (e.g., FP16–AWQ: $\Delta = -0.020$, $t(8) = -6.10$, $p < 0.001$; AWQ–GPTQ: $\Delta = 0.040$, $t(8) = 7.10$, $p < 0.001$), while GPTQ is significantly worse than FP16. For Qwen2.5–3B (Table 8, right), none of the quantized variants differ significantly from FP16, but AWQ and SmoothQuant significantly outperform GPTQ, suggesting that well-designed quantization preserves alignment whereas GPTQ exhibits a modest degradation. For Qwen2.5–1.5B (Table 8, bottom), AWQ is significantly better than FP16 ($\Delta = -0.024$, $t(8) = -4.04$, $p < 0.01$), whereas GPTQ and SmoothQuant are statistically indistinguishable from FP16, and differences among the three quantized variants do not reach significance after correction.

Table 8: Pairwise comparisons of brain-alignment differences across quantization methods for Qwen2.5 models. Each Table reports mean differences ($\Delta$), $t$-statistics, and two-sided significance tests for 7B (left), 3B (right), and 1.5B (bottom).

| Comparison (A–B) | $\Delta$ | $t(8)$ | Sig. |
|---|---|---|---|
| Qwen2.5-7B–AWQ | -0.020 | -6.10 | $p < 0.001$ |
| Qwen2.5-7B–GPTQ | 0.020 | 6.20 | $p < 0.001$ |
| Qwen2.5-7B–SmoothQuant | -0.005 | -3.50 | $p < 0.016$ |
| AWQ–GPTQ | 0.040 | 7.10 | $p < 0.001$ |
| AWQ–SmoothQuant | 0.015 | 4.20 | $p < 0.008$ |
| GPTQ–SmoothQuant | -0.025 | -4.90 | $p < 0.004$ |

| Comparison (A–B) | $\Delta$ | $t(8)$ | Sig. |
|---|---|---|---|
| Qwen2.5-3B–AWQ | -0.010 | -1.23 | n.s. ($p \approx 0.28$) |
| Qwen2.5-3B–GPTQ | 0.014 | 2.24 | n.s. ($p \approx 0.08$) |
| Qwen2.5-3B–SmoothQuant | -0.007 | -1.55 | n.s. ($p \approx 0.18$) |
| AWQ–GPTQ | 0.024 | 3.10 | $p < 0.05$ |
| AWQ–SmoothQuant | 0.003 | 0.34 | n.s. ($p \approx 0.75$) |
| GPTQ–SmoothQuant | -0.020 | -3.12 | $p < 0.05$ |

| Comparison (A–B) | $\Delta$ | $t(8)$ | Sig. |
|---|---|---|---|
| Qwen2.5-1.5B–AWQ | -0.024 | -4.04 | $p < 0.01$ |
| Qwen2.5-1.5B–GPTQ | 0.002 | 0.18 | n.s. ($p \approx 0.86$) |
| Qwen2.5-1.5B–SmoothQuant | -0.004 | -0.71 | n.s. ($p \approx 0.51$) |
| AWQ–GPTQ | 0.026 | 2.03 | n.s. ($p \approx 0.10$) |
| AWQ–SmoothQuant | 0.020 | 2.43 | n.s. ($p \approx 0.06$) |
| GPTQ–SmoothQuant | -0.006 | -0.42 | n.s. ($p \approx 0.69$) |

We also summarize quantization performance at the level of mean $\pm$ standard deviation across subjects in Table 9. Across all Qwen sizes, AWQ and SmoothQuant closely or slightly exceed full models (FP16) in mean best-layer alignment (differences on the order of 0.01–0.02, within one standard deviation), whereas GPTQ tends to be lower than FP16, especially for 7B and 3B. Together, Table 8 and Table 9 show that (i) some apparent improvements in the figures are within noise and are now explicitly reported as non-significant, and (ii) the main qualitative pattern is statistically supported: well-designed quantization (AWQ/SmoothQuant) preserves brain alignment at near-full-precision levels, while GPTQ produce a modest but reliable degradation.

Table 9: Quantization method performance across Qwen models (mean $\pm$ std over 9 subjects).

| Model | Full precision (FP16) | AWQ | GPTQ | SmoothQuant |
|---|---|---|---|---|
| Qwen-7B | $0.886 \pm 0.092$ | $0.906 \pm 0.092$ | $0.866 \pm 0.092$ | $0.891 \pm 0.092$ |
| Qwen-3B | $0.923 \pm 0.080$ | $0.933 \pm 0.085$ | $0.910 \pm 0.091$ | $0.930 \pm 0.085$ |
| Qwen-1.5B | $0.850 \pm 0.087$ | $0.874 \pm 0.099$ | $0.848 \pm 0.088$ | $0.854 \pm 0.084$ |

**Quantization effects in LLaMA-3.** We performed the same best-layer, paired $t$-test analysis for LLaMA-3 models (1B, 3B, 8B). For LLaMA-3-8B, all pairwise differences between FP16, AWQ, GPTQ, and SmoothQuant are highly significant (Table 10, left). Negative $\Delta$ in rows of the form "FP16–AWQ" indicates AWQ > FP16; specifically, FP16–AWQ is negative ($\Delta$ = -0.010, $p < 0.001$), while FP16–GPTQ is positive ($\Delta = 0.020$, $p < 0.001$) and AWQ-GPTQ is strongly positive ($\Delta = 0.030$, $p < 0.001$). This implies the ordering AWQ > FP16 $ge$ SmoothQuant > GPTQ for 8B. For LLaMA-3-3B (Table 10, right), GPTQ is significantly worse than FP16 ($\Delta$ = 0.059, $p < 0.05$), while AWQ and SmoothQuant are not significantly different from FP16. SmoothQuant significantly outperforms both AWQ and GPTQ (AWQ–SmoothQuant: $\Delta$ = -0.011, $p < 0.05$; GPTQ–SmoothQuant: $\Delta$ = -0.023, $p < 0.01$), indicating that GPTQ is the main outlier, with AWQ/SmoothQuant and FP16 forming a higher-performing cluster. For LLaMA-3-1B (Table 10, bottom), GPTQ again shows a clear degradation: FP16–GPTQ is positive and significant ($\Delta$ = 0.076,

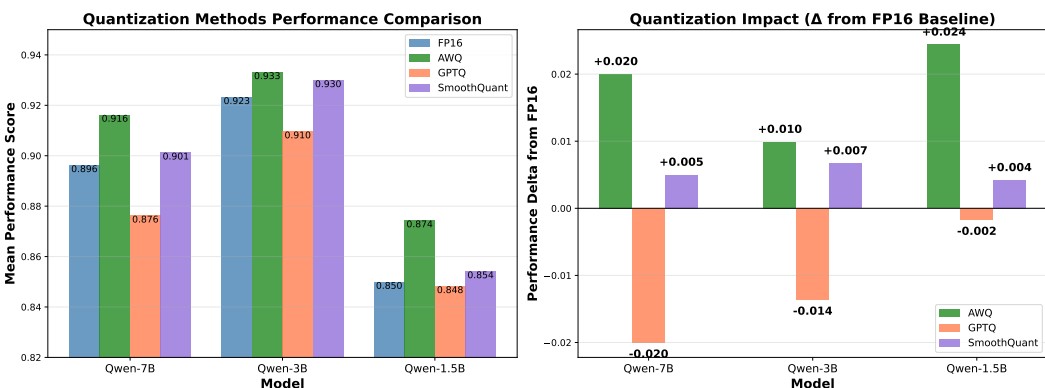

Figure 24: Qwen2.5 Quantization Analysis: (left) Quantization methods comparison, (right) Quantization impact

$p < 0.01$), and GPTQ–SmoothQuant is strongly negative ($\Delta$ = -0.074, $p < 0.01$). In contrast, AWQ and SmoothQuant are statistically indistinguishable from FP16 (all $p > 0.1$), and differences among the three quantized variants other than GPTQ do not reach significance.

Table 11 summarizes quantization performance for LLaMA-3 models (8B, 3B, 1B). For LLaMA-3-8B, AWQ achieves the highest mean alignment (0.911), followed by FP16 and SmoothQuant, with GPTQ lowest (0.881). For LLaMA-3-3B, FP16 has the highest mean (0.929), while all three quantized variants are somewhat lower, with SmoothQuant > AWQ > GPTQ. For LLaMA-3-1B, FP16 and SmoothQuant are nearly identical and clearly above AWQ and GPTQ, with GPTQ again lowest. Overall, the LLaMA results align with our Qwen analyses: GPTQ consistently yields lower brain alignment than FP16, whereas AWQ and SmoothQuant generally preserve full-precision performance, sometimes even slightly improving upon it. This validates our conclusion that carefully designed quantization (AWQ/SmoothQuant) can maintain brain alignment at near full-precision levels, while some schemes (GPTQ) introduce a modest but reliable degradation.

Table 10: Pairwise comparisons of brain-alignment differences across quantization methods for LLaMA-3 models. Each Table reports mean differences ($\Delta$), $t$-statistics, and two-sided significance tests for 8B (left), 3B (right), and 1B (bottom).

| Comparison (A–B) | $\Delta$ | $t(8)$ | Sig. |
|---|---|---|---|
| LLaMA-3-8B – AWQ | -0.010 | -inf | $p < 0.001$ |
| LLaMA-3-8B – GPTQ | 0.020 | inf | $p < 0.001$ |
| LLaMA-3-8B – SmoothQuant | 0.005 | 213621227803258.03 | $p < 0.001$ |
| AWQ – GPTQ | 0.030 | inf | $p < 0.001$ |
| AWQ – SmoothQuant | 0.015 | 640834349907832.12 | $p < 0.001$ |
| GPTQ – SmoothQuant | -0.015 | -640834349907835.25 | $p < 0.001$ |

| Comparison (A–B) | $\Delta$ | $t(8)$ | Sig. |
|---|---|---|---|
| LLaMA-3-3B – AWQ | 0.047 | 1.85 | n.s. ($p \approx 0.12$) |
| LLaMA-3-3B – GPTQ | 0.059 | 2.57 | $p < 0.05$ |
| LLaMA-3-3B – SmoothQuant | 0.036 | 1.47 | n.s. ($p \approx 0.20$) |
| AWQ – GPTQ | 0.012 | 1.87 | n.s. ($p \approx 0.12$) |
| AWQ – SmoothQuant | -0.011 | -3.28 | $p < 0.05$ |
| GPTQ – SmoothQuant | -0.023 | -4.08 | $p < 0.01$ |

| Comparison (A–B) | $\Delta$ | $t(8)$ | Sig. |
|---|---|---|---|
| LLaMA-3-1B – AWQ | 0.035 | 1.88 | n.s. ($p \approx 0.12$) |
| LLaMA-3-1B – GPTQ | 0.076 | 6.64 | $p < 0.01$ |
| LLaMA-3-1B – SmoothQuant | 0.002 | 0.56 | n.s. ($p \approx 0.60$) |
| AWQ – GPTQ | 0.041 | 3.20 | $p < 0.05$ |
| AWQ – SmoothQuant | -0.032 | -1.78 | n.s. ($p \approx 0.14$) |
| GPTQ – SmoothQuant | -0.074 | -5.61 | $p < 0.01$ |

Table 11: Quantization method performance across LLaMA models (mean $\pm$ std over 9 subjects).

| Model | FP16 | AWQ | GPTQ | SmoothQuant |
|---|---|---|---|---|
| LLaMA-3-8B | $0.901 \pm 0.094$ | $0.911 \pm 0.094$ | $0.881 \pm 0.094$ | $0.896 \pm 0.094$ |
| LLaMA-3-3B | $0.929 \pm 0.071$ | $0.882 \pm 0.102$ | $0.870 \pm 0.092$ | $0.893 \pm 0.102$ |
| LLaMA-3-1B | $0.830 \pm 0.091$ | $0.795 \pm 0.133$ | $0.754 \pm 0.115$ | $0.828 \pm 0.092$ |

## K  ROI-SPECIFIC ANALYSIS, BEST LAYER SELECTION AND SUBJECT VARIABILITY.

In our analyses, we extract activations from every transformer layer, and fit a separate voxel-wise encoding model for each layer. For each model, we then compute brain alignment layer-by-layer across the language ROIs and identify the best layer as the one with the highest mean normalized

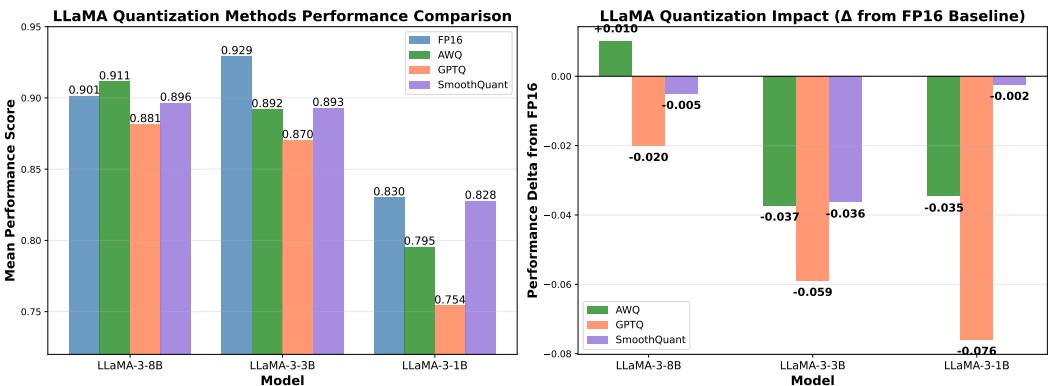

Figure 25: LLaMA-3 Quantization Analysis: (left) Quantization methods comparison, (right) Quantization impact

Table 12: Qwen2.5 Cross-Model Size Comparisons

| Comparison | Base Mean | Comp Mean | Difference | Effect Size | % Sig | Median p | Interpretation |
|---|---|---|---|---|---|---|---|
| 7B vs 3B | 0.7208 | 0.6996 | -0.0213 | 0.596 | 0.0% | 0.549 | Not significant |
| 14B vs 3B | 0.7208 | 0.6912 | -0.0296 | 1.184 | 27.8% | 0.111 | Trend (p < 0.15) |
| 14B vs 7B | 0.6996 | 0.6912 | -0.0083 | 0.874 | 7.8% | 0.356 | Not significant |

predictivity. The main size/quantization comparisons are reported using this model-specific best layer (see Tables 13 and 16).

From Table 13, across language ROIs we find that the best layers are highly consistent within a given model: the same (or adjacent) layer tends to be optimal across ROIs, so we treat the best layer as a model-level property when summarizing results. Overall, across models we observe the familiar pattern that middle-to-late layers yield the strongest brain alignment, with early layers performing clearly worse.

Regarding quantization, we also examined whether AWQ, GPTQ, or SmoothQuant systematically shift the optimal layer. We did not observe any systematic change: for a given architecture, the best layer under quantization is typically the same as in FP16 or within $\pm 1$–2 layers in the same middle/late portion of the network. In other words, quantization affects the magnitude of brain alignment (as analyzed in Tables 9 and 11), but not the qualitative position of the brain-optimal layers. We now clarify this procedure and these observations in the main text and refer explicitly to the layer-wise summaries in Tables 13 and 16.

The best layers are consistent in each model across ROIs; so, we consider best layers specific to each model while reporting the results. Overall, across models, middle to late layers show better brain alignment. Mostly 3b model is the best in terms of brain alignment across ROIs.

## L    ENCODING PERFORMANCE OF DEEPSEEK MODELS

We have now extended our evaluation to an additional model family: Deepseek-R1-Distill at 1.5B, 3B, 7B and 14B. The normalized brain alignment of DeepSeek models across whole brain and language ROI (IFG) is presented in Fig. 26. We also group the quantized variants and present a comparison of the base vs. quantized models across language ROIs in Fig 26. From Fig 26, we observe the same trend: 3B SLMs maintain brain alignment comparable to 14B models, while 1B-1.5B models consistently drop in brain alignment. Notably, the DeepSeek-R1-Distill 14B model shows only a modest improvement over the 7B version, and its alignment is matched by the 3B DeepSeek model, suggesting that 3B SLMs provide sufficient representational capacity for studying brain-LM alignment within this scale regime. We also present the average normalized brain alignment across Language ROIs for DeepSeek-R1 model, comparing SLMs, LLMs, and individual quantized variants in Fig. 27 and the grouped quantized variants in Fig. 28.

Table 13: ROI-Specific Layer Performance Summary

| ROI | Model (Qwen2.5) | Overall Mean±SD | Best Layer | Best Layer Mean±SD | Worst Layer | Layer Range |
|---|---|---|---|---|---|---|
| AG | 1.5B | 0.8341±0.1546 | L14 | 0.9466±0.1099 | L1 | 0.5478–0.9466 |
| AG | 3B | 0.8558±0.1823 | L22 | 1.0091±0.1157 | L1 | 0.5418–1.0091 |
| AG | 7B | 0.8417±0.2119 | L15 | 0.9818±0.1390 | L1 | 0.4535–0.9818 |
| AG | 14B | 0.8178±0.2085 | L24 | 1.0143±0.1431 | L1 | 0.4913–1.0143 |
| ATL | 1.5B | 0.7161±0.1248 | L14 | 0.7904±0.1302 | L1 | 0.5296–0.7904 |
| ATL | 3B | 0.7251±0.1422 | L21 | 0.8304±0.1309 | L1 | 0.5108–0.8304 |
| ATL | 7B | 0.7016±0.1611 | L16 | 0.7891±0.1072 | L1 | 0.4620–0.7891 |
| ATL | 14B | 0.6974±0.1579 | L25 | 0.8362±0.1333 | L1 | 0.4630–0.8362 |
| PTL | 1.5B | 0.7697±0.1300 | L15 | 0.8335±0.1078 | L1 | 0.5944–0.8335 |
| PTL | 3B | 0.7763±0.1394 | L21 | 0.8725±0.1313 | L1 | 0.5843–0.8725 |
| PTL | 7B | 0.7443±0.1616 | L15 | 0.8353±0.1231 | L3 | 0.5166–0.8353 |
| PTL | 14B | 0.7471±0.1513 | L25 | 0.8592±0.1386 | L1 | 0.5353–0.8592 |
| IFG | 1.5B | 0.7726±0.1716 | L14 | 0.8730±0.1480 | L1 | 0.5366–0.8730 |
| IFG | 3B | 0.7801±0.1872 | L21 | 0.9309±0.1377 | L1 | 0.5058–0.9309 |
| IFG | 7B | 0.7665±0.1996 | L15 | 0.9045±0.1305 | L3 | 0.4263–0.9045 |
| IFG | 14B | 0.7639±0.2079 | L25 | 0.9516±0.1027 | L1 | 0.4938–0.9516 |
| MFG | 1.5B | 0.6929±0.1494 | L15 | 0.7618±0.1540 | L1 | 0.5068–0.7618 |
| MFG | 3B | 0.6984±0.1732 | L21 | 0.8006±0.1746 | L1 | 0.5213–0.8006 |
| MFG | 7B | 0.6716±0.1883 | L15 | 0.7732±0.1553 | L1 | 0.4283–0.7732 |
| MFG | 14B | 0.6689±0.1642 | L24 | 0.7921±0.1506 | L1 | 0.4682–0.7921 |
| IFGOrb | 1.5B | 0.6193±0.1589 | L14 | 0.7249±0.1007 | L1 | 0.4160–0.7249 |
| IFGOrb | 3B | 0.6401±0.1712 | L22 | 0.7647±0.0866 | L1 | 0.4072–0.7647 |
| IFGOrb | 7B | 0.6403±0.1966 | L15 | 0.7765±0.1080 | L1 | 0.2891–0.7765 |
| IFGOrb | 14B | 0.6159±0.1878 | L25 | 0.7803±0.0772 | L3 | 0.3628–0.7803 |
| PCC | 1.5B | 0.7638±0.1732 | L15 | 0.8618±0.1261 | L1 | 0.4904–0.8618 |
| PCC | 3B | 0.7651±0.1878 | L22 | 0.9093±0.1203 | L1 | 0.4654–0.9093 |
| PCC | 7B | 0.7509±0.2118 | L15 | 0.8904±0.0938 | L1 | 0.3571–0.8904 |
| PCC | 14B | 0.7235±0.2095 | L24 | 0.9367±0.1163 | L1 | 0.4205–0.9367 |
| dmPFC | 1.5B | 0.6884±0.1392 | L14 | 0.8089±0.0991 | L1 | 0.4443–0.8089 |
| dmPFC | 3B | 0.6964±0.1726 | L21 | 0.8685±0.1049 | L1 | 0.4192–0.8685 |
| dmPFC | 7B | 0.6888±0.1825 | L15 | 0.8226±0.1018 | L1 | 0.3628–0.8226 |
| dmPFC | 14B | 0.6692±0.1913 | L25 | 0.8859±0.1320 | L1 | 0.3712–0.8859 |
| AC | 1.5B | 0.5587±0.0906 | L15 | 0.5963±0.0762 | L1 | 0.4727–0.5963 |
| AC | 3B | 0.5634±0.1056 | L21 | 0.6303±0.0890 | L1 | 0.4668–0.6303 |
| AC | 7B | 0.5204±0.1243 | L15 | 0.5867±0.1017 | L3 | 0.3757–0.5867 |
| AC | 14B | 0.5241±0.0909 | L24 | 0.5831±0.1042 | L1 | 0.4095–0.5831 |

Table 14: Qwen2.5: Subject Variability at Optimal Layers

| Model | Layer | Mean ± SD | SEM | 95% CI | CV (%) | Variability |
|---|---|---|---|---|---|---|
| 1.5B | L14 | 0.7956 ± 0.0892 | 0.0364 | [0.724, 0.867] | 11.21% | LOW |
| 3B | L22 | 0.8309 ± 0.0835 | 0.0341 | [0.764, 0.898] | 10.05% | LOW |
| 7B | L15 | 0.8171 ± 0.0852 | 0.0348 | [0.749, 0.885] | 10.43% | LOW |
| 14B | L24 | 0.8411 ± 0.0883 | 0.0361 | [0.770, 0.912] | 10.50% | LOW |

Table 15: Cross-Model Comparisons for Llama-3

| Comparison | Base Mean | Comp Mean | Difference | Effect Size | % Sig | Median $p$ | Interpretation |
|---|---|---|---|---|---|---|---|
| 8B vs 3B | 0.7409 | 0.7300 | -0.0110 | 0.425 | 5.2% | 0.417 | Not significant |
| 14B vs 3B | 0.7409 | 0.7494 | +0.0084 | 0.811 | 27.4% | 0.227 | Moderate effect |
| 14B vs 8B | 0.7285 | 0.7517 | +0.0232 | 0.577 | 14.6% | 0.320 | Not significant |

## M   ENCODING PERFORMANCE ON NATURALISTIC READING fMRI DATASET

We have now extended our experiments to an additional dataset i.e. we performed voxelwise encoding on the Moth Radio Hour Reading fMRI dataset (Deniz et al., 2019), which contains the same nine subjects and large number of samples under a different linguistic paradigm (i.e. reading). This additional evaluation allows us to assess the generalizability of our findings across datasets and tasks. We use Qwen models (Qwen2.5-1.5b, Qwen2.5-3b, Qwen2.5-7b and Qwen2.5-14b) to validate the brain alignment to examine whether 3b SLMs maintain similar brain alignment to larger versions of the models. From Fig. 29, we observe that 3B SLMs yield brain alignment comparable to the 7B and 14B Qwen2.5 models, whereas 1.5B SLMs exhibit a clear drop in brain alignment on the Reading Brain dataset.

## N   DECODER GAP: BRAIN DECODING (STIMULUS RECONSTRUCTION)

We now perform brain decoding to reconstruct text stimuli from fMRI brain activity using LLaMA-3-8B, and two SLMS (LLaMA-3-3B and LLaMA-3-1B).

Table 16: LLaMA3: ROI-Specific Layer Performance Summary

| ROI | Model (LLaMA-3) | Optimal Layer | Optimal Value | SD | CI Low | CI High | High-Perf Range | Total Layers |
|---|---|---|---|---|---|---|---|---|
| AG | 1B | 9 | 0.9189 | 0.0867 | 0.8495 | 0.9882 | 7–14 | 16 |
| ATL | 1B | 9 | 0.7751 | 0.0789 | 0.7119 | 0.8382 | 5–14 | 16 |
| PTL | 1B | 8 | 0.8149 | 0.1134 | 0.7242 | 0.9056 | 5–14 | 16 |
| IFG | 1B | 8 | 0.8312 | 0.1209 | 0.7345 | 0.9279 | 5–15 | 16 |
| MFG | 1B | 9 | 0.7477 | 0.1623 | 0.6178 | 0.8775 | 7–15 | 16 |
| IFGOrb | 1B | 7 | 0.6937 | 0.0868 | 0.6242 | 0.7631 | 5–15 | 16 |
| PCC | 1B | 7 | 0.8364 | 0.1109 | 0.7476 | 0.9251 | 5–14 | 16 |
| dmPFC | 1B | 9 | 0.7591 | 0.0995 | 0.6795 | 0.8387 | 7–11 | 16 |
| EarlyAud | 1B | 8 | 0.6010 | 0.0802 | 0.5368 | 0.6651 | 5–9 | 16 |
| AG | 3B | 12 | 1.0206 | 0.1073 | 0.9347 | 1.1065 | 9–19 | 28 |
| ATL | 3B | 12 | 0.8150 | 0.1111 | 0.7261 | 0.9039 | 9–21 | 28 |
| PTL | 3B | 13 | 0.8654 | 0.1246 | 0.7657 | 0.9651 | 9–21 | 28 |
| IFG | 3B | 13 | 0.9255 | 0.1244 | 0.8259 | 1.0250 | 9–22 | 28 |
| MFG | 3B | 13 | 0.8127 | 0.1439 | 0.6975 | 0.9278 | 11–22 | 28 |
| IFGOrb | 3B | 12 | 0.8030 | 0.0739 | 0.7439 | 0.8622 | 11–17 | 28 |
| PCC | 3B | 14 | 0.9151 | 0.1105 | 0.8267 | 1.0036 | 9–22 | 28 |
| dmPFC | 3B | 12 | 0.8464 | 0.0846 | 0.7787 | 0.9141 | 11–17 | 28 |
| EarlyAud | 3B | 13 | 0.6271 | 0.0905 | 0.5547 | 0.6995 | 12–17 | 28 |
| AG | 7B | 14 | 0.9802 | 0.1095 | 0.8927 | 1.0678 | 7–29 | 32 |
| ATL | 7B | 14 | 0.8098 | 0.1198 | 0.7139 | 0.9056 | 7–25 | 32 |
| PTL | 7B | 14 | 0.8644 | 0.1082 | 0.7778 | 0.9509 | 7–25 | 32 |
| IFG | 7B | 14 | 0.8974 | 0.1270 | 0.7958 | 0.9990 | 7–25 | 32 |
| MFG | 7B | 13 | 0.7792 | 0.1082 | 0.6926 | 0.8658 | 7–26 | 32 |
| IFGOrb | 7B | 14 | 0.7415 | 0.1004 | 0.6612 | 0.8219 | 7–29 | 32 |
| PCC | 7B | 15 | 0.8835 | 0.1203 | 0.7872 | 0.9797 | 7–29 | 32 |
| dmPFC | 7B | 14 | 0.8353 | 0.1043 | 0.7519 | 0.9188 | 7–18 | 32 |
| EarlyAud | 7B | 13 | 0.5940 | 0.0756 | 0.5336 | 0.6545 | 7–21 | 32 |
| AG | 14B | 17 | 0.9861 | 0.1001 | 0.9061 | 1.0662 | 5–38 | 39 |
| ATL | 14B | 16 | 0.7910 | 0.1175 | 0.6970 | 0.8851 | 6–38 | 39 |
| PTL | 14B | 16 | 0.8204 | 0.1121 | 0.7308 | 0.9101 | 6–35 | 39 |
| IFG | 14B | 16 | 0.8709 | 0.1055 | 0.7865 | 0.9553 | 8–35 | 39 |
| MFG | 14B | 13 | 0.7340 | 0.1315 | 0.6288 | 0.8393 | 5–20 | 39 |
| IFGOrb | 14B | 17 | 0.7772 | 0.0460 | 0.7404 | 0.8141 | 12–30 | 39 |
| PCC | 14B | 16 | 0.9076 | 0.1142 | 0.8162 | 0.9990 | 8–35 | 39 |
| dmPFC | 14B | 16 | 0.8389 | 0.1310 | 0.7341 | 0.9436 | 14–23 | 39 |
| EarlyAud | 14B | 17 | 0.5544 | 0.0720 | 0.4969 | 0.6120 | 5–20 | 39 |

Table 17: LLaMA3: Subject Variability at Optimal Layers

| Model | Layer | Mean ± SD | SEM | 95% CI | CV (%) | Variability |
|---|---|---|---|---|---|---|
| 1B | L8 | 0.8003 ± 0.1065 | 0.0435 | [0.688, 0.912] | 13.31% | MODERATE |
| 3B | L13 | 0.9083 ± 0.0824 | 0.0337 | [0.822, 0.995] | 9.08% | LOW |
| 8B | L10 | 0.8588 ± 0.1115 | 0.0455 | [0.742, 0.976] | 12.98% | MODERATE |
| 14B | L16 | 0.8679 ± 0.0936 | 0.0382 | [0.770, 0.966] | 10.78% | MODERATE |

Inspired by BrainLLM (Ye et al. 2025), we perform end-to-end text stimulus reconstruction from fMRI brain activity. We follow the same BrainLLM methodology, where we use the same Moth Radio Hour dataset (11 stories) with the same train/test split, where ten stories are used for training and one held-out story is used for generation. Concretely, we train a brain-to-text decoder and report standard text-generation metrics-BLEU-1, WER, METEOR, and BERT-F1-for three models: LLaMA-3-8B, LLaMA-3.2-3B, and LLaMA-3.2-1B (Table 18). Across reconstructed segments per model on test dataset, LLaMA-3.2-3B achieves the best performance on all four metrics (BLEU-1 = 0.120, WER = 4.22, METEOR = 0.110, BERT-F1 = 0.825), slightly outperforming LLaMA-3.2-8B and clearly improving over the LLaMA-3.2-1B baseline (BLEU-1 = 0.070, METEOR = 0.055, BERT-F1 = 0.811). These BERT-F1 scores in the 0.81–0.83 range indicate that the decoded text reliably preserves the semantic content of the original stimulus, while BLEU-1 in the 0.07–0.12 range is in line with prior work where exact word-level recovery from fMRI is known to be challenging.

To make the reconstruction quality more interpretable, we also include qualitative examples comparing ground-truth text and decoded outputs (Table. 19). These examples illustrate that the decoder often recovers the overall meaning, emotional tone, and discourse context, even when individual words differ-e.g., reconstructions that correctly express embarrassment, uncertainty, or interactions with children, despite not matching every token verbatim.

Overall, these new brain decoding experiments show that our SLMs are not only good encoders of brain activity, but also support meaningful decoding: they can reconstruct linguistically coherent text from fMRI with high semantic fidelity and reasonable word-level accuracy. We emphasize that decoding is not the primary focus of the present work, but the added section demonstrates that the

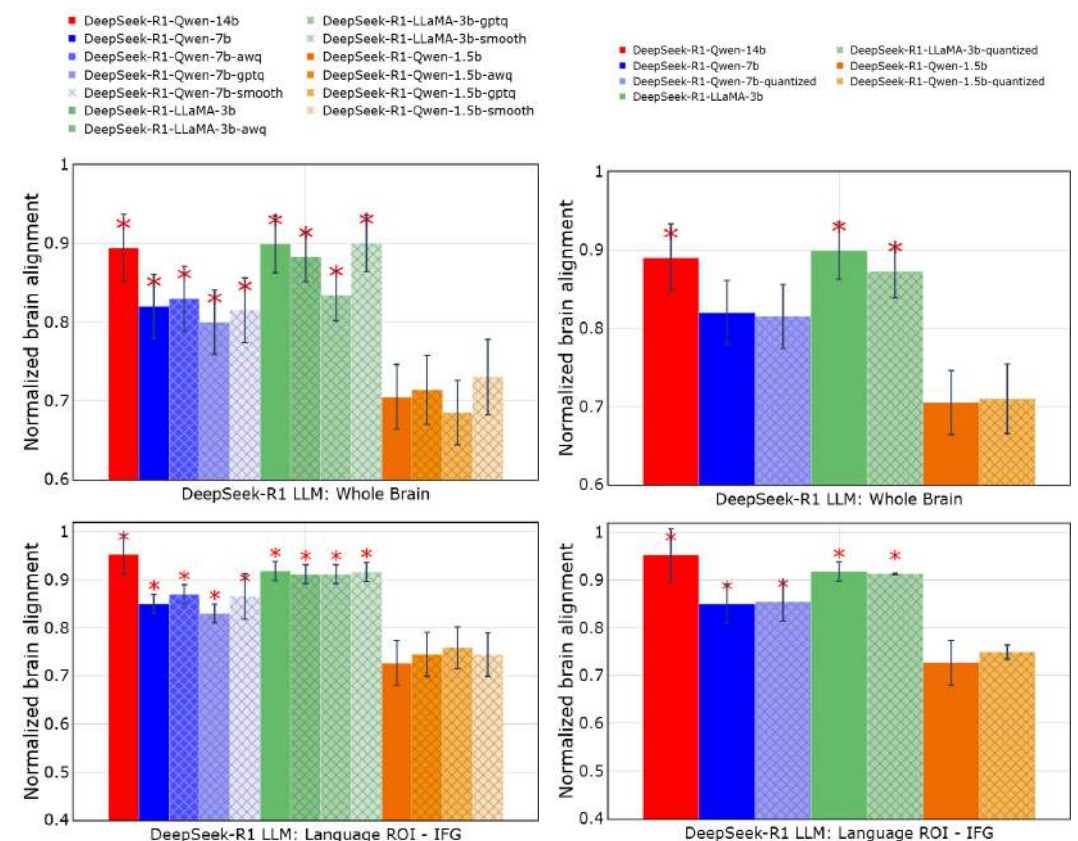

Figure 26: DeepseekR1: Normalized brain alignment was computed by averaging across participants, layers, and voxels. Red: 14b, Blue: 7b, Green: 3b, Orange: 1.5b, Solid: full-precision SLMs/LLMs, Patterned: quantized models. * at a particular bar indicates that the model's prediction performance is significantly better than 1b/1.5b SLMs. The top row shows whole-brain normalized alignment, while the bottom row focuses on a language-selective ROI (IFG).

brain-aligned representations we study are indeed rich enough to support stimulus reconstruction, directly addressing the reviewer's concern.

Table 18: Overall performance metrics for brain-to-text decoding.

| Model | BLEU-1 | WER | METEOR | BERT-F1 | Samples |
|---|---|---|---|---|---|
| LLaMA-3-8B | 0.0699 | 5.7839 | 0.0550 | 0.8108 | 784 |
| LLaMA-3.2-3B | 0.1198 | 4.2237 | 0.1101 | 0.8252 | 784 |
| LLaMA-3.2-1B | 0.1105 | 4.4869 | 0.0990 | 0.8237 | 784 |

## O    EFFECT OF PRUNING

Our original focus was on post-training quantization, because it is widely used in practice, easy to deploy without re-training, and directly targets memory/throughput constraints.

Based on the reviewer's suggestion, we have now included unstructured pruning which is equivalent to quantization and now report preliminary results for Qwen-2.5 models. In particular, we perform unstructured magnitude pruning on the linear layers of Qwen2.5-3B and Qwen2.5-1.5B, at sparsity levels 0.1, 0.25, and 0.5. For Qwen2.5-3B, Table 20 summarizes brain alignment (mean ± s.e.m. across subjects) for the base, quantized variants, and pruned models, showing that AWQ and SmoothQuant slightly improve over the FP16 baseline ($0.933 \pm 0.035$ and $0.930 \pm 0.035$ vs. $0.924 \pm 0.033$), GPTQ is modestly lower ($0.910 \pm 0.037$), and unstructured pruning up to 50% keeps alignment in a narrow range (0.910–0.907 with s.e.m. 0.032–0.043).

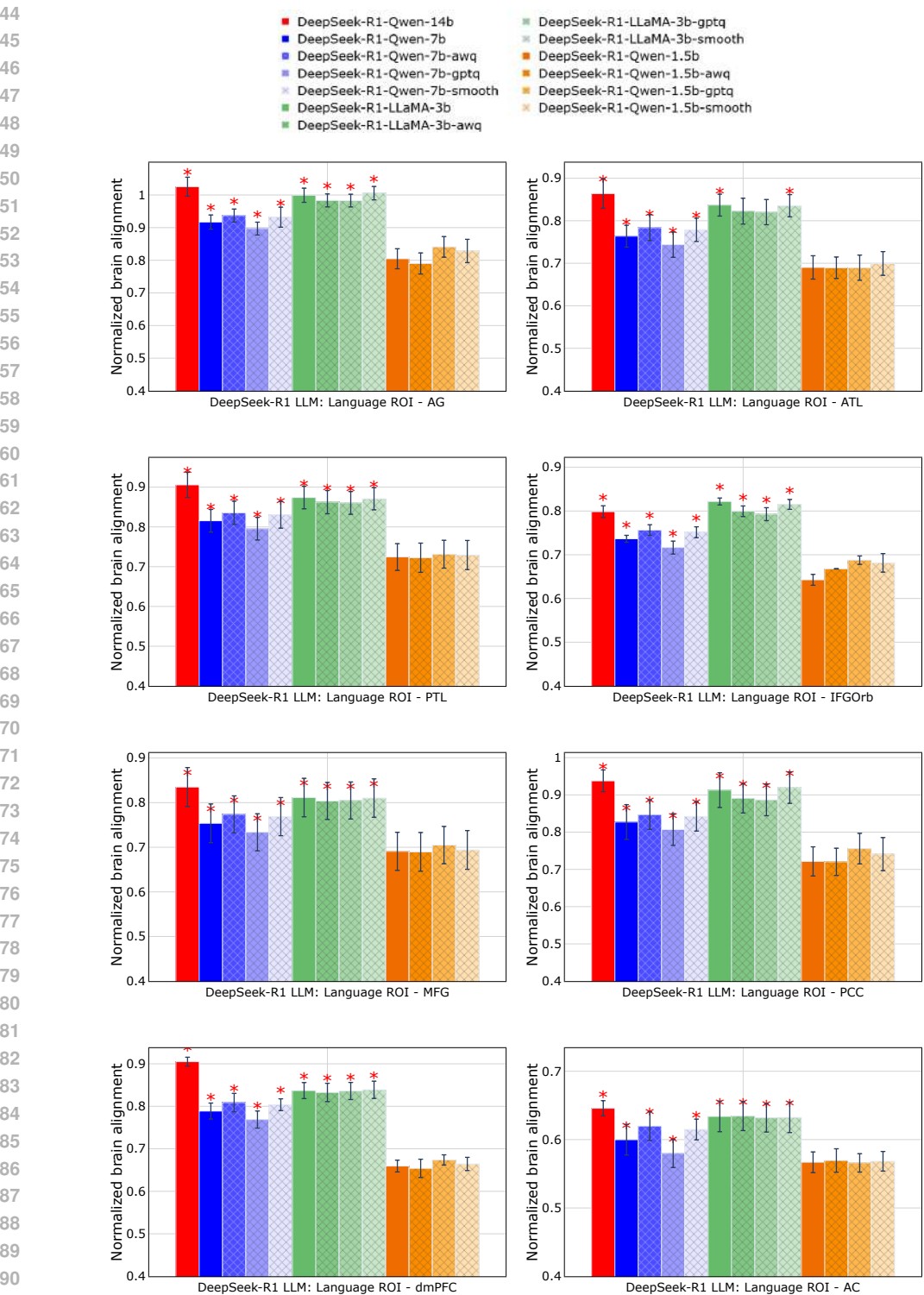

Figure 27: Normalized Predictivity of SLMs, LLMs, and Quantized Language Models for DeepSeek-R1 models.

These results suggest that, for Qwen2.5-3B, moderate unstructured pruning (10–25%) preserves brain alignment at a level comparable to quantized or full-precision models, with little change in SER, while aggressive pruning (50%) begins to degrade linguistic competence (SER increases) de-

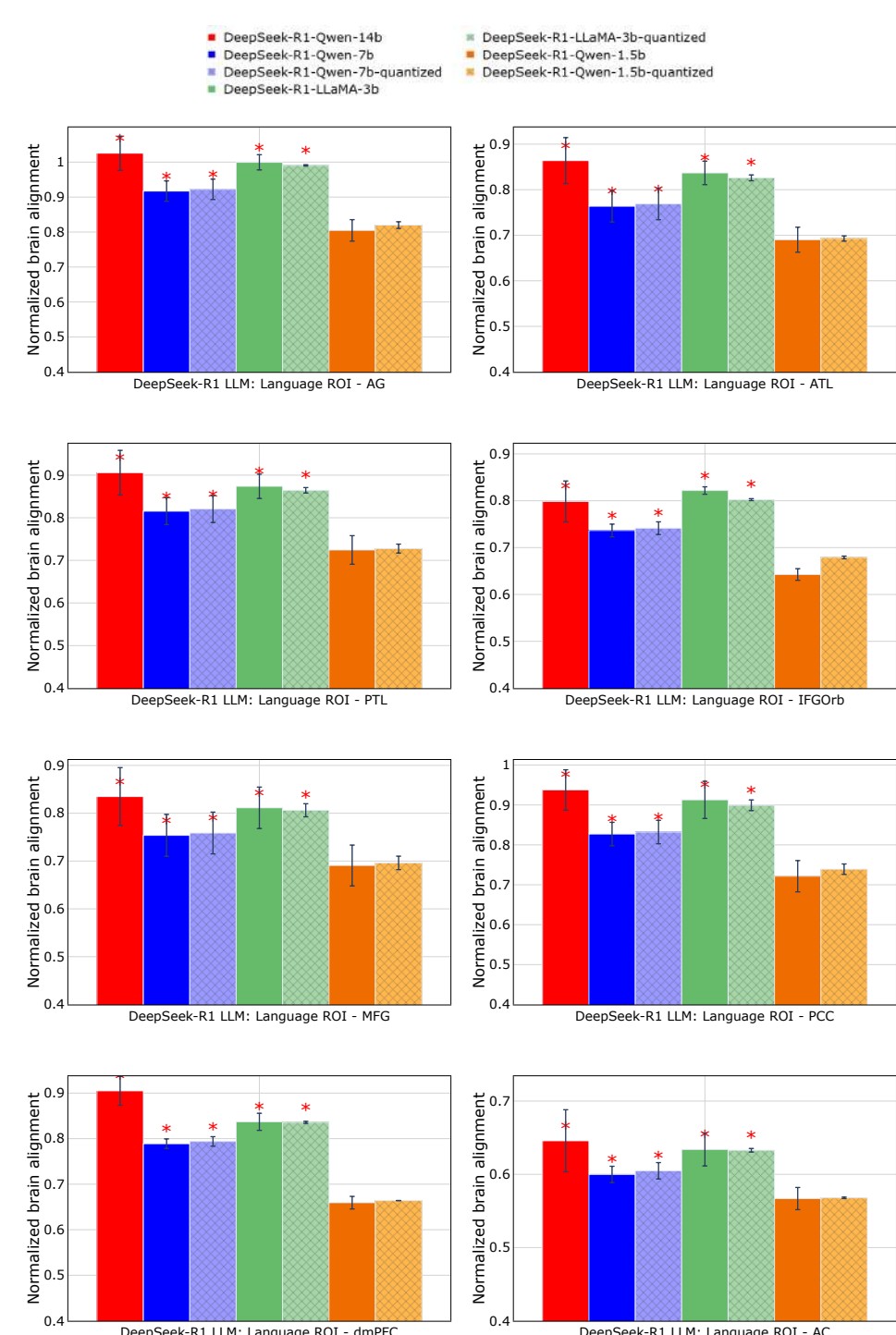

Figure 28: Normalized predictivity of DeepSeek-R1 SLMs and LLMs, including grouped comparisons of the base and quantized variants.

spite only a small drop in alignment. We view these pruning experiments as complementary to our main quantization results: they show that both post-training quantization and unstructured pruning can preserve brain alignment to a surprising degree, but they also highlight potential trade-offs with linguistic competence at high sparsity.

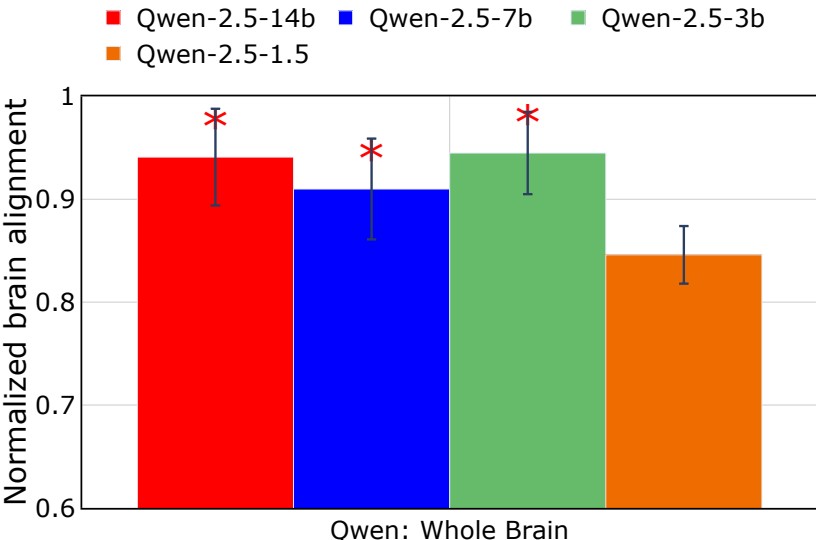

Figure 29: Reading brain dataset (Qwen-2.5): Normalized brain alignment was computed by averaging across participants, layers, and voxels. Red: 14b, Blue: 7b, Green: 3b, Orange: 1.5b, Solid: full-precision SLMs/LLMs. * at a particular bar indicates that the model's prediction performance is significantly better than 1.5b SLMs. The plot shows whole-brain normalized alignment.

Table 19: Example fMRI-based stimulus reconstructions for LLaMA-3-8B, LLaMA-3.2-3B, and LLaMA-3.2-1B.

| Example | Variant | Text | Metrics |
|---------|---------|------|---------|
| 1 | Ground Truth | *little sad leaving here about* | – |
| | LLaMA-3-8B | huge fan of the show so i'm really excited to be able to share this with you guys and i hope you'll enjoy it as much as i enjoyed watching | BLEU-1 = 0.0000, WER = 6.0000, METEOR = 0.0000 |
| | LLaMA-3.2-3B | in this beautiful city. I started to walk in this beautiful in this beautiful city. I started to walk in this beautiful in this beautiful city. I started to | BLEU-1 = 0.0000, WER = 5.8000, METEOR = 0.0000 |
| | LLaMA-3.2-1B | , you know what i'm talking about. You know what i'm talking about. You know what i'm talking about. You know what i'm talking about | BLEU-1 = 0.0400, WER = 4.8000, METEOR = 0.0714 |
| 2 | Ground Truth | *and that i had to hold onto him* | – |
| | LLaMA-3-8B | i was going to have to take him to the hospital but i had no idea what to do with him so i went to a friend of mine and he told | BLEU-1 = 0.3750, WER = 3.5000, METEOR = 0.2885 |
| | LLaMA-3.2-3B | , she's crying and then she sees me and she com, she's crying and then she sees me and she com, she's crying and then she sees | BLEU-1 = 0.1852, WER = 3.2500, METEOR = 0.0505 |
| | LLaMA-3.2-1B | could see her face and she was smiling at me and then she said "hi" and i said "hi" back to her and then she turned around and | BLEU-1 = 0.2500, WER = 3.1250, METEOR = 0.1500 |
| 3 | Ground Truth | *to be a teacher with all these kids* | – |
| | LLaMA-3-8B | for them to see their children i had no idea how hard it was for them to see their children i had no idea how hard it was for them to see | BLEU-1 = 0.0938, WER = 3.8750, METEOR = 0.0962 |
| | LLaMA-3.2-3B | is a great place to live, work, and play. It's also home to some of the best restaurants, bars, and coffee shops in the city. | BLEU-1 = 0.1200, WER = 3.0000, METEOR = 0.2635 |
| | LLaMA-3.2-1B | and i said you know like and i said you know like and i said you know like and i said you know like and i said you know like and i | BLEU-1 = 0.0000, WER = 4.0000, METEOR = 0.0000 |

We acknowledge that we still do not systematically explore structured compression across model components, where different methods could be compared under matched compression ratios to assess their differential impact on both linguistic competence and brain alignment. We now explicitly note that this is an important direction for future work, but falls outside the scope of the current paper given the substantial additional experiments it would require.

Although our model suite includes the DeepSeek-R1-Distill family (which is itself a product of knowledge distillation), in this work we do not systematically study distillation as a compression method. We treat DeepSeek as an additional, pretrained model family for evaluation, and focus our controlled compression experiments on post-training quantization (and preliminary unstructured pruning). A careful comparison of different distillation strategies under matched compression ratios is therefore an important direction for future work.

Table 20: Comparison of quantization and pruning for Qwen2.5-3B.

| Model variant | Method | Sparsity | Normalized Brain Alignment |
|---|---|---|---|
| Qwen-2.5-3B | FP16 (baseline) | 0% | 0.924 ±0.033 |
| Qwen-2.5-3B-AWQ | Quantization (AWQ) | 0% | 0.933 ± 0.035 |
| Qwen-2.5-3B-GPTQ | Quantization (GPTQ) | 0% | 0.910 ± 0.037 |
| Qwen-2.5-3B-Smooth | Quantization (SmoothQuant) | 0% | 0.930 ± 0.035 |
| Qwen-2.5-3B-0.1 | Pruning | 10% | 0.910 ± 0.032 |
| Qwen-2.5-3B-0.25 | Pruning | 25% | 0.908 ± 0.033 |
| Qwen-2.5-3B-0.5 | Pruning | 50% | 0.907 ± 0.043 |

Table 21: Comparison of quantization and pruning for Qwen2.5-3B (mean $\pm$ SEM across subjects).

| Variant | Mean ± SEM | 95% CI | Notes |
|---|---|---|---|
| FP16 (baseline) | 0.830 ± 0.025 | [0.781, 0.879] | full precision |
| AWQ | 0.854 ± 0.031 | [0.793, 0.915] | INT4 quantization |
| GPTQ | 0.828 ± 0.026 | [0.777, 0.879] | INT4 quantization |
| SmoothQuant | 0.836 ± 0.025 | [0.787, 0.885] | INT4 quantization |
| Prune 10% | 0.847 ± 0.026 | [0.796, 0.898] | unstructured pruning |
| Prune 25% | 0.824 ± 0.025 | [0.775, 0.873] | unstructured pruning |
| Prune 50% | 0.608 ± 0.053 | [0.504, 0.712] | unstructured pruning |

## P    MODEL SIZE VS. BRAIN ALIGNMENT

Figs. 30, 31, and 32 plot Model size (GB) vs. Average Normalized brain alignment for Qwen2.5, LLaMA-3, and DeepSeek-R1 models (1.5B/3B/7B/14B) and their AWQ, GPTQ, and SmoothQuant variants. Across all three families, the 3B SLMs and their AWQ/SmoothQuant variants generally lie slightly above or very close to their FP16 counterparts at substantially reduced size, whereas GPTQ variants tend to fall slightly below the FP16 models despite achieving stronger compression.

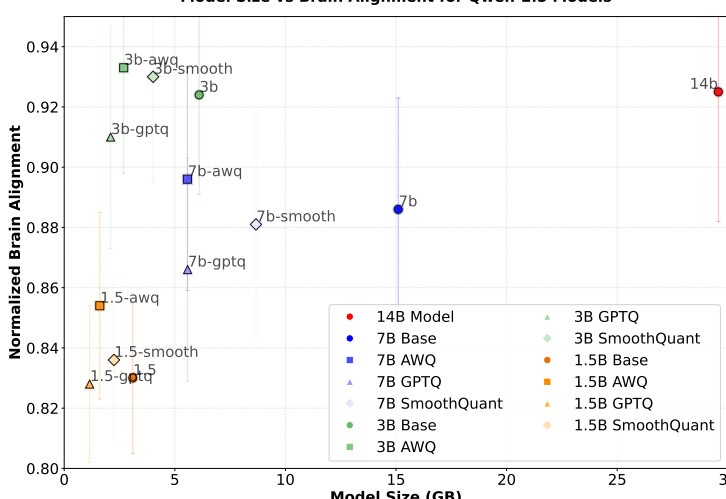

Figure 30: Qwen2.5: Plot of Model Size (x-axis) vs. Normalized Brain Alignment (y-axis).

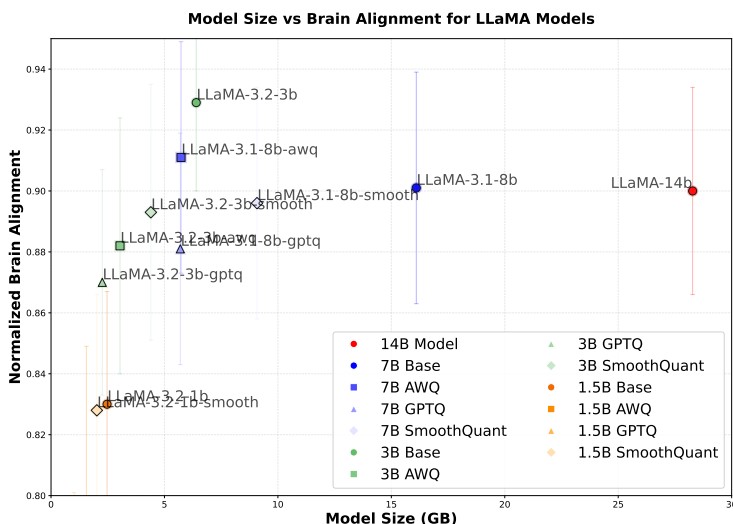

Figure 31: LLaMA-3: Plot of Model Size (x-axis) vs. Normalized Brain Alignment (y-axis).

Table 22: DeepSeek-R1: Pretrained Transformer-based language models: SLMs, LLMs and their quantized variants.

| Model Family | SLMs | | LLMs | | Model Family | SLMs (GB) | | | LLMs (GB) | | |
| --- | --- | --- | --- | --- | --- | --- | --- | --- | --- | --- | --- |
| | Size | Layers | Size | Layers | | AWQ | GPTQ | SmoothQuant | AWQ | GPTQ | SmoothQuant |
| DeepSeek-R1 | 1.5B (3.55GB) | 28 | 7B (15.23GB) | 28 | DeepSeek-R1 | 1.62 | 1.61 | 2.25 | 5.57 | 5.58 | 8.71 |
| | 3B (6.43GB) | 28 | 14B (29.54) | 48 | | 2.69 | 2.37 | 4.02 | | | |

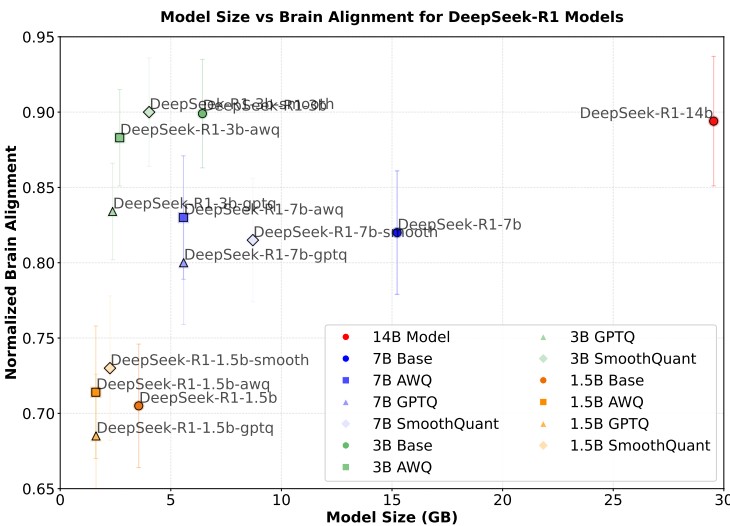

Figure 32: DeepSeek-R1: Plot of Model Size (x-axis) vs. Normalized Brain Alignment (y-axis).

