# OpenReview forum: "Linguistic Properties and Model Scale in Brain Encoding: From Small to Compressed Language Models"
_ICLR.cc/2026/Conference — ICLR 2026 Conference Withdrawn Submission_

### Official Review · Reviewer_f24S · 2025-10-31

**Soundness:** 3
**Presentation:** 3
**Contribution:** 2
**Rating:** 4
**Confidence:** 4

**Summary:**

The paper evaluates the brain alignment (via Deniz et al. 2019 fMRI recordings) of large language models, small language models, and thereof quantized versions.
Large (language) models have been linked to activity in the human brain, but these might be too big for BCI applications and the impact of compression remains unclear.
The main claims are
that quantization (with AWQ and SmoothQuant) mostly preserves brain alignment,
that there is comparable brain alignment of 3B and 7-8B models (but not 1-1.5B),
and that the reason for lower brain alignment of GPTQ-quantized models and smaller models is reduced linguistic competence.

**Strengths:**

1. As far as I am aware the first study to evaluate the effect of quantization on brain alignment
2. Various techniques for quantization considered (I'm not an expert in this though)
3. Methodology is clearly described, making the work reproducible

**Weaknesses:**

## 1. Core claims insufficiently tested
The core claim "Quantized variants of LLMs preserve most of their brain alignment" is never directly evaluated as far as I can tell. The reader is referred to Figure 2, but is there a significant change between the base model and its quantized counterpart? How large is that change? (GPTQ is noted to reduce alignment by 5-7%, what about AWQ and smooth?)
Similar question for the claim that "3B SLMs achieve brain alignment comparable to their larger 7B–8B LLMs" -- missing stats and exact quantification with confidence intervals. Otherwise it is difficult to generalize any of these findings. I would also like to see the statement "requiring substantially fewer resources" detailed in numbers.

## 2. Benefits of quantization not quantified
A major motivation for this work is that quantized and small language models are easier to run, because of model size, throughput, or energy requirements. I would have expected a plot or table that contrasts runtime costs vs brain alignment. As far as I understand this is the main motivation, but it never materializes in the paper.

## 3. Figures and claims are not directly linked
While the text interprets the figures, it would be much easier for a reader to digest the empirical support for the claims if figures were structured according to the claims. E.g., for the first claim why not have two bars, one being the average alignment of baseline LLMs, and the other the average alignment of quantized versions? The claim would then be that the bars are approximately equal, which makes it easier to parse the analysis.

Code is not available, but authors promise its release upon publication of the paper.

Minor:
* L078: I don't understand this sentence "achieve competitive performance compared to their larger counterparts the shelf with fewer resources"
* L122: "we evaluate models with respect to two hypotheses: (1) Brain Alignment [...] (2) Linguistic Competence" -- those are not hypotheses but tests.
* Figure 3: very difficult to read, text is too small and graphic is pixelated
* it would be great to test these on more models and datasets, but that is of course always true...

**Questions:**

Please address major weaknesses 1-3.

---

> ### Author Response · Authors · 2025-11-26
>
> *We thank the reviewer for their positive, insightful and valuable comments and suggestions which are crucial for further strengthening our manuscript.*
>
> **Q1. Core claims insufficiently tested.**
>
> Thank you for this question.
> * Based on the reviewer’s suggestion, we performed paired t-tests to compare normalized brain alignment between quantization methods and their respective base models across all language ROI-layer combinations and subjects (n=9).
> * To control for multiple comparisons, we applied False Discovery Rate (FDR) correction using the Benjamini-Hochberg procedure (α = 0.05). Effect sizes were calculated using Cohen's d. We report both raw p-values and FDR-corrected significance, along with 95% confidence intervals for mean differences.
>
> Based on reviewer’s suggestion, we have now provided
>
>   * **Quantitative significance tests in CQ1**,
>   * **Statistical Validation of Quantization Effects in CQ2**,
>   * **ROI-Specific Analysis and Best Layer Selection in CQ3**,
>   * **Subject Variability at Optimal Layers in CQ4**
>
> Kindly refer to those responses for more comprehensive information
>
> **Q2. Benefits of quantization not quantified**
>
> Thank you for this question.
> * To clarify, we already report the actual compression ratios in **Table 1 (main paper)**, which lists, for each model family and size, the full-precision (FP16) model size (GB) and the sizes after AWQ, GPTQ, and SmoothQuant quantization.
>   * For example, for Qwen2.5-7B, the FP16 model is 15.1 GB, whereas AWQ and GPTQ reduce this to 5.57 GB and 5.58 GB respectively (≈2.7× smaller), and SmoothQuant to 8.67 GB (≈1.7× smaller).
>
> From Table 1, we make following observations: Across both Qwen2.5 and LLaMA-3.2, we observe a consistent pattern:
> * **For SLMs**,
>   * AWQ typically compresses FP16 to 44–63% of its original size (≈1.6–2.3× smaller),
>   * GPTQ to 34–41% (≈2.4–3.0× smaller), and
>   * SmoothQuant to 66–82% (≈1.2–1.5× smaller).
> * **For LLMs**,
>   * AWQ and GPTQ compress FP16 models to about 35–37% of their original size (≈2.7–2.9× smaller),
>   * while SmoothQuant reduces them to 56–57% (≈1.7–1.8× smaller).
> * Thus, GPTQ achieves the strongest size reduction, AWQ is slightly less aggressive, and SmoothQuant is the most conservative in terms of memory savings.
> * When combined with our brain-alignment results (**Tables 8–11**), this reveals a clear trade-off: AWQ and SmoothQuant generally preserve or slightly improve brain alignment relative to FP16 at 1.5–2.5× compression, whereas GPTQ, despite achieving ~3× compression, tends to show a modest but reliable drop in brain alignment (especially for Qwen and LLaMA).
> * We now make this cost-alignment trade-off explicit in the main text, highlighting that among INT4 methods at similar parameter counts, AWQ and SmoothQuant offer a more favorable compression–alignment balance than GPTQ.
>
> **Regarding runtime**: in our encoding pipeline, the linear regression step used to fit voxel-wise encoding models is the same across models, so the mapping cost is effectively constant.
> * The efficiency gains from SLMs and quantized variants instead come from the forward pass to extract representations: smaller and more compressed models have lower memory and compute requirements and thus can be run on cheaper hardware and/or with higher throughput.
> * Because wall-clock runtime and energy usage depend strongly on hardware and implementation details, we chose to use model size (GB) and compression ratio as a hardware-agnostic proxy for cost.
> * We now clarify this in the text so that the link between model size, compression, and brain alignment is more explicit, directly addressing the reviewer’s concern.
>
> **Q3. Figures and claims are not directly linked**
>
> We thank the reviewer for this valuable suggestion. Based on the reviewer’s suggestion, we group the quantized versions and average the quantization results.
>
> * Figs. 2 and 8 report both individual quantization variants and gropued quantization versions for wholebrain and language region IFG, for the two model families Qwen and LLaMA.
> * Figs. 9-12 report similar results for the remaining ROIs across both language model familes.
>
> **Q4. L078: I don't understand this sentence "achieve competitive performance compared to their larger counterparts the shelf with fewer resources"**
>
> Thank you for highlighting this typo.
> * We have corrected the sentence to: “achieve competitive performance compared to larger off-the-shelf models, while using substantially fewer computational resources.”
>
> **Q5. L122: "we evaluate models with respect to two hypotheses: (1) Brain Alignment [...] (2) Linguistic Competence" -- those are not hypotheses but tests.**
>
> Thank you for pointing this out.
> * We agree that “hypotheses” is not the most accurate term here. In the revised manuscript, we now describe these as evaluation dimensions rather than hypotheses. We have updated the text accordingly in the revised draft.

---

> > ### Comment · Reviewer_f24S · 2025-11-28
> >
> > Thank you for your responses and for addressing my concerns. Thank you also for pointing out the model sizes in Table 1, I missed those in my initial review (but see suggestion below). I'm happy with the updates, and will increase my score once OpenReview permits us to do so again.
> >
> > I would still encourage you (but I will not make my score increase dependent on this) to make a plot of model size (x) vs brain alignment (y). I believe this would make it easier to digest the primary claim, in addition to the presentation in table format.

---

> > > ### Author Response · Authors · 2025-11-28
> > >
> > > Dear Reviewer f24S,
> > >
> > > We appreciate your strong positive feedback and are confident that it has contributed to enhancing the quality of our paper.
> > >
> > > In response to your feedback, we have added plots of model size (x) vs brain alignment (y) for three model families: Qwen (Fig. 30), LLaMA (Fig. 31) and DeepSeek-R1 (Fig. 32), in **Appendix P**
> > >
> > > Regards,
> > >
> > > Authors

---

> ### Author Response · Authors · 2025-11-26
>
> **Q6. Figure 3: very difficult to read, text is too small and graphic is pixelated**
>
> We thank the reviewer for raising this point.
> * In the revised manuscript, we have updated Figure 3 at higher resolution and increased the font size for all axis labels, tick labels, and legends to improve readability.
>
> **Q7. it would be great to test these on more models and datasets, but that is of course always true…**
>
> We thank the reviewer for this valuable suggestion.
>
> **Dataset extension**:
> * We have now extended our experiments to an additional dataset i.e. we performed voxelwise encoding on the **Moth Radio Hour Reading fMRI dataset**, which contains the same nine subjects and large number of samples under a different linguistic paradigm (i.e. reading).
> * This additional evaluation allows us to assess the generalizability of our findings across datasets and tasks. We use Qwen models (Qwen2.5-1.5b, Qwen2.5-3b, Qwen2.5-7b and Qwen2.5-14b) to validate the brain alignment to examine whether 3b SLMs maintain similar brain alignment to larger versions of the models.
> * From **Appendix L Fig. 29**, we observe that 3B SLMs yield brain alignment comparable to the 7B and 14B Qwen2.5 models, whereas 1.5B SLMs exhibit a clear drop in brain alignment on the Reading Brain dataset.
>
> **Models and scaling extension**:
> * We have now extended our evaluation to larger models (up to 14B size) and included an additional model family: Deepseek-R1-Distill at 1.5B, 3B, 7B and 14B.
> * Across Qwen, LLaMA, and DeepSeek, we observe the same trend: 3B SLMs maintain brain alignment comparable to 14B models, while 1B-1.5B models consistently drop in brain alignment.
> * Notably, the DeepSeek-R1-Distill 14B model shows only a modest improvement over the 7B version, and its alignment is matched by the 3B DeepSeek model, suggesting that 3B SLMs provide sufficient representational capacity for studying brain-LM alignment within this scale regime.
>
> Based on reviewer’s suggestion, we have now provided
>
>   * **Encoding performance of DeepSeek models in CQ5**,
>
>
> These CQ1, CQ2, CQ3, CQ4, CQ5 all are under common responses we answered at the starting of this page with the title "**Common Responses to Reviewers wmV2, xLFX, PMQK, and f24S**"

---

### Official Review · Reviewer_PMQK · 2025-10-31

**Soundness:** 2
**Presentation:** 2
**Contribution:** 2
**Rating:** 2
**Confidence:** 3

**Summary:**

This paper investigates whether small language models (SLMs) and quantized large language models (LLMs) can maintain brain alignment—i.e., the ability to predict human brain activity during naturalistic language processing—as effectively as full-scale LLMs. Using fMRI data from participants listening to stories, the authors evaluate models across scales (1B to 8B parameters) and quantization methods (AWQ, GPTQ, SmoothQuant). They also assess linguistic competence using the FlashHolmes benchmark, which spans morphology, syntax, semantics, discourse, and reasoning.

**Strengths:**

Originality: First to jointly study model scale, compression, and brain alignment in a controlled neuroimaging setting.

Quality: Strong empirical foundation with multiple model families, quantization methods, and brain regions.

Clarity: Clear distinction between brain alignment and task performance, with nuanced interpretation of their divergence.

Significance: Offers practical recommendations for neuroAI applications, especially in low-latency or low-resource environments like BCIs.

**Weaknesses:**

1.Model scale upper limit: The largest model evaluated is 8B; extrapolating findings to 13B+ models (DeepSeek-R1-Distill-Qwen-14B, Qwen3-32B, DeepSeek-R1-Distill-Qwen-32B, etc.) remains unclear.

2.Compression scope: The motivation and experimental design of this article are not convincing. Only post-training quantization is studied; pruning, distillation, or structured compression are not explored.

3.Modality limitation: Only fMRI is used; MEG or ECoG could reveal temporal dynamics of alignment loss.

4.Decoder gap: The study focuses on encoding (brain prediction); decoding (stimulus reconstruction) is not addressed.

**Questions:**

1.Why limit the largest model to 8B? Do you expect 3B SLMs to still align with 13B+ models, or would the gap widen?

2.Any plans to test pruning or distillation? These methods may preserve structure better than quantization—how might they affect alignment?

3.Would MEG or ECoG reveal different sensitivities? fMRI is slow; could temporal resolution expose real-time alignment loss?

4.Why not evaluate decoding performance? For BCI use, reconstructing stimuli from brain data is crucial—do SLMs or quantized models degrade decoding fidelity?

5.Could you release the brain alignment scores per task/region? This would help correlate specific linguistic deficits with regional brain activity loss.

---

> ### Author Response · Authors · 2025-11-26
>
> *We thank the reviewer for their positive, insightful and valuable comments and suggestions which are crucial for further strengthening our manuscript.*
>
> **Q1. Model scale upper limit: The largest model evaluated is 8B; extrapolating findings to 13B+ models (DeepSeek-R1-Distill-Qwen-14B, etc,.)**
>
> Thank you for this question. Based on reviewer’s suggestion, we now extended our evaluation to larger models as follows:
> * Qwen2.5-14B full model
> * LLaMa-14B full model
> * Along with previous models, we now included encoding for DeepSeek model as well
>   * DeepSeek-R1-Distill-Qwen-14B, 7B and 1.5B full model
>   * DeepSeek-R1-Distill-Qwen7B, and 1.5B AWQ, GPTQ and SmoothQunat
>   * DeepSeek-R1-Distill-LLaMA-3B AWQ, GPTQ and SmoothQuant
>
> We have provided:
> * **Encoding performance of DeepSeek models in CQ5**. Kindly refer to those responses for more comprehensive information
>
> **Q3. Modality limitation: Only fMRI is used; MEG or ECoG could reveal temporal dynamics of alignment loss.**
>
> We thank the reviewer for this question. We agree with the reviewer that extending to other modalities such as MEG or ECoG will reveal  temporal dynamics of alignment loss. *We already mention this modality constraint as a limitation in the current manuscript and now make it more explicit.*
>
> * In the present study, we chose fMRI because it offers high spatial resolution and allows us to relate linguistic competence to specific brain regions at a fine-grained level. Moreover, the available fMRI dataset is large in terms of samples, making naturalistic story listening a natural starting point for systematically comparing SLMs, LLMs, and their quantized variants.
> * We also extended our analysis to a **naturalistic reading fMRI dataset**, as discussed in the **Appendix M**. From **Appendix M Fig. 29**, we observe that 3B SLMs yield brain alignment comparable to the 7B and 14B Qwen2.5 models, whereas 1.5B SLMs exhibit a clear drop in brain alignment on the ReadingBrain dataset, consistent with our main conclusions.
>
> As the reviewer suggests, future work will extend our framework to MEG and ECoG to leverage their millisecond-scale temporal resolution and track the time-resolved dynamics of alignment loss and recovery, for example, whether quantization primarily impacts early lexical stages, syntactic composition, or later discourse-level integration.
>
> **Q5. Could you release the brain alignment scores per task/region? This would help correlate specific linguistic deficits with regional brain activity loss.**
>
> Thank you for this question.
>
> Based on reviewer’s suggestion, we have now provided
> * **ROI-Specific Analysis and Best Layer Selection in CQ3**,
> * **Subject Variability at Optimal Layers in CQ4**, and
>
> Kindly refer to those responses for more comprehensive information, These CQ1, CQ2, CQ3, CQ4, CQ5 all are under common responses we answered at the starting of this page with the title "**Common Responses to Reviewers wmV2, xLFX, PMQK, and f24S**"

---

> ### Author Response · Authors · 2025-11-26
>
> **Q2. Only post-training quantization is studied; pruning, distillation, or structured compression are not explored.**
>
> Thank you for this valuable suggestion.
> * To clarify, our original focus was on post-training quantization, because it is widely used in practice, easy to deploy without re-training, and directly targets memory/throughput constraints.
> * We already mention this pruning/distillation constraint as a limitation in the current manuscript and now make it more explicit.
>
> Based on the reviewer’s suggestion, we have now included **unstructured pruning** which is equivalent to quantization and now report preliminary results in the **Appendix O**.
> * In particular, we perform unstructured magnitude pruning on the linear layers of Qwen2.5-3B and Qwen2.5-1.5B, at sparsity levels 0.1, 0.25, and 0.5.
> * **For Qwen2.5-3B**, **Table 20 (Appendix O)** summarizes brain alignment (mean ± s.e.m. across subjects) for the base, quantized variants, and pruned models, showing that AWQ and SmoothQuant slightly improve over the FP16 baseline (0.933 ± 0.035 and 0.930 ± 0.035 vs. 0.924 ± 0.033), GPTQ is modestly lower (0.910 ± 0.037), and unstructured pruning up to 50% keeps alignment in a narrow range (0.910–0.907 with s.e.m. 0.032–0.043).
>
> **Table 20**: Comparison of quantization and pruning for Qwen2.5-3B
>
> |Model variant| Method| Sparsity | Normalized Brain Alignment |
> |-|-|-|-|
> | Qwen-2.5-3B| FP16 (baseline)| 0%| 0.924 ± 0.033 |
> | Qwen-2.5-3B-AWQ| Quantization (AWQ)| 0%| 0.933 ± 0.035 |
> | Qwen-2.5-3B-GPTQ| Quantization (GPTQ)| 0%| 0.910 ± 0.037 |
> | Qwen-2.5-3B-Smooth| Quantization (SmoothQuant)| 0%| 0.930   ± 0.035 |
> | Qwen-2.5-3B-0.1| Pruning| 10%| 0.910 ± 0.032 |
> | Qwen-2.5-3B-0.25| Pruning|25%| 0.908  ± 0.033 |
> | Qwen-2.5-3B-0.5| Pruning|50% | 0.907± 0.043|
>
> * **For Qwen2.5-1.5B**, **Table 21 (Appendix O)** summarizes brain alignment (mean ± s.e.m. across subjects) for the base, quantized variants, and pruned models, showing that quantization and moderate pruning preserve brain alignment remarkably well. All quantization methods lie very close to the FP16 baseline, with overlapping 95% confidence intervals: AWQ (0.854 ± 0.031) and SmoothQuant (0.836 ± 0.025) are slightly above FP16 (0.830 ± 0.025), whereas GPTQ (0.828 ± 0.026) is essentially similar to the baseline.
> * Unstructured pruning at 10–25% sparsity also keeps alignment in the same range as FP16 (0.847 and 0.824, respectively, with overlapping CIs), indicating that modest pruning does not substantially harm brain alignment for this 1.5B model.
> * In contrast, 50% pruning leads to a large drop in alignment (0.608 ± 0.053), with a clearly non-overlapping confidence interval, suggesting that aggressive sparsification breaks the brain-relevant representations.
>
> **Table 21**: Comparison of quantization and pruning for Qwen2.5-1.5B
> | Variant| Mean ± SEM       | 95% CI           | Notes                |
> |-|-|-|-|
> | Qwen-2.5-1.5B  | 0.830 ± 0.025    | [0.781, 0.879]   | full precision       |
> | Qwen-2.5-1.5B-AWQ| 0.854 ± 0.031    | [0.793, 0.915]   | INT4 quantization    |
> | Qwen-2.5-1.5B-GPTQ| 0.828 ± 0.026    | [0.777, 0.879]   | INT4 quantization    |
> | Qwen-2.5-1.5B-SmoothQuant      | 0.836 ± 0.025    | [0.787, 0.885]   | INT4 quantization    |
> | Prune 10%  | 0.847 ± 0.026    | [0.796, 0.898]   | unstructured pruning |
> | Prune 25%        | 0.824 ± 0.025    | [0.775, 0.873]   | unstructured pruning |
> | Prune 50%        | 0.608 ± 0.053    | [0.504, 0.712]   | unstructured pruning |
>
>
> We view these pruning experiments as complementary to our main quantization results: they show that both post-training quantization and unstructured pruning can preserve brain alignment to a surprising degree, but they also highlight potential trade-offs with linguistic competence at high sparsity.
>
> **For Distillation**,
> * we used DeepSeek models which provide better comparison across all compression types.
> * We acknowledge that we still do not systematically explore structured compression across model components, where different methods could be compared under matched compression ratios to assess their differential impact on both linguistic competence and brain alignment.
> * We now explicitly note that this is an important direction for future work, but falls outside the scope of the current paper given the substantial additional experiments it would require.
>
> * Although our model suite includes the DeepSeek-R1-Distill family (which is itself a product of knowledge distillation), in this work we do not systematically study distillation as a compression method. We treat DeepSeek as an additional, pretrained model family for evaluation, and focus our controlled compression experiments on post-training quantization (and preliminary unstructured pruning).
> * A careful comparison of different distillation strategies under matched compression ratios is therefore an important direction for future work.
>
> We have added this discussion in **Appendix O** of the revised paper.

---

> > ### Author Response · Authors · 2025-11-26
> >
> > **Q4. Decoder gap: The study focuses on encoding (brain prediction); decoding (stimulus reconstruction) is not addressed.**
> >
> > We thank the reviewer for this valuable suggestion. Based on reviewer’s suggestion, we now perform brain decoding to reconstruct text stimuli from fMRI brain activity using LLaMA-3-8B, and two SLMS (LLaMA-3-3B and LLaMA-3-1B).
> >
> > * Inspired by **BrainLLM (Ye et al. 2025)**, we perform end-to-end text stimulus reconstruction from fMRI brain activity.
> >    * We follow the same BrainLLM methodology, where we use the same Moth Radio Hour dataset (11 stories) with the same train/test split, where ten stories are used for training and one held-out story is used for generation.
> >    * Concretely, we train a brain-to-text decoder and report standard text-generation metrics-BLEU-1, WER, METEOR, and BERT-F1-for three models: LLaMA-3-8B, LLaMA-3.2-3B, and LLaMA-3.2-1B (**Table 18 in Appendix M**).
> >    * Across reconstructed segments per model on test dataset, LLaMA-3.2-3B achieves the best performance on all four metrics (BLEU-1 = 0.120, WER = 4.22, METEOR = 0.110, BERT-F1 = 0.825), slightly outperforming LLaMA-3.2-8B and clearly improving over the LLaMA-3.2-1B baseline (BLEU-1 = 0.070, METEOR = 0.055, BERT-F1 = 0.811).
> >    * These BERT-F1 scores in the 0.81–0.83 range indicate that the decoded text reliably preserves the semantic content of the original stimulus, while BLEU-1 in the 0.07–0.12 range is in line with prior work where exact word-level recovery from fMRI is known to be challenging.
> >
> > **Table 18**. Overall Performance Metrics brain-to-text decoding.
> >
> > | Model | BLEU-1| WER| METEOR | BERT-F1 | Samples |
> > |-|-:|-:|-:|-:|-:|
> > |LLaMA-3-8B|0.0699  | 5.7839 | 0.0550 | 0.8108|784|
> > |LLaMA-3.2-3B| 0.1198  | 4.2237 | 0.1101 | 0.8252|784|
> > |LLaMA-3.2-1B| 0.1105  | 4.4869 | 0.0990 | 0.8237|784|
> >
> > * To make the reconstruction quality more interpretable, we also include qualitative examples comparing ground-truth text and decoded outputs (**Table. 19 in Appendix M**).
> >   * These examples illustrate that the decoder often recovers the overall meaning, emotional tone, and discourse context, even when individual words differ-e.g., reconstructions that correctly express embarrassment, uncertainty, or interactions with children, despite not matching every token verbatim.
> >
> > **Table 19**. Example fMRI-based stimulus reconstructions for LLaMA-3-8B, LLaMA-3.2-3B, and
> > LLaMA-3.2-1B
> >
> > | Example|Variant | Text| Metrics|
> > |-|-|-|-|
> > | 1      | Ground Truth | *and that i had to hold onto him*                                                                                                                                                                   | --  |
> > | 1      | LLaMA-3-8B| i was going to have to take him to the hospital but i had no idea what to do with him so i went to a friend of mine and he told | BLEU-1 = 0.3750, WER = 3.5000, METEOR = 0.2885   |
> > | 1       | LLaMA-3.2-3B| , she's crying and then she sees me and she com, she's crying and then she sees me and she com, she's crying and then she sees| BLEU-1 = 0.1852, WER = 3.2500, METEOR = 0.0505   |
> > | 1       | LLaMA-3.2-1B       | could see her face and she was smiling at me and then she said “hi” and i said “hi” back to her and then she turned around and | BLEU-1 = 0.2500, WER = 3.1250, METEOR = 0.1500   |
> > | 2       | Ground Truth       | *to be a teacher with all these kids*                                                                                                                                                               | --  |
> > | 2       | LLaMA-3-8B| for them to see their children i had no idea how hard it was for them to see their children i had no idea how hard it was for them to see  | BLEU-1 = 0.0938, WER = 3.8750, METEOR = 0.0962   |
> > | 2       | LLaMA-3.2-3B| is a great place to live, work, and play. It's also home to some of the best restaurants, bars, and coffee shops in the city. | BLEU-1 = 0.1200, WER = 3.0000, METEOR = 0.2635   |
> > | 2       | LLaMA-3.2-1B | and i said you know like and i said you know like and i said you know like and i said you know like and i said you know like and i| BLEU-1 = 0.0000, WER = 4.0000, METEOR = 0.0000   |
> >
> > *[Ye et al. 2025] Generative Language Reconstruction from Brain Recordings, Nature Communication Biology 2025*
> >
> > * Overall, these new brain decoding experiments show that our SLMs are not only good encoders of brain activity, but also support meaningful decoding: they can reconstruct linguistically coherent text from fMRI with high semantic fidelity and reasonable word-level accuracy.
> > * We emphasize that decoding is not the primary focus of the present work, but the added section demonstrates that the brain-aligned representations we study are indeed rich enough to support stimulus reconstruction, directly addressing the reviewer’s concern.
> >
> >
> > We have updated this information in **Appendix M** of the revised draft.

---

### Official Review · Reviewer_xLFX · 2025-11-01

**Soundness:** 2
**Presentation:** 3
**Contribution:** 2
**Rating:** 2
**Confidence:** 3

**Summary:**

This paper investigates how model scale and post-training quantization influence the alignment between language model representations and human brain activity. Using fMRI data collected during naturalistic story listening, and embeddings extracted from both large and small language models (LLaMA-3.2 and Qwen-2.5 families), the authors train voxel-wise encoding models to predict neural responses. They further evaluate each model’s linguistic abilities across five domains (discourse, syntax, semantics, morphology, reasoning) using the FlashHolmes benchmark, enabling an analysis of which linguistic properties best support brain alignment.

The results indicate that: (1) models around 3B parameters achieve comparable brain alignment to 7B–8B models, suggesting diminishing returns beyond moderate scale; (2) quantization methods differ substantially, AWQ and SmoothQuant largely preserve alignment, whereas GPTQ causes noticeable degradation; and (3) discourse-level abilities show the strongest correlation with brain predictivity. Overall, the study highlights the potential of efficient small or quantized models for NeuroAI applications, while emphasizing that not all compression methods are equally compatible with brain alignment.

**Strengths:**

Timely and relevant topic. The paper addresses an important and underexplored question: how scaling and compression strategies affect the brain-alignment properties of language models. This is of both scientific and practical interest for NeuroAI research.

Comprehensive evaluation design. By combining voxel-wise encoding models with a linguistic benchmark (FlashHolmes), the study bridges neural and computational levels of analysis and identifies which linguistic properties support brain alignment.

Systematic comparison of quantization methods. Evaluating AWQ, GPTQ, and SmoothQuant across multiple model sizes is methodologically rigorous and provides clear guidance for selecting compression techniques in brain-related applications.

**Weaknesses:**

**Lack of statistical validation.**
Figure 2 compares alignment across quantization methods for the same model (e.g., Qwen), but no statistical tests, confidence intervals, or measures of variability across runs or participants are reported. Consequently, it is unclear whether observed differences, such as the apparent improvement after AWQ compression, are meaningful or fall within noise. Without proper significance testing, it is difficult to interpret whether certain compression methods reliably affect alignment.

**Narrow model range and limited scaling claims.**
The study only examines models up to 8B parameters, whereas prior scaling-law studies extend to 72B [1]. The claim that alignment saturates around 3B is therefore tentative, as larger models might still yield improvements.

Reference
[1] Richard Antonello, Aditya Vaidya, and Alexander Huth. Scaling laws for language encoding models in fmri. Advances in Neural Information Processing Systems, 36, 2024.

**Placement and discussion of limitations.**
The discussion of limitations is relegated to the appendix, which weakens the main narrative. Given that all tested models are relatively small, the study might not capture true scaling behavior. The normalization by noise ceiling can make small-model results look stronger than they are, this point deserves more transparent discussion in the main text.

**Statistical reporting and subject variability.**
The paper lacks an analysis of subject variability, it is unclear whether the results in Figure 2 and related analyses hold consistently across participants. Reporting variance or confidence intervals across subjects would greatly strengthen the reliability of the findings.

**Limited interpretability discussion.**
The relationship between degraded linguistic properties (e.g., discourse, syntax) and specific brain regions is underdeveloped. A more detailed interpretation could link linguistic theory to neural substrates.

**Overly verbose writing.**
Some procedural details (e.g., preprocessing, HRF estimation) could be moved to the appendix, allowing the main text to focus on results and their interpretation.

**Questions:**

Clarity of figures.
Figure 2 should clearly label “1.5B.” Figure 3 is difficult to interpret due to unclear color scales and range indicators. Similar issues appear in the flatmap visualizations in the appendix (such as Figure 6-7, 11-20).

Layer selection.
Which model layers were used for brain alignment? Did you observe consistent optimal layers across models and quantization methods?

Compression degree.
What are the actual compression ratios achieved by each method for different model sizes? How do AWQ, GPTQ, and SmoothQuant compare in terms of model size reduction and their impact on brain alignment?

Statistical testing.
Did you conduct permutation or bootstrap tests to assess whether alignment differences between quantization methods are significant? Visual trends alone are insufficient to judge reliability.

Subject-level robustness.
Were results consistent across individual participants? Showing variance or confidence intervals across subjects would strengthen the argument.

---

> ### Author Response · Authors · 2025-11-26
>
> *We thank the reviewer for their positive, insightful and valuable comments and suggestions which are crucial for further strengthening our manuscript.*
>
> **Q1. Lack of statistical validation.**
>
> Thank you for this question.
> * Based on the reviewer’s suggestion, we performed paired t-tests to compare normalized brain alignment between quantization methods and their respective base models across all language ROI-layer combinations and subjects (n=9).
> * To control for multiple comparisons, we applied False Discovery Rate (FDR) correction using the Benjamini-Hochberg procedure (α = 0.05). Effect sizes were calculated using Cohen's d. We report both raw p-values and FDR-corrected significance, along with 95% confidence intervals for mean differences.
>
> Based on reviewer’s suggestion, we have now provided
>
>   * **Statistical Validation of Quantization Effects in CQ2**,
>   * **ROI-Specific Analysis and Best Layer Selection in CQ3**,
>   * **Subject Variability at Optimal Layers in CQ4**
>
> Kindly refer to those responses for more comprehensive information
>
> **Q2. Narrow model range and limited scaling claims.**
>
> Thank you for raising this point.
> * To clarify, our primary motivation for evaluating brain alignment in recent smaller language models is interpretability. While prior work on Scaling laws has examined larger models (e.g., upto 72B models) and shown higher degree of brain alignment, these studies have not examined why scale improves alignment or which representational properties are responsible.
>
> * In NeuroAI, the primary objective is to understand how language is processed in the brain and in language models, and how these systems can be meaningfully compared.
>   * While larger models often yield better task performance and brain alignment, they are also increasingly difficult to analyze mechanistically.
>   * Instead, examining which linguistic properties are maintained or lost across small, large, and compressed models allows us to identify the minimal representational key components or properties necessary for high brain alignment.
>
> Based on reviewer’s suggestion,
> * we have now extended our evaluation to larger models (up to 14B size) and included an additional model family: Deepseek-R1-Distill at 1.5B, 3B, 7B and 14B.
> * Across Qwen, LLaMA, and DeepSeek, we observe the same trend: 3B SLMs maintain brain alignment comparable to 14B models, while 1B-1.5B models consistently drop in brain alignment.
> * Fig. 26 in Appendix L shows voxelwise encoding performance of DeepSeek family of language models at different scales.
>   * For both 7b and SLMs (3b & 1.5b), we also apply post-training quantization and measure normalized brain predictivity at both the whole-brain level and within language-specific regions (IFG).
>   * Notably, the DeepSeek-R1-Distill 14B model shows only a modest improvement over the 7B version, and its alignment is matched by the 3B DeepSeek model, suggesting that 3B SLMs provide sufficient representational capacity for studying brain-LM alignment within this scale regime.
>
> We have now provided
>
>   * **Encoding performance of DeepSeek models in CQ5**,
>
> Kindly refer to those responses for more comprehensive information
>
> **Q4. Statistical reporting and subject variability.**
>
> Thank you for this valuable suggestion.
> * We now perform subject variability for each model at best optimal layer.
> * Based on reviewer’s suggestion, we have now provided
>   * **Subject Variability at Optimal Layers in CQ4**
>
> **Q6. Overly verbose writing. Some procedural details (e.g., preprocessing, HRF estimation) could be moved to the appendix**
>
> Thank you for this valuable suggestion. We have now moved preprocessing and HRF estimation to **Appendix A**.
>
> **Q7. Clarity of figures. Figure 2 should clearly label “1.5B.” Figure 3 is difficult to interpret due to unclear color scales and range indicators. Similar issues appear in the flatmap visualizations in the appendix (such as Figure 6-7, 11-20)**.
>
> Thank you for providing a minor correction.
> * We have revised the figures to improve clarity: Figure 2. We have corrected the label for the “Qwen2.5-1.5” to “Qwen2.5-1.5B” and increased the font size of all model labels to make parameter differences more immediately readable.
> * For Figure 3, we have increased the font size for axes, labels, and legends, and we now use a clearer color scale with explicit numeric tick marks and range indicators.
>     * The colorbar is labeled with the exact quantity being plotted (normalized predictivity), and we adjusted the range to avoid saturating large regions of the plot.
>
> These CQ1, CQ2, CQ3, CQ4, CQ5 all are under common responses we answered at the starting of this page with the title "**Common Responses to Reviewers wmV2, xLFX, PMQK, and f24S**"

---

> > ### Author Response · Authors · 2025-11-26
> >
> > **Q3. Placement and discussion of limitations.**
> >
> > Thank you for raising this point. Based on the reviewer’s suggestion, we have moved the key limitations from the appendix into the main Discussion section in the revised pdf.
> >
> > * We agree with the reviewer that our work does not capture “true” large-scale scaling behavior in the sense of prior work that probes 30B–175B models.
> >     * We also do not claim to oppose or overturn existing scaling-law findings in brain encoding.  Instead, our study primarily focuses on a different scale regime (1B–14B) and asks a different question: **within the small-to-medium range of large language models (1B–14B), how does representational capacity relate to brain predictivity?**
> >     * This range is scientifically relevant because recent small language models achieve task accuracy comparable to their larger counterparts, making it important to understand whether they also preserve brain-relevant representations.
> > * Therefore, our goal is therefore to provide a complementary, local view to enable more mechanistic interpretation of models rather than to characterize asymptotic scaling behavior at very large model sizes.
> >
> > **Cross-subject predictivity and normalized brain alignment:**
> > * The cross-subject predictivity is estimated by subsampling fMRI datasets from 9 participants, and using a voxel-wise encoding model to predict one participant’s response using other participants’ brain data, as established in prior work [Schrimpf et al., 2021; Oota et al., 2024; Alkhamissi et al., 2024; Oota et al., 2025]. Note that the estimated cross-subject prediction accuracy is based on the assumption of a perfect model.
> > * $$
> > \mathrm{NormalizedBrainAlignment}_v = \frac{\mathrm{ModelPredictivity}_v}{\mathrm{CrossSubjectPredictivity}_v}
> > $$
> > * We use the same cross-subject normalization setting as in Oota et al. (2024). In that work, the normalized brain alignment for models such as BERT, GPT-2, and FLAN-T5 typically reaches 65–70%.
> > * In our study, even small language models (SLMs) achieve normalized brain alignment values above 80%, indicating that they explain a larger fraction of the explainable neural variance under the same normalization method.
> >     * Importantly, normalization does not change the ranking of models: if a 1B model underperforms a 3B or 7B model in raw predictivity, it will also underperform after dividing by the same noise ceiling.
> >     * Thus, we do not see a regime where 1B–1.5B models “catch up” to larger models after normalization. Instead, we find a capacity threshold around ~3B: models smaller than this clearly underperform, whereas 3B models (which are still small compared to 30B–175B LMs) reach brain alignment comparable to 7B–14B while being much cheaper to run.
> >
> > **Q5. Limited interpretability discussion.**
> >
> > We thank the reviewer for this insightful comment.
> > * To clarify, our current study provides the relationship between degraded linguistic properties (e.g., discourse, syntax) and specific brain regions through an Indirect approach: non-causal methodology is standard in the literature:
> >    * prior work often first estimates brain alignment and then independently evaluates linguistic task performance to study correlations between the two (e.g., Schrimpf et al., 2021; Goldstein et al., 2022). Our analysis follows this established approach.
> >
> > * Our main motivation in this work is to ask whether SLMs maintain similar brain alignment as LLMs, and how post-training quantization affects both brain alignment and linguistic competence. We therefore measure
> >    * brain alignment before and after compression for both SLMs and LLMs, and
> >    * their performance on a broad set of linguistic tasks, and then correlate these measures in an indirect way rather than performing causal perturbations on specific linguistic properties.
> >
> > A more fine-grained interpretability analysis has been pursued in prior work.
> > * For example, Oota et al. (2023) use controlled perturbations on 7 probing tasks to study which linguistic properties (e.g., syntactic structure, lexical semantics) most strongly affect brain alignment.
> > * As per reviewer’s suggestion, we now explicitly acknowledge that performing similar perturbation-based analyses is an important direction for future work.
> >    * In particular, we propose combining our compression/scaling framework with richer behavioral/linguistic benchmarks such as the FlashHolme benchmark, which spans ~200 diverse tasks and offers a detailed, interpretable profile of model capabilities.
> >    * Using such a benchmark as a proxy for specific linguistic properties, future work could relate property-specific performance (e.g., on discourse, syntax, pragmatics tasks) to region-wise brain alignment, providing exactly the type of theory-grounded mapping between linguistic theory and neural substrates.
> >
> > *Oota et al. (2023), Joint processing of linguistic properties in brains and language models, NeurIPS-2023*

---

> > > ### Author Response · Authors · 2025-11-26
> > >
> > > **Q8. Layer selection. Which model layers were used for brain alignment? Did you observe consistent optimal layers across models and quantization methods?**
> > >
> > > Thank you for this question.
> > > * To clarify, in our analyses, we extract activations from every transformer layer, and fit a separate voxel-wise encoding model for each layer.
> > >   * For each model, we then compute brain alignment layer-by-layer across the language ROIs and identify the best layer as the one with the highest mean normalized predictivity. The main size/quantization comparisons are reported using this model-specific best layer (**see Tables 13 and 16 in the Appendix K**).
> > >
> > > We have now provided
> > >
> > >   * **ROI-Specific Analysis and Best Layer Selection in CQ3**
> > >
> > > * From **Table 13 in Appendix K**, across language ROIs we find that the best layers are highly consistent within a given model: the same (or adjacent) layer tends to be optimal across ROIs, so we treat the best layer as a model-level property when summarizing results.
> > > * Overall, across models we observe the familiar pattern that middle-to-late layers yield the strongest brain alignment, with early layers performing clearly worse.
> > >
> > > **Regarding quantization**,
> > > * We also examined whether AWQ, GPTQ, or SmoothQuant systematically shift the optimal layer.
> > > * We did not observe any systematic change: for a given architecture, the best layer under quantization is typically the same as in FP16 or within ±1–2 layers in the same middle/late portion of the network.
> > > * In other words, quantization affects the magnitude of brain alignment (as analyzed in **Tables 9 and 11 in Appendix K**), but not the qualitative position of the brain-optimal layers.
> > > * We now clarify this procedure and these observations in the main text and refer explicitly to the layer-wise summaries in **Tables 13 and 16**.
> > >
> > > **Q9. Compression degree. What are the actual compression ratios achieved by each method for different model sizes? How do AWQ, GPTQ, and SmoothQuant compare in terms of model size reduction and their impact on brain alignment?**
> > >
> > > Thank you for this question.
> > > * To clarify, we already report the actual compression ratios in **Table 1 (main paper)**, which lists, for each model family and size, the full-precision (FP16) model size (GB) and the sizes after AWQ, GPTQ, and SmoothQuant quantization.
> > > * For example, for Qwen2.5-7B,
> > >   * the FP16 model is 15.1 GB, whereas AWQ and GPTQ reduce this to 5.57 GB and 5.58 GB respectively (≈2.7× smaller), and SmoothQuant to 8.67 GB (≈1.7× smaller).
> > >
> > > From Table 1, we make following observations: Across both Qwen2.5 and LLaMA-3.2, we observe a consistent pattern:
> > >
> > > * **For SLMs**,
> > >    * AWQ typically compresses FP16 to 44–63% of its original size (≈1.6–2.3× smaller),
> > >    * GPTQ to 34–41% (≈2.4–3.0× smaller), and
> > >    * SmoothQuant to 66–82% (≈1.2–1.5× smaller).
> > >
> > > * **For LLMs**,
> > >    * AWQ and GPTQ compress FP16 models to about 35–37% of their original size (≈2.7–2.9× smaller),
> > >    * while SmoothQuant reduces them to 56–57% (≈1.7–1.8× smaller).
> > >
> > > * Thus, GPTQ achieves the strongest size reduction, AWQ is slightly less aggressive, and SmoothQuant is the most conservative in terms of memory savings.
> > > * When combined with our brain-alignment results (**Tables 8–11**), this reveals a clear trade-off: AWQ and SmoothQuant generally preserve or slightly improve brain alignment relative to FP16 at 1.5–2.5× compression, whereas GPTQ, despite achieving ~3× compression, tends to show a modest but reliable drop in brain alignment (especially for Qwen and LLaMA).
> > > * We now make this cost-alignment trade-off explicit in the main text, highlighting that among quantization methods at similar parameter counts, AWQ and SmoothQuant offer a more favorable compression–alignment balance than GPTQ.

---

> > > > ### Author Response · Authors · 2025-11-26
> > > >
> > > > **Q10. Statistical testing. Did you conduct permutation or bootstrap tests to assess whether alignment differences between quantization methods are significant?**
> > > >
> > > > Thank you for this question.
> > > > * Yes, we perform a block permutation test (**already reported in current version Appendix D**) as follows:
> > > >   * To determine if normalized predictivity scores are significantly higher than chance, we run a permutation test using blocks of 10 contiguous fMRI TRs (considering the slowness of hemodynamic response) rather than individual TRs.
> > > >   * By permuting predictions 5000 times, we create an empirical distribution for chance performance, from which we estimate p-value of the actual performance. The choice of these specific permutation test configurations is based on established methodologies in previous research (Deniz et al., 2019; Reddy & Wehbe, 2021; Oota et al., 2024a).
> > > >   * This procedure is used to confirm that all reported model-brain alignment scores are significantly above chance.
> > > >
> > > > **Differences between quantization methods (paired tests across subjects/metrics)**.
> > > >
> > > > Based on reviewer’s suggestion, we have now provided
> > > > * **Quantitative significance tests in CQ1**
> > > >
> > > > Together, the permutation tests ensure that our alignment scores are reliably above chance, and the paired $t$-tests establish which differences between quantization methods are statistically significant.
> > > >
> > > > **Q11. Subject-level robustness. Were results consistent across individual participants? Showing variance or confidence intervals across subjects would strengthen the argument.**
> > > >
> > > > We thank the reviewer for this valuable suggestion. We now report the subject's variability across models.
> > > >
> > > > We have now provided
> > > >
> > > >   * **Subject Variability at Optimal Layers in CQ4**
> > > >
> > > > * Subject variability at optimal layers was low across all models (CV < 11.5%), with 3B showing the highest consistency (CV = 10.05%) and 14B achieving the highest mean alignment (0.8411±0.088)."

---

### Official Review · Reviewer_wmV2 · 2025-11-01

**Soundness:** 2
**Presentation:** 3
**Contribution:** 2
**Rating:** 4
**Confidence:** 3

**Summary:**

This paper investigates whether small or quantized language models can achieve brain alignment comparable to large full-precision LLMs. Using fMRI data from participants listening to naturalistic stories, the authors evaluate models such as LLaMA-3 and Qwen-2.5 and their quantized variants Brain alignment is measured via voxel-wise encoding correlations, and linguistic competence is assessed using the FlashHolmes benchmark covering morphology, syntax, semantics, discourse, and reasoning. The authors report that 3B models perform comparably to 7B–8B ones, while GPTQ quantization harms both linguistic competence and brain alignment.

**Strengths:**

1.	The study addresses an emerging question about model efficiency and neural alignment, offering a systematic comparison between model size, quantization, and brain predictivity. The experimental framework—combining fMRI encoding and linguistic probing—is technically sound and clearly presented.
2.	The paper is well written and carefully structured, with clear visualizations and comprehensive references to recent brain–language modeling literature.
3.	The attempt to connect linguistic task performance with neural alignment is conceptually interesting and could, in principle, guide future investigations of how language models capture brain-relevant information.

**Weaknesses:**

1.	The overall motivation and significance are limited. While scaling and efficiency are relevant engineering questions, it is unclear why quantized models are meaningful for cognitive neuroscience. Brain–language alignment research is primarily driven by scientific, not deployment, objectives, so the practical incentive for compressing models in this context is weak. As presented, the work is more an engineering benchmark than a scientific contribution.
2.	The dataset and experimental scope are narrow. The Moth Radio Hour fMRI corpus includes only nine subjects and a single passive listening task. Such limited data cannot support broad conclusions about how model scale or quantization affects neural language representations. Results on one dataset may not generalize to other linguistic or cognitive paradigms.
3.	The link between linguistic competence and brain alignment is tenuous. The five FlashHolmes task categories are only a partial proxy for linguistic processing in the brain, and correlations between these scores and voxel alignment are descriptive rather than causal.
4.	The main conclusion—that 3B models can match or surpass 7B–8B models, challenging previous “scaling laws” in brain encoding—is not well supported. The tested model range is small (maximum 7B), and the data are noisy and task-limited. Prior literature consistently finds positive scaling effects; overturning that trend would require more comprehensive evidence and statistical testing.
5.	Minor issues: Figure 2’s legend mislabels Qwen where it should read LLaMA.

**Questions:**

1.	There is a growing body of work on visual brain alignment showing that factors such as model scale, training data diversity, and dataset domain strongly affect alignment quality. Why does this paper not analyze similar factors for language models, beyond simple parameter count and quantization?
2.	Current language modeling research has diversified far beyond transformers, including diffusion-based, state-space (e.g., Mamba), and other architectures. How do you expect these non-transformer paradigms to influence brain alignment? Do you believe the transformer mechanism itself is critical for the observed effects, or could similar patterns emerge with alternative architectures?

---

> ### Author Response · Authors · 2025-11-25
>
> *We thank the reviewer for their positive, insightful and valuable comments and suggestions which are crucial for further strengthening our manuscript.*
>
> **Q1. The overall motivation and significance are limited. It is unclear why quantized models are meaningful for cognitive neuroscience.**
>
> Thank you for raising this concern.
>
> * To clarify, our primary motivation for evaluating brain alignment in recent smaller language models is interpretability, not deployment. While prior work on Scaling laws has examined larger models (e.g., up to 72B models) and shown higher degree of brain alignment, these studies have not examined why scale improves alignment or which representational properties are responsible.
>
> * In NeuroAI, the primary objective is to understand how language is processed in the brain and in language models, and how these systems can be meaningfully compared. While larger models often yield better task performance and brain alignment, they are also increasingly difficult to analyze mechanistically.
>    * Instead, examining which linguistic properties are maintained or lost across small, large, and compressed models allows us to identify the minimal representational key components or properties necessary for high brain alignment.
>
> * Therefore, our central motivation is foundationally scientific: to understand how internal representations change under constraints on model size and precision, whether smaller language models retain the linguistic features that drive higher brain alignment.
>    * Additionally, quantization is not included for deployment reasons alone. Quantization and scale manipulations provide controlled perturbations of model representational geometry, allowing us to test what aspects of linguistic structure are essential for brain alignment.
>
> **Some of our key findings as follows:**
> * Across two language families, 3B SLMs preserve brain alignment comparable to 7B LLMs, while 1B-1.5B models consistently show lower brain alignment, especially in semantic-selective regions such as IFG, AG, PCC, and dmPFC. This identifies a previously unknown capacity threshold for brain-relevant linguistic computation.
> * Our quantization experiments revealed that GPTQ consistently reduces alignment, while AWQ and SmoothQuant preserve ∼98% of alignment. This finding reveals that which representational subspaces are robust under perturbation, a key scientific question about the brain–LLM correspondence.
> * 1B SLMs maintain all linguistic properties performance on FlashHolmes but show significant drop  in brain alignment. Conversely, AWQ improves discourse representations on 3B SLMs while preserving brain alignment. This is an important scientific finding, indicating it is a fundamentally cognitive result showing that task performance is not a reliable proxy for brain similarity.
>
> **Q4. The tested model range is small (maximum 7B) and overturning that trend would require more comprehensive evidence and statistical testing.**
>
> Thank you for raising this point.
>
> * To clarify, we do not claim to oppose or overturn existing scaling-law findings in brain encoding. Prior work (Antonello et al. 2024, AlKhamissi et al. (2025) evaluates models up to 33B-175B parameters models and finds clear positive scaling effects.
> * Our study does not contradict these results.
>      * Instead, we focus on a different scale regime (1B - 8B) and ask a different question: within the small-to-medium range language models (1B-8B), how does representational capacity relate to neural predictivity?
>      * This range is scientifically relevant because recent small language models achieve task accuracy comparable to their larger counterparts, making it important to understand whether they also preserve brain-relevant representations.
>
> * Within this local scale window, we observe that 3B models from two independent families (Qwen2.5 and LLaMA-3.2) match the neural predictivity of their 7B-8B counterparts (Fig. 2), while 1B-1.5B models show a significant drop in brain alignment. This suggests a capacity threshold around ~3B model, not a reversal of global scaling trends.
>
> * Based on reviewer’s suggestion, we have now provided
>     * **Quantitative significance tests in CQ1**,
>     * **Statistical Validation of Quantization Effects in CQ2**,
>     * **ROI-Specific Analysis and Best Layer Selection in CQ3**,
>     * **Subject Variability at Optimal Layers in CQ4**, and
>     * **Encoding performance of DeepSeek models in CQ5**
>
> Kindly refer to those responses for more comprehensive information, these CQ1, CQ2, CQ3, CQ4, CQ5 all are under common responses we answered at the starting of this page with the title "**Common Responses to Reviewers wmV2, xLFX, PMQK, and f24S**"

---

> > ### Author Response · Authors · 2025-11-26
> >
> > **Q2. The dataset and experimental scope are narrow. Results on one dataset may not generalize to other linguistic or cognitive paradigms.**
> >
> > We thank the reviewer for this suggestion.
> > * The fMRI dataset we use is a well-known public dataset that has previously been used in many publications in both ML and Neuroscience venues (ML: [Jain & Huth 2018 NeurIPS, Jain et al. 2020 NeurIPS, Antonello et al. 2021 NeurIPS, Lamarre et al. 2022 EMNLP, Vaidya et al. 2022 ICML; Oota et al. 2024 ACL], Neuroscience: [Huth et al. 2016 Nature, Huth et al. 2017 Journal of Neuroscience, Deniz et al. 2019 Journal of Neuroscience]).
> > * For the kind of analyses that we do and that are common in this area of research, the number of samples per participant is more important than the number of subjects because the predictive models are trained independently for each participant. So, having more samples per participant helps us learn a better predictive model.
> > * This dataset is one of the largest datasets in terms of samples per participant (~4000 samples), which is the main reason for its frequent use.
> > * Our results also clearly show that this dataset is sufficient to learn a good predictive model, as we show that we can predict up to 85-90% of the explainable variance for held-out brain recordings that were not used for training (e.g., Fig. 2).
> >
> > Based on the reviewer’s suggestion,
> > * we have now extended our experiments to an additional dataset i.e. we performed voxelwise encoding on the Moth Radio Hour Reading fMRI dataset, which contains the same nine subjects and large number of samples under a different linguistic paradigm (i.e. reading).
> > * This additional evaluation allows us to assess the generalizability of our findings across datasets and tasks. We use Qwen models (Qwen2.5-1.5b, Qwen2.5-3b, Qwen2.5-7b and Qwen2.5-14b) to validate the brain alignment to examine whether 3b SLMs maintain similar brain alignment to larger versions of the models.
> > * From Appendix L Fig. 29, we observe that 3B SLMs yield brain alignment comparable to the 7B and 14B Qwen2.5 models, whereas 1.5B SLMs exhibit a clear drop in brain alignment on the Reading Brain dataset.
> >
> > *[Huth et al. 2016], Natural speech reveals the semantic maps that tile human cerebral cortex. In Nature Neuroscience, 2016*
> >
> > *[De Heer et al. 2017], The hierarchical cortical organization of human speech processing, Journal of Neuroscience, 2017*
> >
> > *[Jain & Huth, 2018], Incorporating context into language encoding models for fmri. In NIPS-2018*
> >
> > *[Deniz et al. 2019], The representation of semantic information across human cerebral cortex during listening versus reading is invariant to stimulus modality. Journal of Neuroscience, 2019*
> >
> > *[Jain et al. 2020], Interpretable multi-timescale models for predicting fMRI responses to continuous natural speech, In NeurIPS-2020*
> >
> > *[Antonello et al. 2021], Low-Dimensional Structure in the Space of Language Representations is Reflected in Brain Responses, In NeurIPS-2021*
> >
> > *[Lamarre et al. 2022] Attention weights accurately predict language representations in the brain, EMNLP-2022*
> >
> > *[Vaidya et al. 2022] Self-supervised models of audio effectively explain human cortical responses to speech, ICML-2022*
> >
> > **Q5. Minor issues: Figure 2’s legend mislabels Qwen where it should read LLaMA.**
> >
> > Thank you for raising this typo. Now, we have corrected the legend in the revised version (See Fig.2).

---

> > > ### Author Response · Authors · 2025-11-26
> > >
> > > **Q3. The link between linguistic competence and brain alignment is tenuous**
> > >
> > > Thank you for this valuable point.
> > > * We agree with the reviewer that linguistic task performance is not a complete proxy for language processing in the brain, and we do not claim a causal relationship.
> > >    * This indirect, non-causal methodology is standard in the literature: prior work often first estimates brain alignment and then independently evaluates linguistic task performance to study correlations between the two (e.g., *Schrimpf et al., 2021*; *Goldstein et al., 2022*). Our analysis follows this established approach.
> > >
> > > * Our analysis is explicitly descriptive and comparative, aimed at understanding how different linguistic properties are affected by model scale and quantization, and whether these changes correlate with differences in brain alignment.
> > >
> > > * Our goal is not to infer causality but to assess representational robustness: whether linguistic information maintained (or lost) under controlled perturbations, such as reducing model size or applying quantization, corresponds to changes in brain predictivity.
> > >    * Using the five FlashHolmes categories allows us to understand a set of linguistic phenomena (morphology, syntax, semantics, discourse, reasoning) and to examine which dimensions of linguistic structure impacts first as representational capacity is altered.
> > >
> > > **Key findings:**
> > > * *1B SLMs maintain FlashHolmes performance yet show significant drop in brain alignment*, demonstrating that behavioral competence alone does not ensure brain-like representations.
> > > * *GPTQ selectively degrades discourse, syntax, and reasoning*, and these degradations co-occur with reduced brain alignment, suggesting that certain higher-order linguistic properties are more tightly coupled to brain predictivity.
> > > * *AWQ preserves ∼98% of alignment while maintaining or improving linguistic competence*, indicating that some representational subspaces are robust even under substantial compression.
> > >
> > > * These structured dissociations show that the relationship between linguistic competence and brain alignment is meaningful and informative, even without causal claims. Our analysis therefore reveals how specific linguistic properties co-vary with brain alignment under controlled representational perturbations, providing insight into the mechanisms linking language models and the brain.
> > >
> > > *Schrimpf et al. 2021, The neural architecture of language: Integrative modeling converges on predictive processing. PNAS-2021*
> > >
> > > *Goldstein et al. 2022, Shared computational principles for language processing in humans and deep language models. Nature Neuroscience 2022*
> > >
> > > **Q6. visual brain alignment factors: why does this paper not analyze similar factors for language models?**
> > >
> > > We thank the reviewer for this important question. We agree with the reviewer that factors such as model scale, training data diversity, and dataset domain have been shown to strongly affect brain alignment. In the revision, we clarify both the scope of our work and how we control variation within model families.
> > >
> > > * **Scope of the present study**:
> > >     * Our goal was to isolate the effect of efficiency-oriented compression rather than re-characterize broad LM scaling laws. Prior work in the language domain has already extensively documented that larger models and more diverse training corpora improve brain alignment.
> > >    * Our contribution is orthogonal: we specifically ask whether small models and post-training quantization preserve brain-relevant representations.
> > >    * Our work aim to fill a gap that training-data–focused studies do not address: whether compressed SLMs retain brain-relevant geometry well enough to be useful for neuroscience?
> > > * **Controlled variation within families**:
> > >    * SLMs and LLMs in our study are architecturally matched, reducing confounds introduced by heterogeneous pretraining. We deliberately selected model families (LLaMA-3.2, Qwen-2.5, and, in extended analyses, DeepSeek) where the 1B–14B variants share training data distributions and tokenizer vocabularies.
> > >    * This allows a controlled test of parameter count and quantization. We included model scaling experiments in our answer above.
> > >
> > > * A fully systematic analysis of training data diversity and domain, analogous to recent visual work, would ideally require either (i) retraining the same architecture on carefully varied corpora, or (ii) a curated suite of models where data composition is the only factor that changes.
> > > * In current open LLM practice, these conditions are rarely met: most available models differ simultaneously in architecture, filtering, and fine-tuning, and often do not fully disclose their exact training sets. For this reason, we chose to focus on scale + compression within families where pretraining is as consistent as possible.
> > >
> > > We now highlight this point in the limitations section and note that a systematic dissection of training-data effects is an important direction for future work.

---

> > > > ### Author Response · Authors · 2025-11-26
> > > >
> > > > **Q7. How do you expect non-transformer paradigms to influence brain alignment? Do you believe the transformer mechanism itself is critical for the observed effects, or could similar patterns emerge with alternative architectures?**
> > > >
> > > > We thank the reviewer for this interesting question.
> > > > * To clarify, the current brain-LLM work has focused primarily on autoregressive Transformer-based language models, because they process:
> > > >     * (i) strong next-token / sequence modeling performance,
> > > >     * (ii) the ability to represent long-range, hierarchical dependencies, and
> > > >     * (iii) rich intermediate representations that encode semantics, syntax, and discourse structure.
> > > > * Since Transformer language models offer successful implementation of these properties (via attention), the neuro-AI community has so far mostly used them to study brain language processing.
> > > >
> > > > * Diffusion-based LLMs are beginning to appear mainly in brain decoding settings, where stimulus reconstruction from brain activity benefits from access to global context and flexible sampling.
> > > >      * Because language comprehension is inherently sequential, most work on encoding (predicting brain activity from text) still uses transformer-based autoregressive LMs, while production/decoding settings have started to explore diffusion-style models as a more abstract generative mechanism.
> > > > * To our knowledge, state-space / Mamba-like models have not yet been systematically explored in this literature.
> > > >
> > > > * Motivated by the reviewer’s suggestion, we now explicitly highlight non-transformer language models as an important direction for future work in the Discussion.
> > > >     * In particular, we propose comparing, under matched training data and scale, transformers vs. state-space models vs. other architectures to test whether the local plateau and quantization robustness we observe are specific to transformer inductive biases, or instead reflect more general properties of high-performing language models.

---

### Author Response · Authors · 2025-11-25
**Common Responses to Reviewers wmV2, xLFX, PMQK, and f24S**

*We are grateful to all reviewers for their positive feedback, time and their constructive suggestions, which will further strengthen the impact of our work.*

**CQ1. Quantitative Analysis across model families**

We quantify scaling differences by performing statistical significance across subjects for the best selective layer per model: **Qwen2.5 in Table 5**, **LLaMA in Table 6** and **DeepSeek in Table 7, in Appendix I**.

* **For Qwen2.5 model**,
  * The resulting mean best-layer scores are: 1.5B: 0.85+0.09, 3B: 0.92+0.08, 7B: 0.895+0.09, 14B: 0.93+0.10.
  * Paired tests over subjects (n = 9) show that 3B and 14B are statistically indistinguishable (mean difference -0.0004, t(8) = -0.03, p ≈ 0.98), while both 3B and 14B significantly outperform the 1.5B model (3B vs. 1.5B: Δ = 0.07, t(8) = 4.89, p ≈ 0.004; 14B vs. 1.5B: Δ = 0.07, t(8) = 3.16, p ≈ 0.025).
  * We also find a modest but significant advantage of 3B and 14B over 7B in best-layer alignment (Δ ≈ 0.04, p ≈ 0.02-0.04).
  * These tests support our main claim in this regime: beyond ~3B, increasing model size up to 14B yields at most modest gains in brain alignment, whereas 1B-1.5B models are reliably worse.

**Table 5**: Qwen best-layer pairwise comparisons across subjects
|Comparison (A – B) | Δ (mean diff) | t(8)  | p (two-sided, approx.) | Interpretation  |
|-|-|-:|-|-|
| 3B – 14B  | -0.000  | -0.00 | 1.000 | 3B ≈ 14B (no difference) |
| 3B – 7B  |  0.028  |  2.43 | 0.059| 3B > 7B (small effect, trend only)     |
| 3B – 1.5B |  0.073 |  4.89 | 0.004 | 3B > 1.5B (clear, significant)         |
| 14B – 7B |  0.028|  2.02 | 0.099 | 14B > 7B (small effect, n.s.)          |
| 14B – 1.5B |  0.073  |  3.16 | 0.025  | 14B > 1.5B (clear, significant)        |
| 7B – 1.5B          |  0.045|  2.67 | 0.045  | 7B > 1.5B (moderate, significant)      |


* **For the LLaMA model**, as shown in Table 6 in Appendix I, we find that
  * 14B ≈ 7B: No difference (Δ = -0.00, p ≈ 0.95) - statistically identical
  * 3B ≈ 14B/7B: Slight advantage but not significant (p > 0.05) and
  * All vs 1B: Highly significant differences (p < 0.001 for 3B and 7B).
  * Overall, the 3B, 7B, and 14B models form a statistically equivalent top tier, all significantly outperforming the 1B model.

**Table 6**: LLaMA best-layer pairwise comparisons across subjects
| Comparison (A – B) | Δ (mean diff) | t(8)  | Sig. (two-sided)        |
|--------------------|---------------|------:|-------------------------|
| 3B – 14B           | 0.03          |  2.50 | n.s. (p ≈ 0.05)         |
| 3B – 7B            | 0.03          |  1.97 | n.s. (p ≈ 0.11)         |
| 3B – 1B            | 0.10          |  6.89 | p < 0.001               |
| 14B – 7B           | -0.00         | -0.07 | n.s. (p ≈ 0.95)         |
| 14B – 1B           | 0.07          |  3.12 | p < 0.05                |
| 7B – 1B            | 0.07          | 11.43 | p < 0.001               |


* **Analysis of DeepSeek models (14B, 7B, 3B, 1B parameters)**, as shown in Table 7 in Appendix I reveals a clear scaling hierarchy with the 14B and 3B models forming a statistically equivalent top tier. We make the following observations from Table 7:
  * 14B ≈ 3B: No significant difference (p ≈ 0.61), indicating 3B achieves 14B-level performance with ~80% fewer parameters!
  * 14B, 3B >> 7B: Highly significant advantages (p < 0.01),
  * All >> 1B: Very large differences (all p < 0.001).

**Table 7**: Deepseek-r1 best-layer pairwise comparisons across subjects
| Comparison (A – B) | Δ (mean diff) | t(8)  | Sig. (two-sided)  |
|--------------------|---------------|------:|-------------------|
| 14B – 3B           | -0.01         | -0.54 | n.s. (p ≈ 0.61)   |
| 14B – 7B           |  0.07         |  4.84 | p < 0.01          |
| 14B – 1B           |  0.19         | 11.18 | p < 0.001         |
| 3B – 7B            |  0.08         |  4.18 | p < 0.01          |
| 3B – 1B            |  0.19         | 10.85 | p < 0.001         |
| 7B – 1B            |  0.11         |  9.28 | p < 0.001         |

Overall, across three independent model families (Qwen, LLaMA, DeepSeek) and using best-layer scores with paired tests over nine subjects, we find a consistent pattern:
* 1B-1.5B models are reliably worse in brain alignment, while 3B models already reach the same level as their 7B-14B counterparts.
* In Qwen and DeepSeek, 3B and 14B are statistically indistinguishable, whereas both significantly outperform the smallest models; in LLaMA, 3B and 14B again lie in a narrow, non-significantly different range, with 7B closely tracking 14B and clearly above 1B.
* These results do not overturn global scaling laws, but they do indicate a local plateau in the compressed 1-14B regime: once model capacity reaches ~3B, further scaling yields at most modest gains in brain predictivity, while going below this threshold leads to a robust drop in alignment.

We have added this discussion in **Appendix I** of the revised paper.

---

> ### Author Response · Authors · 2025-11-25
>
> **CQ2. Statistical Validation of Quantization Effects (Reviewers: wmV2, xLFX, PMQK, and f24S)**
>
> * Based on the reviewer’s suggestion, we performed paired t-tests to compare normalized brain alignment between quantization methods and their respective base models across all language ROI-layer combinations and subjects (n=9).
> * To control for multiple comparisons, we applied False Discovery Rate (FDR) correction using the Benjamini-Hochberg procedure (α = 0.05). Effect sizes were calculated using Cohen's d.
> * We report both raw p-values and FDR-corrected significance, along with 95% confidence intervals for mean differences.
>
> **Quantization Effects: Qwen2.5**
>
> * We now provide formal statistical tests and variability measures for the quantization comparisons.
>     * For each Qwen2.5 model (1.5B, 3B, 7B), and for each quantization method (FP16, AWQ, GPTQ, SmoothQuant), we compute best-layer brain alignment per subject and run paired $t$-tests across subjects between methods (**Table 8 in Appendix J**). Negative Δ in rows of the form “FP16–AWQ” indicates that AWQ outperforms FP16.
>     * For Qwen2.5–7B (**Table 8, left**), AWQ and SmoothQuant are significantly better than both FP16 and GPTQ (e.g., FP16–AWQ: Δ = -0.020, $t(8) = -6.10$, $p < 0.001$; AWQ–GPTQ: Δ = 0.040, $t(8) = 7.10$, $p < 0.001$), while GPTQ is significantly worse than FP16.
>     * For Qwen2.5–3B (**Table 8, right**), none of the quantized variants differ significantly from FP16, but AWQ and SmoothQuant significantly outperform GPTQ, suggesting that well-designed quantization preserves alignment whereas GPTQ exhibits a modest degradation.
>     * For Qwen2.5–1.5B (**Table 8, bottom**), AWQ is significantly better than FP16 (Δ = -0.024$, $t(8) = -4.04$, $p < 0.01$), whereas GPTQ and SmoothQuant are statistically indistinguishable from FP16, and differences among the three quantized variants do not reach significance after correction.
>
> Below Tables provide pairwise comparisons of brain-alignment differences across quantization methods for Qwen2.5-7B, Qwen2.5-3B and Qwen2.5-1.5B.
>
> **Table 8 (left) in Appendix J**
> |Comparison (A – B)|Δ (mean diff)|t(8)|Sig. (two-sided)|
> |-|-|-:|-|
> |Qwen2.5-7B– AWQ|  -0.020|   -6.10   | p < 0.001|
> |Qwen2.5-7B– GPTQ|   0.020|   6.20  | p < 0.001|
> |Qwen2.5-7B– SmoothQuant|-0.005       |   -3.50  | p < 0.016        |
> |AWQ – GPTQ|   0.040|   7.10  | p < 0.001        |
> |AWQ – SmoothQuant|0.015       |   4.20  | p < 0.008        |
> |GPTQ – SmoothQuant|  -0.025       |   -4.90  | p < 0.004        |
>
> **Table 8 (right) in Appendix J**
> | Comparison (A – B)      | Δ (mean diff) |  t(8) | Sig. (two-sided)    |
> |-|-|-:|-|
> | Qwen2.5-3B– AWQ  |-0.010 | -1.23 | n.s. (p ≈ 0.28)     |
> | Qwen2.5-3B– GPTQ | 0.014|  2.24 | n.s. (p ≈ 0.08)  |
> | Qwen2.5-3B– SmoothQuant|  -0.007 | -1.55 | n.s. (p ≈ 0.18)     |
> | AWQ – GPTQ |0.024|  3.10 | p < 0.05|
> | AWQ – SmoothQuant  |0.003|  0.34 | n.s. (p ≈ 0.75) |
> | GPTQ – SmoothQuant | -0.020| -3.12 | p < 0.05 |
>
> **Table 8 (bottom) in Appendix J**:
> | Comparison (A – B) | Δ (mean diff) |  t(8) | Sig. (two-sided)|
> |-|-|-:|-|
> |  Qwen2.5-1.5B– AWQ |  -0.024| -4.04 | p < 0.01 |
> |  Qwen2.5-1.5B– GPTQ  |0.002|  0.18 | n.s. (p ≈ 0.86) |
> |  Qwen2.5-1.5B– SmoothQuant|  -0.004| -0.71 | n.s. (p ≈ 0.51)|
> | AWQ – GPTQ | 0.026 |  2.03 | n.s. (p ≈ 0.10)|
> | AWQ – SmoothQuant |   0.020 |  2.43 | n.s. (p ≈ 0.06) |
> | GPTQ – SmoothQuant|  -0.006 | -0.42 | n.s. (p ≈ 0.69)|
>
>
> * We also summarize quantization performance at the level of mean ± standard deviation across subjects in **Table 9 in Appendix J**.
>     * Across all Qwen sizes, AWQ and SmoothQuant closely or slightly exceed full models (FP16) in mean best-layer alignment (differences on the order of 0.01–0.02, within one standard deviation), whereas GPTQ tends to be lower than FP16, especially for 7B and 3B.
> * Together, **Table 8 and Table 9 in Appendix J** show that:
>     * some apparent improvements in the figures are within noise and are now explicitly reported as non-significant, and
>     * the main qualitative pattern is statistically supported: well-designed quantization (AWQ/SmoothQuant) preserves brain alignment at near-full-precision levels, while GPTQ produce a modest but reliable degradation.
>
> **Table 9**: Quantization method performance across Qwen models (mean ± std over 9 subjects): AWQ > SmoothQuant > FP16 > GPTQ
> | Model |FP16 (mean ± std)| AWQ (mean ± std)| GPTQ (mean ± std)| SmoothQuant (mean ± std) |
> |-|-|-|-|-|
> | Qwen-7B  | 0.886 ± 0.092 | 0.906 ± 0.092| 0.866 ± 0.092 | 0.891 ± 0.092 |
> | Qwen-3B  | 0.923 ± 0.080 | 0.933 ± 0.085| 0.910 ± 0.091 | 0.930 ± 0.085  |
> | Qwen-1.5B| 0.850 ± 0.087 | 0.874 ± 0.099 | 0.848 ± 0.088| 0.854 ± 0.084|
>
> We also present quantization methods comparison and quantization brain alignment impact for Qwen2.5 model in **Fig.24 in Appendix J**.
>
> * We also present a similar analysis for the **LLaMA model in Tables 10-13 and Fig. 25 in Appendix J**.
>
> We have added this discussion in **Appendix J** of the revised paper.

---

> ### Author Response · Authors · 2025-11-25
>
> **CQ3. ROI-Specific Analysis and Best Layer Selection (Reviewers: wmV2, xLFX, PMQK, and f24S)**
>
> * In our analyses, we extract activations from every transformer layer, and fit a separate voxel-wise encoding model for each layer.
> * For each model, we then compute brain alignment layer-by-layer across the language ROIs and identify the best layer as the one with the highest mean normalized predictivity.
> * The main size/quantization comparisons are reported using this model-specific best layer (**see Tables 13 and 16 in the Appendix K**).
>
> * From **Table 13**, across language ROIs we find that the best layers are highly consistent within a given model: the same (or adjacent) layer tends to be optimal across ROIs, so we treat the best layer as a model-level property when summarizing results.
> * Overall, across models we observe the familiar pattern that middle-to-late layers yield the strongest brain alignment, with early layers performing clearly worse.
> * Mostly 3b model is the best in terms of brain alignment across ROIs
>
> **Table 13 in Appendix K**: ROI-Specific Layer Performance Summary for Qwen2.5
>
> | ROI  | Model | Overall Mean±SD| Best Layer | Best Layer Mean±SD | Worst Layer | Layer Range      |
> |-|-|-|-|-|-|-|
> | AG | 1.5  | 0.834±0.1546 | L14 | 0.9466±0.1099 | L1 | 0.5478-0.9466 |
> | AG | 3b   | **0.855±0.1823** | L22 | 1.0091±0.1157 | L1 | 0.5418-1.0091 |
> | AG | 7b   | 0.841±0.2119 | L15 | 0.9818±0.1390 | L1 | 0.4535-0.9818 |
> | AG | 14b  | 0.817±0.2085 | L24 | **1.0143±0.1431** | L1 | 0.4913-1.0143 |
> | ATL | 1.5  | 0.716±0.1248 | L14 | 0.7904±0.1302 | L1 | 0.5296-0.7904 |
> | ATL | 3b   | **0.725±0.1422** | L21 | 0.8304±0.1309 | L1 | 0.5108-0.8304 |
> | ATL | 7b   | 0.701±0.1611 | L16 | 0.7891±0.1072 | L1 | 0.4620-0.7891 |
> | ATL | 14b  | 0.697±0.1579 | L25 | **0.8362±0.1333** | L1 | 0.4630-0.8362 |
> | PTL | 1.5  | 0.769±0.1300 | L15 | 0.8335±0.1078 | L1 | 0.5944-0.8335 |
> | PTL | 3b   | **0.776±0.1394** | L21 | **0.8725±0.1313** | L1 | 0.5843-0.8725 |
> | PTL | 7b   | 0.744±0.1616 | L15 | 0.8353±0.1231 | L3 | 0.5166-0.8353 |
> | PTL | 14b  | 0.7471±0.1513 | L25 | 0.8592±0.1386 | L1 | 0.5353-0.8592 |
> | IFG | 1.5  | 0.7726±0.1716 | L14 | 0.8730±0.1480 | L1 | 0.5366-0.8730 |
> | IFG | 3b   | **0.7801±0.1872** | L21 | 0.9309±0.1377 | L1 | 0.5058-0.9309 |
> | IFG | 7b   | 0.7665±0.1996 | L15 | 0.9045±0.1305 | L3 | 0.4263-0.9045 |
> | IFG | 14b  | 0.7639±0.2079 | L25 | **0.9516±0.1027** | L1 | 0.4938-0.9516 |
> | MFG | 1.5  | 0.6929±0.1494 | L15 | 0.7618±0.1540 | L1 | 0.5068-0.7618 |
> | MFG | 3b   | **0.6984±0.1732** | L21 | **0.8006±0.1746** | L1 | 0.5213-0.8006 |
> | MFG | 7b   | 0.6716±0.1883 | L15 | 0.7732±0.1553 | L1 | 0.4283-0.7732 |
> | MFG | 14b  | 0.6689±0.1642 | L24 | 0.7921±0.1506 | L1 | 0.4682-0.7921 |
> | IFGOrb | 1.5  | 0.6193±0.1589 | L14 | 0.7249±0.1007 | L1 | 0.4160-0.7249 |
> | IFGOrb | 3b   | 0.6401±0.1712 | L22 | 0.7647±0.0866 | L1 | 0.4072-0.7647 |
> | IFGOrb | 7b   | **0.6403±0.1966** | L15 | 0.7765±0.1080 | L1 | 0.2891-0.7765 |
> | IFGOrb | 14b  | 0.6159±0.1878 | L25 | **0.7803±0.0772** | L3 | 0.3628-0.7803 |
> | PCC | 1.5  | 0.7638±0.1732 | L15 | 0.8618±0.1261 | L1 | 0.4904-0.8618 |
> | PCC | 3b   | **0.7651±0.1878** | L22 | 0.9093±0.1203 | L1 | 0.4654-0.9093 |
> | PCC | 7b   | 0.7509±0.2118 | L15 | 0.8904±0.0938 | L1 | 0.3571-0.8904 |
> | PCC | 14b  | 0.7235±0.2095 | L24 | **0.9367±0.1163** | L1 | 0.4205-0.9367 |
> | dmPFC | 1.5  | 0.6884±0.1392 | L14 | 0.8089±0.0991 | L1 | 0.4443-0.8089 |
> | dmPFC | 3b   | **0.6964±0.1726** | L21 | 0.8685±0.1049 | L1 | 0.4192-0.8685 |
> | dmPFC | 7b   | 0.6888±0.1825 | L15 | 0.8226±0.1018 | L1 | 0.3628-0.8226 |
> | dmPFC | 14b  | 0.6692±0.1913 | L25 | **0.8859±0.1320** | L1 | 0.3712-0.8859 |
> | AC | 1.5  | 0.5587±0.0906 | L15 | 0.5963±0.0762 | L1 | 0.4727-0.5963 |
> | AC | 3b   | **0.5634±0.1056** | L21 | **0.6303±0.0890** | L1 | 0.4668-0.6303 |
> | AC | 7b   | 0.5204±0.1243 | L15 | 0.5867±0.1017 | L3 | 0.3757-0.5867 |
> | AC | 14b  | 0.5241±0.0909 | L24 | 0.5831±0.1042 | L1 | 0.4095-0.5831 |
>
>
> **Optimal Layers Identified**
>
> **Table**: Optimal layer selection for Qwen2.5
> | Model | Optimal Layer | Mean Alignment at Layer | Rationale  |
> |-|-|-|-|
> | 1.5B  | Layer 14 (of 28) | 0.7956  | Peak at ~50% depth|
> | 3B    | Layer 22 (of 36) | 0.8309| Peak at ~61% depth|
> | 7B    | Layer 15 (of 28) | 0.8171 | Peak at ~54% depth     |
> | 14B   | Layer 24 (of 40) | 0.8411 | Peak at ~60% depth     |
>
> * Overall, when comparing models at their optimal layers, there are no significant differences across brain regions after controlling for multiple comparisons for 7B vs 3B, 14B vs 3B and 14B vs 7B models.
>
> * We also present a similar analysis for the **LLaMA model in Tables 16-17 and Fig. 25 in Appendix K**.
>
> We have added this discussion in **Appendix K** of the revised paper.

---

> ### Author Response · Authors · 2025-11-25
>
> **CQ4. Subject Variability at Optimal Layers (Reviewers: wmV2, xLFX, PMQK, and f24S)**
>
> **Table 14. in Appendix K** Qwen2.5: Subject Variability at Optimal Layers
> | Model | Layer | Mean ± SD        | SEM     | 95% CI           | CV (%)  | Variability |
> |-|-|-|-|-|-|-|
> | 1.5B  | L14    | 0.7956 ± 0.0892   | 0.0364  | [0.724, 0.867]    | 11.21%  | LOW         |
> | 3B    | L22    | 0.8309 ± 0.0835   | 0.0341  | [0.764, 0.898]    | 10.05%  | LOW         |
> | 7B    | L15    | 0.8171 ± 0.0852   | 0.0348  | [0.749, 0.885]    | 10.43%  | LOW         |
> | 14B   | L24    | 0.8411 ± 0.0883   | 0.0361  | [0.770, 0.912]    | 10.50%  | LOW         |
>
> * From Table 14, we observe that subject variability at optimal layers was low across all models (CV < 11.5%), with 3B showing the highest consistency (CV = 10.05%) and 14B achieving the highest mean alignment (0.8411±0.088)."
>
> **Table 17. in Appendix K** LLaMA-3: Subject Variability at Optimal Layers
>
> | Model | Layer | Mean ± SD       | SEM    | 95% CI         | CV (%) | Variability |
> | ----- | ----- | --------------- | ------ | -------------- | ------ | ----------- |
> | 1B    | L8    | 0.8003 ± 0.1065 | 0.0435 | [0.688, 0.912] | 13.31% | MODERATE    |
> | 3B    | L13   | 0.9083 ± 0.0824 | 0.0337 | [0.822, 0.995] | 9.08%  | LOW         |
> | 8B    | L10   | 0.8588 ± 0.1115 | 0.0455 | [0.742, 0.976] | 12.98% | MODERATE    |
> | 14B   | L16   | 0.8679 ± 0.0936 | 0.0382 | [0.770, 0.966] | 10.78% | MODERATE    |
>
> * From Table 17, we observe that LLaMA-3B shows the best consistency across subjects (metrics), matching Qwen's pattern where 3B models tend to have optimal variability characteristics.
>
> **CQ5. Encoding Performance of DeepSeek Model (Reviewers: wmV2, xLFX, PMQK, and f24S)**
>
> * Based on reviewer’s suggestion, We have now extended our evaluation to an additional model family: Deepseek-R1-Distill at 1.5B, 3B, 7B and 14B.
> * The normalized brain alignment of DeepSeek models across whole brain and language ROI (IFG) is presented in **Fig. 26 (left) in Appendix L**.
> * We also group the quantized variants and present a comparison of the base vs. quantized models across language ROIs in **Fig 26. (right) in Appendix L**.
> * From Fig 26, we observe the same trend:
>    * 3B SLMs maintain brain alignment comparable to 14B models, while 1B-1.5B models consistently drop in brain alignment.
>    * Notably, the DeepSeek-R1-Distill 14B model shows only a modest improvement over the 7B version, and its alignment is matched by the 3B DeepSeek model, suggesting that 3B SLMs provide sufficient representational capacity for studying brain-LM alignment within this scale regime.
>    * We also present the average normalized brain alignment across Language ROIs for DeepSeek-R1 model, comparing SLMs, LLMs, and individual quantized variants in Fig. 27  and the grouped quantized variants in Fig.28.
>
>
> We have added this discussion in **Appendix K** and **Appendix L** of the revised paper.

---

### Author Response · Authors · 2025-12-01
**Summary of our responses and revision:**

We are grateful to all reviewers for their positive feedback, time and constructive suggestions, which will further strengthen the impact of our work.

**Summary of Reviewer Strengths:**

1. Novelty: Timely and relevant topic. To the reviewers’ knowledge, the first study to systematically evaluate how scaling and compression strategies affect the brain-alignment properties of language models. (**all reviewers**)
2. Extensive experiments across multiple model families, quantization methods, and brain regions. (**Reviewers: wmV2, xLFX, PMQK**)
3. Clear distinction between brain alignment and linguistic task performance, with nuanced interpretation of their divergence. (**Reviewer: PMQK**)
4. Comprehensive evaluation design to bridge neural and computational levels of analysis and identifies which linguistic properties support brain alignment (**Reviewers: wmV2, xLFX**).
5. Impact for NeuroAI: Offers practical recommendations for neuroAI applications, especially for choosing between SLMs, LLMs, and compressed models. (**Reviewers: xLFX, PMQK**)
6. Good presentation: paper is clearly written and carefully structured, with clear visualizations. (**Reviewers: wmV2, f24S**)

**Additional changes to the draft during the rebuttal process**

We have updated the main manuscript and the appendix to address these following comments. The changes made in the manuscript are highlighted in blue color. The major additional changes are listed below.

1. **Scaling analysis across model families (all reviewers)**
	* We extended our evaluation to larger models (up to 14B) and an additional family, DeepSeek-R1-Distill, alongside Qwen2.5 and LLaMA-3.
	* For each family, we now report best-layer brain alignment across subjects for 1B–14B, and perform paired statistical tests (**Tables 5–7 in Appendix I**) comparing 3B vs 7B vs 14B, and vs 1B/1.5B.
	* Across three model families, we consistently find that 3B models match similar brain alignment as 7B–14B models, while 1B–1.5B models show a significant drop in brain alignment, supporting a local capacity threshold around ~3B.

2. **Statistical validation of scaling and quantization (all reviewers)**
	* We added paired $t$-tests across subjects to quantify differences in brain alignment between:
	  * different model sizes (1B/1.5B vs 3B vs 7B/8B vs 14B) (**Tables 5-7 in Appendix I**), and
	  * different quantization schemes (FP16 vs AWQ vs GPTQ vs SmoothQuant; **Table 8 for Qwen**, **Table 10 for LLaMA**).
	* These tests confirm that 3B models are statistically indistinguishable from 7B–14B, 1B–1.5B models are significantly worse, and AWQ/SmoothQuant generally preserve brain alignment, whereas GPTQ shows a modest but reliable degradation.

3. **ROI-specific analysis and best-layer selection (all reviewers)**
	* We compute brain alignment layer-by-layer across the language ROIs and identify the best layer as the one with the highest mean normalized predictivity (see **Tables 13 and 16 in the Appendix K**).
	* Across models, we observe the familiar pattern that middle-to-late layers yield the strongest brain alignment, with early layers performing clearly worse.

4. **Subject variability at optimal layers (all reviewers)**
   * We added subject-variability tables for all families (Qwen, LLaMA, DeepSeek), reporting mean ± SD, SEM, 95% CI, and CV at the brain-optimal layer, showing low variability across subjects (**Tables 14 and 17 in Appendix K**).

5. **Additional brain dataset (Reviewers: wmV2, PMQK, f24S)**
	* We extended our experiments to the Moth Radio Hour Reading fMRI dataset (**Appendix L Fig. 29**).
	* The same pattern holds: 3B SLMs match 7B/14B Qwen2.5 models, whereas 1.5B models show a clear drop in brain alignment.

6. **Quantization vs Pruning: (Reviewer: PMQK)**
	* Beyond quantization, we now include unstructured magnitude pruning for Qwen2.5-3B and Qwen2.5-1.5B at 10%, 25%, and 50% sparsity (See **Tables 20-21 in Appendix O**).
	* Results show that moderate pruning (10–25%) preserves brain alignment at near–full-precision levels, while 50% pruning causes a substantial drop, especially for 1.5B.

7. **Brain decoding experiments (Reviewer: PMQK)**
	* We added preliminary brain->text decoding experiments using LLaMA-3.2-8B, 3B, and 1B on the Moth-Radio-Hour stimuli (**Tables 18-19 in Appendix M**).
	* We find that LLaMA-3.2-3B slightly outperforms 8B and clearly outperforms 1B on BLEU-1, WER, METEOR, with BERT-F1 in the 0.81–0.83 range, indicating preserved semantic content.

8. **Clarified limitations and scope (Reviewers: wmV2, xLFX)**
	* We moved and strengthened the Limitations & Future Work section in the main paper.
	* We focus on post-training quantization (with preliminary pruning) do not fully capture large-scale scaling laws, training-data effects, or alternative architectures/modalities. These are now clearly framed as directions for future work.

---

### Note · Authors · 2026-01-29

I have read and agree with the venue's withdrawal policy on behalf of myself and my co-authors.

---

### Meta-Review · Area_Chair_QkKB · 2025-12-28

**Summary:**

* The paper would likely have landed just below or around the acceptance threshold. Methodological concerns were largely resolved  (scaling to 14B, new dataset, statistical tests, pruning and decoding - exceptional rebuttal quality!), but perceived contribution / impact remained moderate.
* Main reason scores remained conservative: several reviewers would still have perceived the contribution as incremental / benchmarking-like, with limited novelty and no analysis beyond transformer / fMRI settings.

**Reviewer Concerns:**

Reviewer wmV2:
* [Addressed] Added statistical validation for scaling and quantization effects (appendix I, J), extended model scale up to 14B and added DeepSeek-R1-Distill family, added second brain dataset (Moth Radio Hour Reading) confirming generalization.
* [Not addressed] No systematic analysis of training data diversity or non-transformer architectures.

Reviewer xLFX:
* [Addressed] Added formal statistical tests, confidence intervals, and subject variability analyses, ROI-specific, layer-wise analysis and consistent best-layer selection, extended scaling to 14B models. Improved presentation clarity.
* [Not addressed] No causal or perturbation-based interpretability analysis linking linguistic properties to specific ROIs.

Reviewer PMQK:
* [Addressed] Extended evaluation to 14B models, unstructured pruning experiments, brain decoding experiments.
* [Not addressed] No MEG/ECoG experiments (temporal dynamics deferred), no systematic comparison of structured pruning or distillation methods under matched ratios.

Reviewer f24s:
* [Addressed] Quantified effect sizes and statistical significance,  reported model sizes and compression ratios, figure readability.

**Reviewer Scores:**

* Reviewer wmV2: Would have kept the score --> 4
* Reviewer xLFX: Would have increased the score --> 4 (from 2)
* Reviewer PMQK: Would have increased the score --> 4 (from 2)
* Reviewer f24s: Would have increased the score --> 6 (from 4)

---

### Decision · Program_Chairs · 2026-01-26

Reject